# The DNA virome varies with human genes and environments

Nolan Kamitaki[1,2,3,4,5,6] ✉, David Tang[1,2,3,4], Steven A. McCarroll[3,5,6,7] ✉ & Po-Ru Loh[1,2,3] ✉

Many viruses have adapted to persist in infected humans for life[1,2]. Variable host control of their ongoing abundance (viral load) can lead to clearance or disease[3–5]. Here we analysed the viral DNA load of 31 common viruses in human blood and saliva using whole-genome sequencing data from UK Biobank ($n = 490,401$), All of Us ($n = 414,817$) and Simons Foundation Powering Autism Research for Knowledge (SPARK; $n = 12,519$). Viral DNA load varied markedly with age, time of day and season; most viruses were also present at greater abundance in men than in women. Human genetic variation at dozens of loci associated with DNA load of seven viruses: Epstein–Barr virus (EBV, 45 loci), human herpesvirus (HHV)-7 (37 loci), HHV-6B, Merkel cell polyomavirus and three anelloviruses. Variation at the major histocompatibility complex (MHC) locus generated the strongest associations ($P = 5.8 \times 10^{-9}$ to $2.5 \times 10^{-1459}$), which were specific to each virus. The *HLA-B*08:01* allele also exhibited a host–virus genetic interaction with EBV subtype ($P = 7.4 \times 10^{-70}$). Other human genetic effects implicated genes encoding proteins that process peptides for antigen presentation, such as *ERAP1* (HHV-7, $P = 2.7 \times 10^{-78}$) and *ERAP2* (EBV, $P = 4.6 \times 10^{-111}$). Mendelian randomization analyses supported a strong causal effect of EBV DNA load on increased risk of Hodgkin's lymphoma ($P = 1.8 \times 10^{-3}$), but not multiple sclerosis ($P = 0.52$). This suggests that higher chronic EBV load increases lymphoma risk, whereas associations of EBV infection with autoimmune conditions reflect host immune responses to particular viral epitopes.

After infection by a virus, the long-term balance between viral replication and antiviral immune recognition within a person can result in spontaneous clearance[3], stable persistence around a set point of viral load, as with lifelong EBV infection[4], or progression to disease, such as the progression from human immunodeficiency virus (HIV) infection to acquired immunodeficiency syndrome (AIDS)[5]. EBV infection is a large risk factor for Hodgkin's lymphoma[6] and multiple sclerosis[7] years after primary infection, and is associated with other autoimmune disorders for reasons that are not well understood[8].

Large-scale DNA sequencing studies of blood samples from thousands of individuals have begun to characterize the blood DNA virome—the population of DNA viruses and retroviruses that is resident within each person's blood system—which spans a broad spectrum of viral species[9,10]. Two commonly observed taxa are herpesviruses, which are latently present in most humans, and anelloviruses, which are thought to be commensal or even symbiotic[11]. The amount of a virus present within an individual—viral load—varies over time and is a potential biomarker of immune function[12–14]. Increased viral load may indicate reduced immunocompetence and can lead to disease (such as AIDS or liver cancer from viral hepatitis[15]), either as a direct result of inadequate control of an existing virus or through susceptibility to new infection.

The extent to which viral load in blood—and in other tissues such as saliva—varies among individuals and across the lifespan is less well characterized, and the factors that drive such variation—and its potential effects on human health—are largely unknown. Human genetic variants in the MHC region on chromosome 6 and a few other genomic loci have been shown to influence load of HIV[16,17] and hepatitis C[18], and chronic persistence (versus spontaneous clearance) of hepatitis B[19] and C[20]. Less is known about variation in load of those viruses that commonly reside latently in healthy individuals, although immune (antibody) response to EBV and other common viruses is known to be influenced by host genetics[21,22]. The drivers of viral load variation in tissues other than blood also remain largely unexplored.

## The DNA virome in blood and saliva

Many DNA viruses transit between latent infection and lytic activation in blood and saliva. Blood and saliva are also the two primary sources of genomic DNA in human population genetic studies. To quantify viral DNA load from widely available whole-genome sequencing (WGS) data, we analysed sequencing reads that were previously generated from high-coverage WGS (mean 32.5× to 42×) of 490,401 blood samples in UK Biobank[23] (UKB), 365,918 blood samples and 48,899 saliva samples in the All of Us (AoU) dataset[24] and 12,519 saliva samples in SPARK[25] (Fig. 1a). We aligned sequencing reads of potential non-human origin to a reference panel that we assembled of 31 common viruses (27 DNA viruses

[1]Division of Genetics, Department of Medicine, Brigham and Women's Hospital and Harvard Medical School, Boston, MA, USA. [2]Center for Data Sciences, Brigham and Women's Hospital, Boston, MA, USA. [3]Program in Medical and Population Genetics, Broad Institute of MIT and Harvard, Cambridge, MA, USA. [4]Department of Biomedical Informatics, Harvard Medical School, Boston, MA, USA. [5]Stanley Center for Psychiatric Research, Broad Institute of MIT and Harvard, Cambridge, MA, USA. [6]Department of Genetics, Harvard Medical School, Boston, MA, USA. [7]Howard Hughes Medical Institute, Harvard Medical School, Boston, MA, USA. ✉e-mail: nolan_kamitaki@hms.harvard.edu; smccarro@broadinstitute.org; poruloh@broadinstitute.org

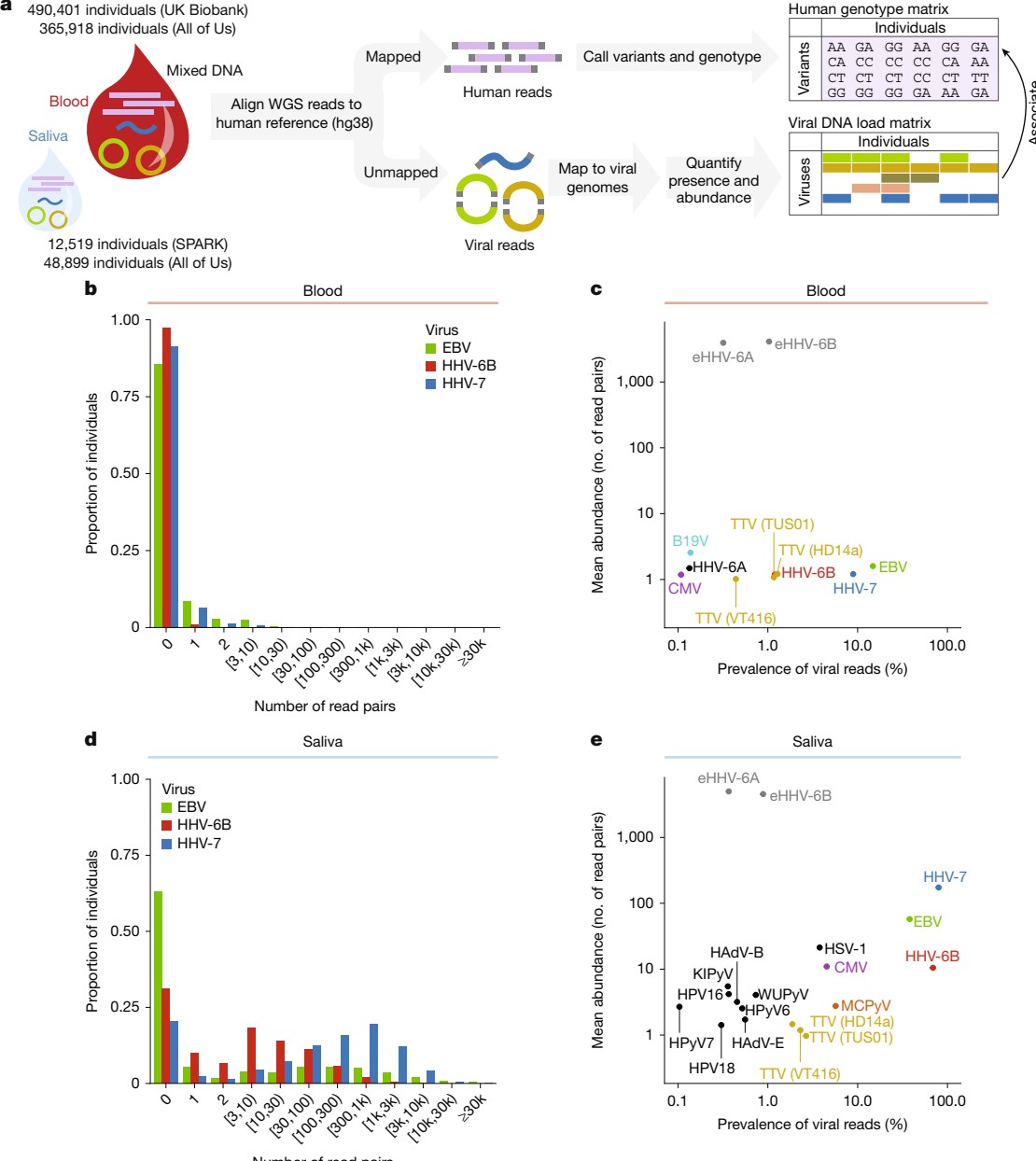

**Fig. 1 | Detection and quantification of viral DNA load in blood and saliva WGS. a**, Detection (presence or absence) and quantification (abundance) of viral DNA sequences in WGS data derived from human blood and saliva samples, including blood samples from the UKB ($n = 490,401$) and AoU ($n = 365,918$) cohorts and saliva samples from the SPARK ($n = 12,519$) and AoU ($n = 48,899$) cohorts. Sequence reads that did not map to the human genome were aligned against a reference panel of 31 viral genomes to identify and tabulate viral DNA fragments. **b**, Distribution of viral DNA fragment counts (read pairs) from EBV, HHV-7 and HHV-6B genomes in blood-derived WGS data from UKB participants ($n = 490,401$). **c**, Prevalence (population frequency) and mean abundance of viral DNA sequences in UKB blood WGS data. Mean abundance indicates the geometric mean of the number of observed read pairs among samples in which the virus was detected. For HHV-6, data from carriers of chromosomally integrated endogenous HHV-6A and HHV-6B are indicated separately in grey. Viruses that appear in subsequent figures are indicated with colours. CMV, cytomegalovirus. **d**, Distribution of viral DNA fragment counts (read pairs) from EBV, HHV-7 and HHV-6B genomes in saliva-derived WGS data from SPARK participants ($n = 12,519$). **e**, Prevalence and abundance of viral DNA sequences in SPARK saliva WGS data. HAdV-B, human adenovirus species B; HAdV-E, human adenovirus species E; HPV16, human papillomavirus 16; HPyV6, human polyomavirus 6; HSV, herpes simplex virus; KIPyV, KI polyomavirus; WUPyV, WU polyomavirus.

and 4 retroviruses; Extended Data Fig. 1a and Supplementary Table 1), selected on the basis of earlier WGS-based blood virome studies[9,10]. In the following analyses, we use the term 'prevalence' to refer to the proportion of research participants with at least one WGS read pair that aligns to a given viral genome and 'abundance' to mean the number of such read pairs from a given biological sample. The viral DNA fragments counted in these measurements of viral DNA load could be derived from viral genomes present within host cells or from cell-free

infectious particles (virions) present in bodily fluids. In blood, viral DNA is known to be more frequently host-cell-derived for EBV[26] and torque teno viruses (TTVs; a common form of anelloviruses)[14].

As in other cohorts[9,10], viral DNA was infrequently detected in blood WGS data from UKB and AoU, with most individuals having no sequencing reads that could be confidently attributable to any virus (Fig. 1b,c, Extended Data Fig. 1b and Supplementary Tables 2 and 3). This was expected despite the near-ubiquity of herpesviruses and anelloviruses

in humans across populations[11] (for example, 94.7% EBV seroprevalence in UKB), given the low fractions of latently infected cells that carry viral DNA[4] (Methods): viral reads are stochastically observed at rates proportional to the number of copies of a viral genome present in a person's blood. Despite this sparsity, several properties of the dataset, such as enrichment of viral reads in seropositive individuals and consistent depth-of-coverage within viral genomes, supported genuine viral origin of these reads (Extended Data Figs. 1c,d and 2 and Supplementary Note 1). Additionally, a large majority of the sequencing reads ascertained to be of viral origin had been aligned to a viral reference genome with high mapping quality (Supplementary Note 2). Different viruses tended not to be co-observed much more frequently than by chance, with the exception of anelloviruses (Extended Data Fig. 1e). A small fraction of blood samples (675 out of 490,401 in UKB) exhibited unusually large numbers of parvovirus B19-derived reads[9,10]; these donors had greatly depleted reticulocyte counts ($P = 1.4 \times 10^{-67}$; Extended Data Fig. 1f), consistent with active viral infection of red blood cell precursors by B19V[27]. High, approximately 15× coverage of the HHV-6A or HHV-6B genome by WGS reads was also observed in around 1% of donors (Fig. 1c and Extended Data Fig. 1b,g), indicating inheritance of haplotypes containing copies of these viral genomes that have integrated into human chromosomes (endogenous HHV-6 (eHHV-6))[28]. These carriers (0.32% of donors with eHHV-6A and 1.03% with eHHV-6B, comparable to previous work[28,29]) were excluded from analyses of HHV-6 viral DNA load.

In contrast to blood, saliva WGS data contained viral DNA sequences that were highly prevalent and abundant in SPARK (Fig. 1d,e and Supplementary Table 4) and AoU (Extended Data Fig. 1h and Supplementary Table 5) data. Several herpesviruses exhibited prominent bimodal distributions of observed viral read counts, with saliva WGS samples tending to either contain no reads from the virus or to contain many such reads (tens, hundreds or thousands; Fig. 1d). This suggests that for these viruses that infect more than 90% of people, viral DNA in saliva originates not from constantly present latent infections, but rather from frequent bursts of active (lytic) viral replication and release that vary widely in intensity. For HHV-7, these lytic episodes appear to occur continuously, such that HHV-7 DNA was detectable in nearly all saliva samples from adults (Extended Data Fig. 1h), whereas for EBV, viral shedding in saliva appears to be common but intermittent, with roughly half of saliva samples exhibiting no EBV DNA (Fig. 1d). In contrast to herpesviruses, DNA from anelloviruses was detected in saliva only modestly more frequently than in blood, typically at low abundances (one or a few reads; Fig. 1c,e), consistent with anelloviruses having a distinct replication cycle[30]. Several viruses that were rarely detected in blood (less than 0.1% prevalence) were detected in saliva at low to moderate prevalence (Fig. 1c,e). Merkel cell polyomavirus (MCPyV)—which commonly infects skin cells and can cause Merkel cell carcinoma—was seen in 5.7% of saliva samples (Fig. 1e), supporting a previous study[31]. Endogenous HHV-6A and HHV-6B were observed in SPARK at rates similar to UKB (0.37% and 0.78%, respectively; Fig. 1e and Extended Data Fig. 1i). Viral DNA load levels were broadly independent of autism spectrum disorder status in SPARK (Extended Data Fig. 1j,k) and were consistent between SPARK and AoU saliva samples (Fig. 1e and Extended Data Fig. 1h).

## Viral DNA load varies with age, sex and time

The measurement of the prevalence and abundance of diverse viruses in the blood and saliva of so many people made it possible to recognize correlations with age, sex and circadian and seasonal dynamics through cross-sectional analyses of the large cohort. Age and sex exhibited strong associations with viral DNA load (Fig. 2a–d). In blood samples from UKB, which recruited participants aged 40–70 years, viral DNA prevalence of EBV and TTVs was greater in older individuals ($P = 6.7 \times 10^{-356}$ and $1.4 \times 10^{-451}$, respectively; regression), as recently observed for TTVs[14], whereas viral DNA prevalence was greater in younger individuals for HHV-7 ($P = 6.7 \times 10^{-200}$) and HHV-6B

($P = 1.7 \times 10^{-28}$), suggesting that viral load of HHV-7 and HHV-6B decline with age (Fig. 2b). In saliva samples from SPARK, which represented infancy to old age, viral DNA loads within age tranches (measured by viral read prevalence or mean abundance) exhibited trajectories that varied markedly with age and across viruses (Fig. 2a and Extended Data Figs. 1d and 3a–d). Most viruses appear to increase rapidly in prevalence in the first several years of life (Fig. 2a), presumably reflecting primary infection, after which EBV DNA prevalence continued to increase with age ($P = 7.6 \times 10^{-140}$), whereas HHV-6B DNA prevalence decreased sharply and steadily from childhood onwards ($P = 6.4 \times 10^{-315}$; Fig. 2a), perhaps indicating increasing control by the adaptive immune system of the host over life. HHV-7 viral read prevalence in saliva increased steadily to more than 95% before modestly decreasing in middle age ($P = 6.1 \times 10^{-585}$; Fig. 2a), consistent with the pattern in UKB blood samples (Fig. 2b), whereas HHV-7 viral abundance in saliva began to decrease in childhood ($P = 6.6 \times 10^{-194}$; Extended Data Fig. 3c), suggesting diminishing intensity or frequency of lytic events.

Notably, viral DNA load was consistently higher in men than in women, across seven viruses and in both blood and saliva (Fig. 2a–d). This aligns with observations of stronger immune responses and lower viral load of HIV and hepatitis B in women relative to men[32]. These sex differences in viral DNA load appeared to emerge in adolescence (Fig. 2a). Viral DNA prevalence also displayed substantial circadian and seasonal variation (Fig. 2e,f and Extended Data Fig. 3e–l) that was robust to potential confounders (Supplementary Note 3), although confounding by unmeasured visit-related factors cannot be fully excluded. Viral read prevalence increased by about 1.2–1.3-fold from 09:00 to 20:00 (based on recorded times of blood sample acquisition) for both EBV and HHV-7, but with somewhat different trajectories over the course of the day (Kronos[33] harmonic regression $P = 1.2 \times 10^{-9}$ and $3.3 \times 10^{-23}$, respectively; Fig. 2e). EBV viral read prevalence was also about 1.3-fold higher in winter compared with summer months, whereas HHV-7 showed different, more modest seasonal variation (Kronos $P = 2.9 \times 10^{-101}$ and $9.8 \times 10^{-12}$, respectively; Fig. 2f). These patterns replicated in AoU saliva samples (for EBV; Extended Data Fig. 3k,l) and were not driven by cyclical variation in lymphocyte percentages in UKB blood samples, which exhibited weaker and different patterns (Extended Data Fig. 3m,n). The high prevalence of EBV reads in saliva with similar circadian patterns as blood, together with the bimodal abundance of EBV DNA in saliva (Fig. 1d), suggests lytic events may be quite frequent and play a larger role in determining systemic load. Viral DNA load also exhibited prominent, several-fold variation across genetic ancestries, with different viruses enriched in different ancestry groups (Extended Data Fig. 4, Supplementary Table 6 and Supplementary Note 4). These patterns were robust to controlling for available socioeconomic variables (Supplementary Note 4).

## Human genetics shapes viral DNA load

Almost all human traits are shaped by genetic variation in ways that can elucidate molecular mechanisms. To identify such human genetic effects on viral DNA load, we performed genome-wide association studies (GWAS) on viral DNA load phenotypes in UKB and AoU (Methods). For viral DNA load in blood, we then meta-analysed results across UKB and AoU. Common variation (minor allele frequency (MAF) > 0.1%) at 82 loci in the human genome associated (at $P < 5 \times 10^{-8}$) with viral DNA load of one or more viruses in blood (of 6 tested): 45 loci for EBV, 3 loci for HHV-6B, 37 loci for HHV-7, 6 loci for TTV (TUS01), 6 loci for TTV (HD14a) and 2 loci for TTV (VT416) (Fig. 3a–c, Extended Data Fig. 5 and Supplementary Table 7). Common variants also associated with viral DNA load in saliva samples from AoU for EBV (2 loci), HHV-6B (3 loci), HHV-7 (12 loci) and MCPyV (2 loci) (Extended Data Fig. 6). We verified that these associations were not driven by misalignment of human reads to viral genomes (Extended Data Fig. 7a,b and Supplementary Note 5), were concordant between UKB and AoU (Extended Data Fig. 7c–e), reflected

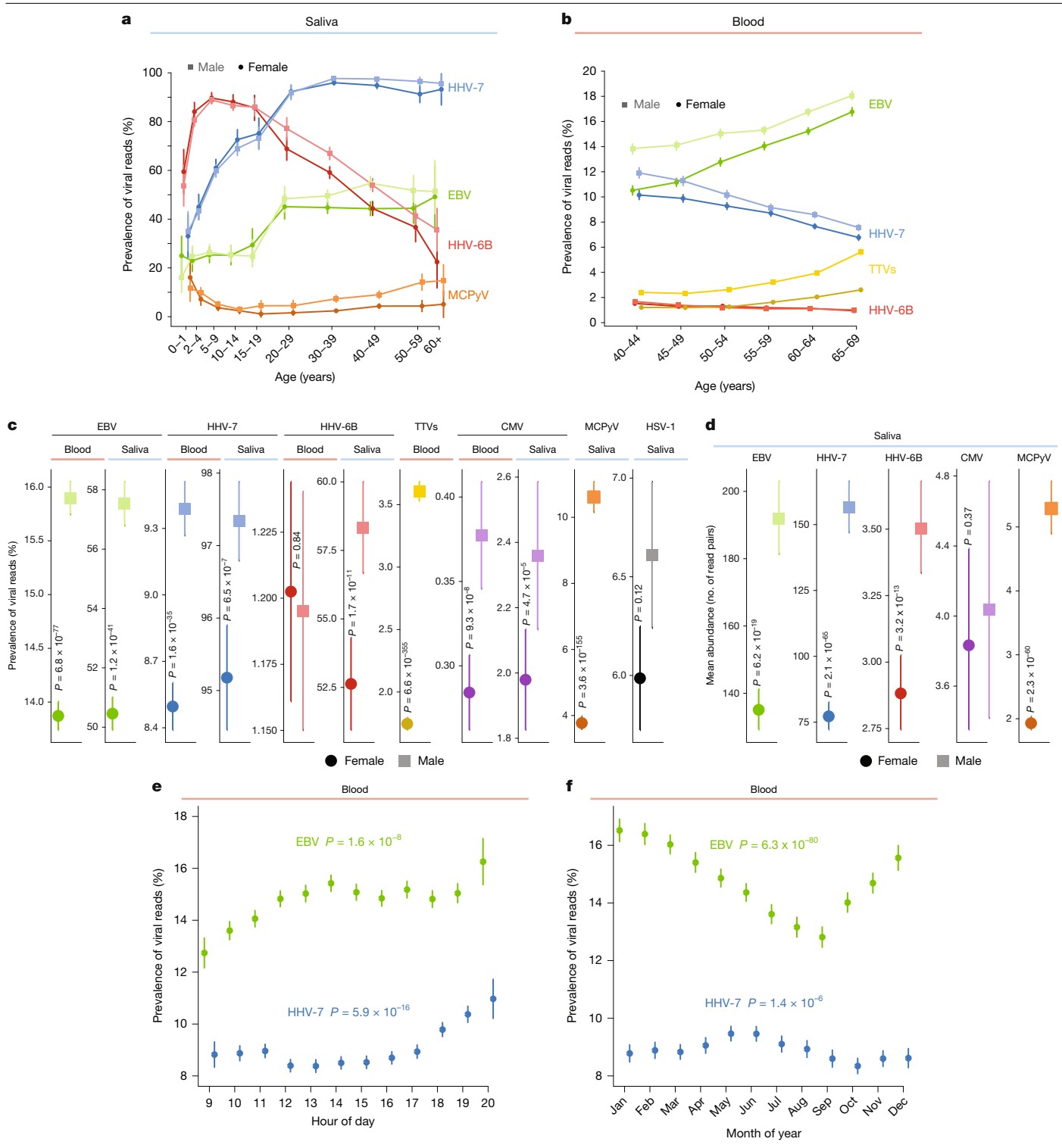

**Fig. 2 | Age, sex, time of day and time of year associations with viral DNA load. a**, Prevalence of viral DNA sequences by age and sex in saliva-derived WGS from SPARK ($n = 12,519$ total, 58–1,582 participants per age–sex bin). Error bars represent 95% confidence intervals. **b**, Prevalence of viral DNA sequences by age and sex in blood-derived WGS from UKB. Anellovirus (TTV) measurements group together reads aligned to the TUS01, HD14a and VT416 reference genomes ($n = 482,882$ total, 22,896–63,810 participants per age–sex bin). Error bars represent 95% confidence intervals. **c**, Prevalence of viral DNA sequences by sex in blood-derived WGS from UKB (EBV, HHV-7, HHV-6B, TTV: $n = 222,094$ men and 263,132 women) and AoU (CMV: $n = 143,918$ men and 220,649 women) and saliva-derived WGS from AoU (EBV, CMV, MCPyV, HSV-1: $n = 17,335$ men and 31,425 women) and SPARK (HHV-7, HHV-6B: $n = 3,378$ fathers and 3,380 mothers). Error bars represent 95% confidence intervals. **d**, Abundance of viral DNA

sequences by sex in saliva-derived WGS from AoU (EBV: $n = 9,973$ men and 15,856 women; CMV: $n = 409$ men and 622 women; MCPyV: $n = 1,840$ men and 1,185 women) and SPARK (HHV-7: $n = 3,288$ fathers and 3,217 mothers; HHV-6B: $n = 1,945$ fathers and 1,762 mothers). Centres indicate mean and error bars represent 95% confidence intervals. **e**, Prevalence of viral DNA sequences in UKB blood WGS by time of day of sample collection (rounded to the nearest hour, $n = 490,136$ total, 6,618–52,949 participants per hour). Error bars represent 95% confidence intervals. **f**, Prevalence of viral DNA sequences in UKB blood WGS data by month of sample collection ($n = 490,136$ total, 27,364–51,195 participants per month). Error bars represent 95% confidence intervals. $P$ values from two-sided linear regression (**c**,**d**) or one-sided ANOVA test between models with and without collection hour (**e**) or month (**f**) (see Methods).

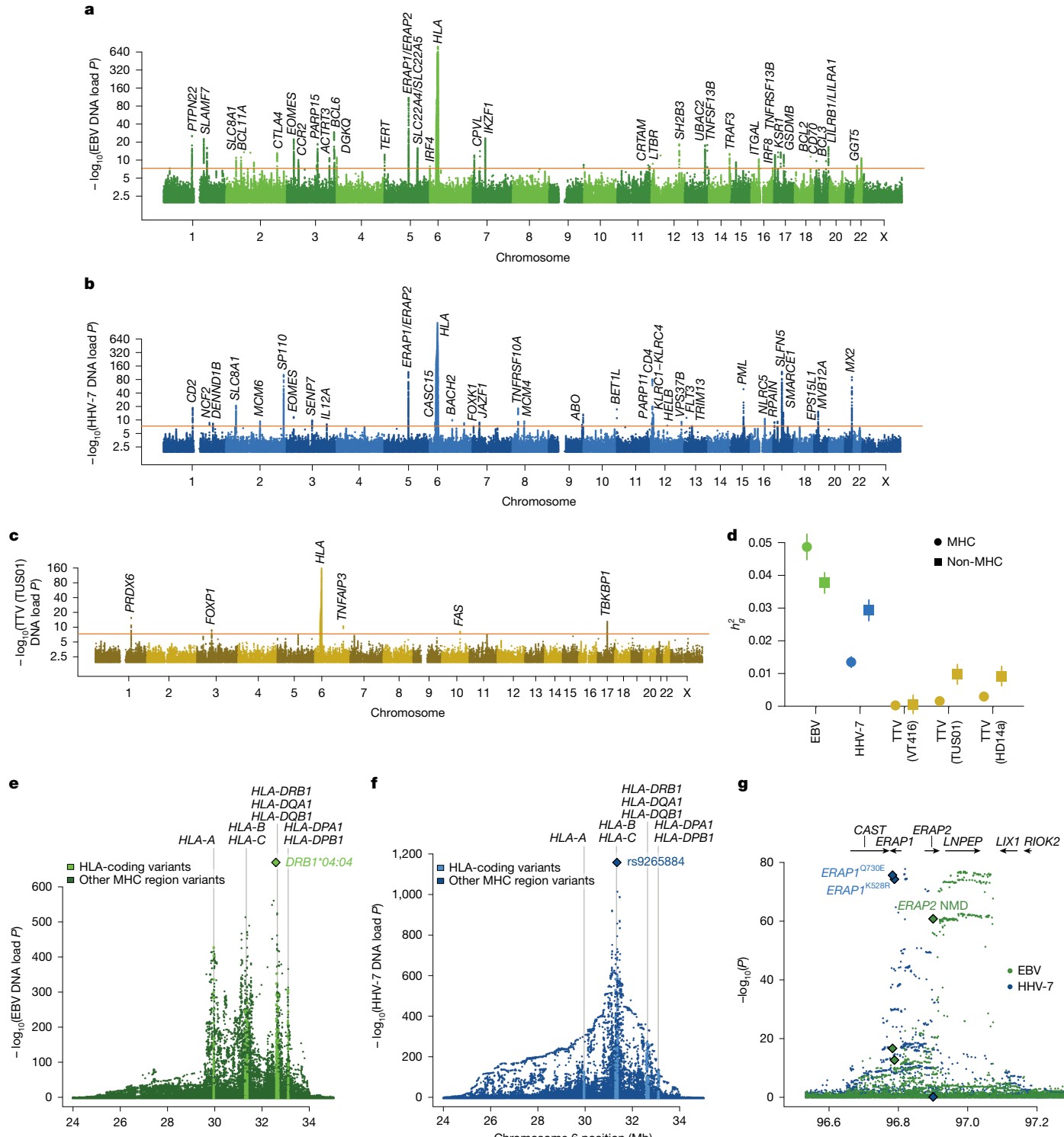

**Fig. 3 | Genome-wide association analyses of viral DNA load in blood.**
**a**, Genome-wide associations of human genetic variants with EBV DNA load in blood (inverse-normal transformed abundance), meta-analysed across UKB ($n = 453,770$) and AoU ($n = 201,168$) European-ancestry sub-cohorts. **b**, Genome-wide associations with HHV-7 DNA load in blood (inverse-normal transformed abundance), meta-analysed across UKB ($n = 453,770$) and AoU ($n = 200,091$) European-ancestry sub-cohorts. **c**, Genome-wide associations with anellovirus TUS01 DNA load in blood (inverse-normal transformed abundance), meta-analysed across UKB ($n = 453,770$) and AoU ($n = 200,091$) European-ancestry sub-cohorts. **d**, Estimates of heritability from common genetic variants for viral

DNA load phenotypes from variants within versus outside of the MHC region of the human genome, defined as 24–35 Mb on chromosome 6 ($n = 419,421$ unrelated UKB participants with European ancestry). Error bars represent 95% confidence intervals. **e**, Associations of variants in the MHC region of the human genome with EBV DNA load in blood in UKB participants with European ancestry. **f**, Analogous to **e**, for HHV-7 DNA load. **g**, Associations of common genetic variants at the *ERAP1–ERAP2–LNPEP* locus with EBV DNA load (green) and HHV-7 DNA load (blue) in blood. Arrows indicate direction of transcription for nearby genes. Associations of two missense variants in *ERAP1* and a cryptic splice variant in *ERAP2* are indicated with diamonds.

distinct effects in blood versus saliva (Extended Data Fig. 7f–i), and did not overlap any of the ten telomeric sites of eHHV-6 chromosomal integration that we identified in UKB data (Extended Data Fig. 7j,k and Supplementary Note 6).

For six of the above seven viruses (all except HHV-6B), the MHC region harboured the strongest associations between common human genetic variants and viral DNA load (Fig. 3a–c, Extended Data Figs. 5 and 6 and Supplementary Table 7). The MHC region as a whole contributed large proportions of viral DNA load heritability (56% for EBV, 32% for HHV-7 and 14–30% for the anelloviruses in UKB; Fig. 3d), as previously observed for HIV[17]. Association patterns in the MHC region—the specific associating single nucleotide polymorphisms (SNPs) and their relative strengths of association—were unique to each virus (Fig. 3e,f and Supplementary Table 8). For EBV viral DNA load in blood, the top genetic association was generated by the human leukocyte antigen (HLA) *DRB1*04:04* allele (Fig. 3e), which associated with almost twofold increased prevalence of EBV reads (23.0% (s.e.m. 0.2%) in heterozygous carriers versus 13.6% (s.e.m. 0.05%) in non-carriers). DRB1 is a co-receptor that EBV binds to enter human cells[34]; *DRB1*04:04* might encode an isoform with higher affinity for EBV protein gp42, facilitating infection. Several other independent associations with EBV DNA load were centred in or near other HLA genes (Fig. 3e). This pattern was also observed (but less pronounced) for HHV-7 (Fig. 3f). Some of these associations are likely to reflect the role of the HLA system in viral antigen presentation and immune response; specific HLA alleles that strongly enable presentation of peptides from a given virus may drive distinct effects of HLA alleles on load of different viruses. Genetic variation in the MHC region has also been observed to associate with measurements of several anti-EBV antibodies[22], but only a subset of these serological measurements (VCA-p18 and ZEBRA) exhibited association patterns similar to that of EBV DNA load (Extended Data Fig. 8 and Supplementary Note 7), suggesting that levels of some but not all antibodies are influenced by viral DNA load.

The strong associations of HLA alleles with viral DNA load enabled us to characterize how these genetic effects vary across individuals as a function of age and other genotypes (Extended Data Fig. 9a–h and Supplementary Note 8). Some associations of variants in the MHC region with EBV DNA load also exhibited specificity to blood or saliva; in particular, the *HLA-A*02:01* allele appeared to decrease EBV DNA load in blood ($P = 2.9 \times 10^{-431}$) but have no effect on EBV DNA load in saliva ($P = 0.099$; Extended Data Fig. 9i), consistently across cohorts (Extended Data Fig. 9j). One possible explanation is that *A*02:01* could present a peptide from a latent-phase EBV antigen, which could help control infection in blood but have little effect on recognition of the lytically infected plasma or epithelial cells that more directly influence viral DNA load in saliva.

Haplotypes at the *ERAP1–ERAP2–LNPEP* locus (on chromosome 5), which encodes the ERAP1, ERAP2 and LNPEP peptidases that process peptides for MHC display[35], associated strongly with DNA load of EBV ($P = 1.8 \times 10^{-77}$ in UKB) and HHV-7 ($P = 2.7 \times 10^{-78}$; Fig. 3g). For HHV-7, the association signal appeared to be driven by two missense variants in *ERAP1* (in high linkage disequilibrium with each other, $r^2 = 0.7$) that affect substrate preference (Q730E) and enzymatic activity (K528R)[36]. By contrast, EBV DNA load associated more strongly with variation near *ERAP2* and *LNPEP*, including a common cryptic splice variant (rs2248374) that generates a truncated *ERAP2* transcript that undergoes nonsense-mediated decay[37].

Beyond the very large effects from the *HLA* and *ERAP1–ERAP2–LNPEP* loci, each virus associated with additional human genomic loci (Fig. 3a–c, Extended Data Figs. 5 and 6 and Supplementary Table 7). These loci contained many genes with known or likely roles in viral infection, immune evasion and proliferation. Some of these loci, such as the B cell lymphoma (BCL) genes (associated with EBV DNA load), are likely to encode alleles that affect survival and proliferation of infected host cells[38]. By contrast, *PML*, *SP110* and *MX2* (associated with HHV-7 and/or

HHV-6B DNA load) encode nuclear proteins with roles in restricting herpesvirus replication[39,40]. Associations at genes that influence apoptosis (*FAS*, *TNFAIP3* and *TBKBP1*) suggest that virally induced apoptosis signalling might be particularly important for anellovirus survival[41]. At the *ABO* locus, which determines the presence of blood group antigens bound by some viruses[42], alleles that produce type A and type B antigens both associated with higher HHV-7 DNA load (Extended Data Fig. 9k,l). Associations at the *LILR* (leukocyte immunoglobulin-like receptor) and *CPVL* (carboxypeptidase vitellogenic like) genes, like the strong associations at *HLA* and *ERAP1/ERAP2*, indicated the importance of genetic variation that modulates viral peptide presentation and recognition[43,44].

Gene-level burden tests of rare loss-of-function and missense variants in UKB additionally identified four genes in which rare protein-coding variants associated with increased viral DNA load: *TERT* and *ASAH2B* for EBV, and *MX2* and *ZNF584* for HHV-7 ($P < 2.8 \times 10^{-6}$; Supplementary Table 9). These associations appeared to be virus-specific, except for *ZNF584*, for which rare coding variants also associated with increased EBV DNA load ($P = 3.1 \times 10^{-3}$). At two loci (*CPVL* and *TNFRSF13B*) at which a missense variant generated the top common-variant association with EBV DNA load, the burden test demonstrated concordant associations of loss-of-function variants in these genes (Extended Data Fig. 9m,n).

The strong ancestry differences in viral DNA load (Extended Data Fig. 4 and Supplementary Note 4) led us to query whether GWAS of EBV and HHV-7 DNA load in AoU participants with African ancestry ($n = 77,573$) might identify specific loci contributing to these differences. These analyses identified a novel association at *ACKR1*: the rs2814778-CC genotype that generates the malaria-protective Duffy-null phenotype[45] appeared to explain 28% [21%–35%] and 45% [38%–52%] of the African-versus-European ancestry differences in EBV and HHV-7 DNA prevalence, respectively, and these associations are only partially explained by neutropaenia in Duffy-null individuals[46] (Extended Data Fig. 10 and Supplementary Note 9).

## *HLA-B*08:01* association is EBV type-specific

EBV is known to exist in two main strains, type 1 and type 2, which differ primarily in the sequences of key latency-associated EBV genes[47] and may interact differently with the host immune system. This raises the possibility that some human genetic variants involved in host–virus genetic interactions—for example, HLA alleles with affinities for specific viral peptides—might associate specifically with viral DNA load of one EBV type but not the other, as has been observed for hepatitis C[48]. To search for such effects, we separately quantified type 1 and type 2 EBV by realigning WGS reads to EBV type 1 and type 2 reference genomes and identifying reads that aligned within two type-informative genomic regions (*EBNA2* and *EBNA3A–EBNA3C*)[49] (Fig. 4a, Extended Data Fig. 11a,b and Methods). The prevalence of reads from EBV type 2 relative to type 1 varied significantly by an individual's place of birth (Fig. 4b,c and Extended Data Fig. 11c), as expected[50]. EBV type 2 was relatively more prevalent in individuals born in Africa[50] compared with individuals born in Europe ($P = 5.4 \times 10^{-7}$) or North America ($P = 0.042$; Fig. 4b), and its relative prevalence also varied regionally among individuals born in the UK (Fig. 4c), independently of overall EBV prevalence (Extended Data Fig. 11d). Notably, quantitative antibody responses to EBV antigens were slightly higher in individuals with type 2 EBV than in those with type 1 EBV (Extended Data Fig. 11e–h), suggesting that type 2 EBV might generate a stronger autoimmune response for autoantigen epitope mimics.

Genetic association analyses leveraging this additional information about EBV type suggested that HLA variation confers type-specific protection against EBV infection (Fig. 4d–g). In analyses of EBV type 1 and type 2 DNA load separately, association patterns in the MHC region (Fig. 4d,e) were broadly similar to those that we had observed for total EBV DNA load (Fig. 3e), presumably reflecting the contributions of many human genetic effects that were difficult to resolve. However, a case-case association analysis—in which genetic data from individuals

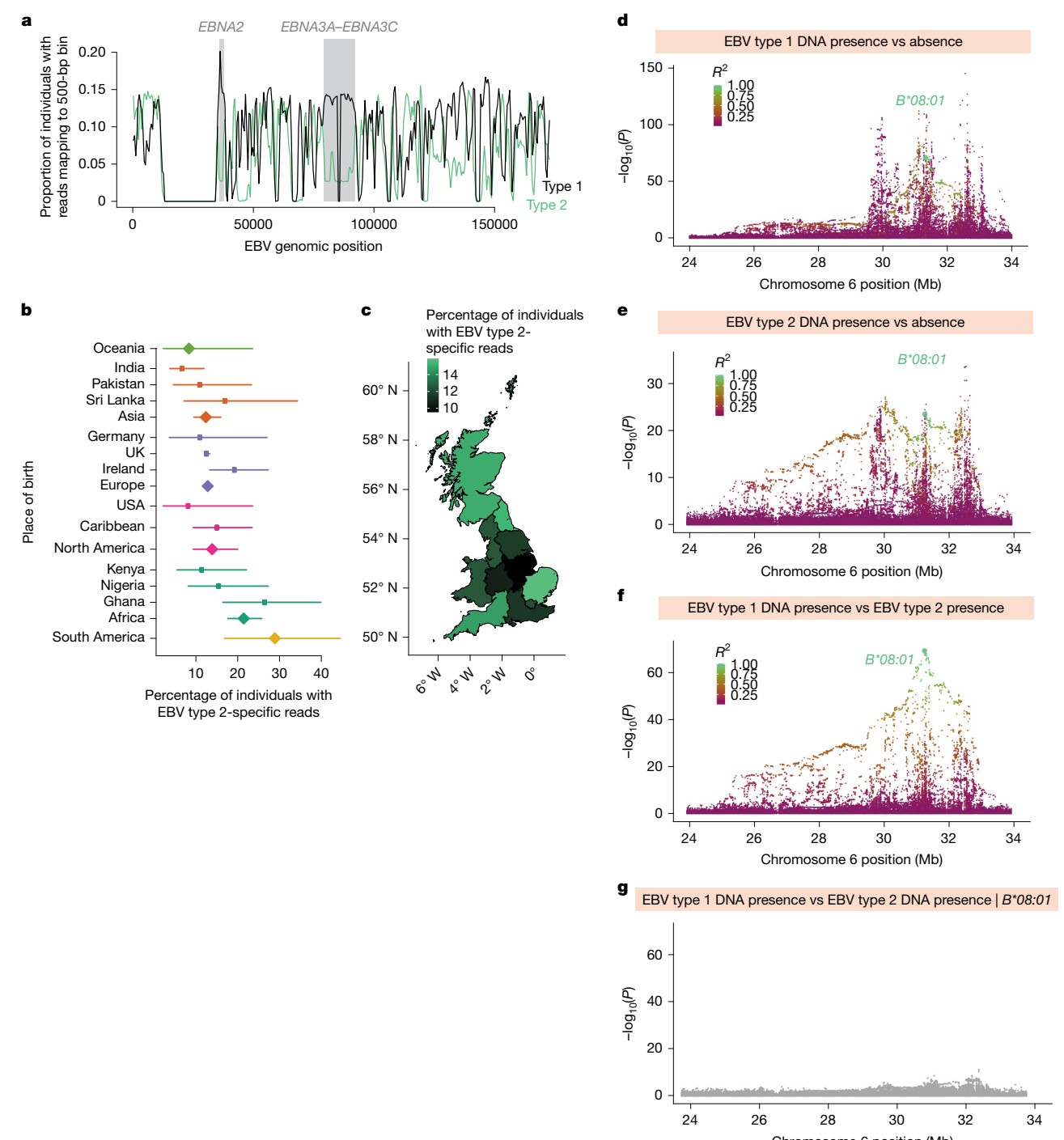

**Fig. 4 | Analyses of EBV type 1 and type 2 identify a type-specific association with *HLA-B\*08:01*. a**, Frequencies of observing WGS reads mapping to each 500 bp segment of the EBV type 1 (black line) and type 2 (green line) reference genomes in saliva WGS samples from SPARK (*n* = 12,519). Highlighted in grey are two regions used to distinguish EBV types, both containing genes with known type-specific sequences, *EBNA2* and *EBNA3A–EBNA3C*. Read alignments in these two regions show consistent ratios of type 1-aligned to type 2-aligned reads, whereas other EBV genomic regions (that are not type-informative) do not. **b**, Proportion of UKB participants positive for EBV type 2 (among EBV DNA-positive individuals) by location of birth. In each group, the proportion was estimated as the number of UKB blood WGS samples containing two or more EBV type 2-specific reads divided by the number containing two or more type 1-specific or two or more type 2-specific reads (total *n* = 11,511 participants). Estimates are shown for countries of birth with more than 35 samples containing a type-specific read, and for continental groupings (aggregated across all countries; diamonds). Error bars represent 95% confidence intervals.

**c**, Proportion of UKB participants positive for EBV type 2 by region of birth within the UK (*n* = 9,769). Map source: Office for National Statistics, licensed under the Open Government Licence v.3.0 and Contains OS data. Crown copyright and database rights 2017–2026. **d**, Associations of variants in the MHC region of the human genome with EBV type 1 positivity (presence of two or more WGS reads that mapped to type-informative regions of the EBV type 1 genome) in UKB blood WGS (*n* = 418,403 unrelated participants with European ancestry). Variants are coloured by linkage disequilibrium relative to *HLA-B\*08:01* (green-to-purple shading), indicated by the large green dot. **e**, Associations of variants with EBV type 2 positivity. **f**, Associations of variants with EBV type 2 positivity, using individuals with EBV type 1 positivity as controls (*n* = 9,233 unrelated participants with European ancestry). **g**, Conditional associations of variants with EBV type 2 positivity, using individuals with EBV type 1 positivity as controls, after including *HLA-B\*08:01* genotype as a covariate. *P* values by two-sided linear regression (**d**–**g**).

with higher type 2 DNA load were compared against genetic data from those with higher type 1 DNA load—generated a readily interpretable association pattern led by the MHC class I *B*08:01* allele ($P = 7.4 \times 10^{-70}$; Fig. 4f). *B*08:01* appeared to be protective against type 1 EBV (odds ratio (OR) = 0.62 [0.59–0.66] for observing a type 1 read; $P = 7.1 \times 10^{-72}$) but associated with increased risk for type 2 EBV (OR = 1.63 [1.48–1.80]; $P = 4.1 \times 10^{-24}$), suggesting that it may have high affinity for one or more peptides derived from the type 1 EBNA-2 or EBNA-3 proteins, but not homologous type 2 proteins. A possible candidate is an antigen from EBNA-3A that enables type 1-specific cytotoxic T-lymphocyte activity and is B*08:01-restricted[51,52]. Conditional association analyses that included *B*08:01* as a covariate suggested that this haplotype accounts for most of the type-specific association signal in the MHC region, with some remaining associations of variation in the MHC class I and II regions (Fig. 4g).

## EBV DNA load is a risk factor for Hodgkin's lymphoma

We next sought to identify associations between viral DNA load and the abundant biological and clinical phenotypes available for the 490,401 UKB research participants. We identified 181 significant associations between viral DNA load and disease phenotypes, and 366 with blood count, biomarker and metabolite measurements (Bonferroni-corrected $P < 0.05$; Fig. 5a and Supplementary Tables 10–13). Most of the disease associations involved higher viral DNA loads in affected individuals (Extended Data Fig. 12a), and many associated diseases involved inflammation or weakened immune function, such as AIDS, anaemia, diabetes and renal failure (Fig. 5a). Organ transplantation associated strongly with increased anellovirus DNA load, presumably reflecting use of post-transplant immunosuppression medication[14]. Concordantly, individuals taking common immunosuppressive drugs had elevated EBV and anellovirus DNA loads (Supplementary Table 14); this could explain some of the observed disease associations. Carrier status for endogenous HHV-6B or HHV-6A did not significantly associate with any biological or clinical phenotypes (Supplementary Table 10).

Many of the strongest associations of EBV and HHV-7 DNA load with disease phenotypes were directly or indirectly related to smoking (for example, nicotine dependence, respiratory disorders and lung cancer; Fig. 5a and Supplementary Tables 10 and 11). Cumulative smoking exposure (pack-years) strongly associated with increased prevalence of EBV DNA, which was nearly twice as high for the heaviest smokers compared with nonsmokers ($P = 3.3 \times 10^{-195}$; Fig. 5b). By contrast, smoking exposure associated with decreased prevalence of HHV-7 DNA ($P = 9.6 \times 10^{-22}$; Fig. 5b). Cigarettes smoked per day associated with similar trends ($P = 3.3 \times 10^{-563}$ and $2.7 \times 10^{-9}$ for increased EBV prevalence and decreased HHV-7 prevalence, respectively; Fig. 5c), suggesting that smoking has strong, opposite effects on viral DNA load for EBV and HHV-7. These associations were replicated in AoU blood samples (Extended Data Fig. 12b) and in AoU saliva samples; smoking associated with an increase not only in EBV prevalence ($P = 3.6 \times 10^{-18}$) but also in EBV abundance among EBV DNA-positive individuals ($P = 4.2 \times 10^{-8}$, Extended Data Fig. 12c,d).

The large number of genetic effects on EBV DNA load identified by our GWAS provided an opportunity to investigate the causal direction of phenotypic associations—that is, whether viral DNA load affects (or is affected by) the clinical phenotypes with which it associates. We did this by using these DNA load-influencing SNPs as instrument variables for Mendelian randomization, testing whether or not the genetic variants that associate with viral DNA load also associate with the traits with which viral DNA load associates in populations[53]. Whereas most viral DNA load–phenotype associations are likely to be a consequence of pathology, treatment, or environmental exposures (such as smoking) leading to increased viral DNA presence in blood, the finding that the alleles that increase risk for viral DNA load also consistently confer risk for disease offers evidence that viral DNA load is causally upstream.

To use this approach to assess the potential effect of EBV DNA load on autoimmune diseases and blood cancers that are epidemiologically associated with EBV infection[2], we analysed GWAS summary statistics from a meta-analysis of FinnGen[54], UKB[55] and the Million Veteran Program[56] using four Mendelian randomization methods, excluding the MHC region to guard against pleiotropy from linkage disequilibrium[53] (Supplementary Table 15). For multiple sclerosis, for which prior EBV infection is a large risk factor[7], these analyses indicated that latent viral DNA load after primary infection is unlikely to further influence the risk of developing multiple sclerosis (MR–Egger $P = 0.52$; Fig. 5d and Extended Data Fig. 12e), despite UKB participants with multiple sclerosis having higher EBV DNA load ($P = 0.005$). For systemic lupus erythematosus and rheumatoid arthritis, which also associated with higher EBV DNA load in UKB ($P = 4.6 \times 10^{-7}$ and $2.2 \times 10^{-23}$, respectively), Mendelian randomization analyses were inconclusive about causality (Extended Data Fig. 12f,g).

By contrast, all analyses (across the 4 Mendelian randomization methods) suggested that viral DNA load after EBV infection significantly influences the development of Hodgkin's lymphoma ($P = 1.2 \times 10^{-3}$ to 0.014; Fig. 5e,f and Supplementary Table 15). Effect estimates for non-Hodgkin's lymphoma, chronic lymphocytic leukaemia and lymphoid leukaemia were smaller and not consistently significant across Mendelian randomization approaches, suggesting specificity for Hodgkin's lymphoma (Extended Data Fig. 12h–j). Moreover, EBV DNA abundance in people without Hodgkin's lymphoma at the time of blood draw for WGS was quantitatively predictive of their likelihood to go on to develop Hodgkin's lymphoma (OR = 2.01 [1.40, 2.84] for incident Hodgkin's lymphoma among EBV DNA-positive versus DNA-negative individuals; Fig. 5g). Increased EBV viral DNA load in blood—which could reflect a higher proportion of latently infected B cells—may increase the opportunities for oncogenic transformation leading to Hodgkin's lymphoma.

## Discussion

Here, by profiling common DNA viruses in blood and saliva in population sequencing data, we have characterized the major factors that are likely to shape these constituents of the human virome. Human genetics, age, sex, time of day, month of the year and smoking behaviour consistently associate strongly with viral DNA load across many viruses, but the relative magnitudes and even directions of these effects are often virus-specific. These results on common, latent viruses, made possible by cross-sectional analyses across over 900,000 biobank participants, complement previous insights from targeted studies of highly pathogenic viruses. These results also corroborate parallel studies of EBV in UKB and AoU[57–59]; here, we extended these analyses to 30 other viruses, and for EBV, we also observed interactions of genetic effects with biosample type and with EBV genetic variation.

The largely distinct, polygenic human genetic effects that we observed for DNA load of several common viruses (EBV, HHV-6B, HHV-7, MCPyV and anelloviruses) expand on earlier work identifying key loci influencing control of HIV[16,17] and hepatitis C[18,48] load. The MHC region contributed the strongest genetic effects for each virus that we studied except HHV-6B, similar to other GWAS of infectious disease phenotypes[60], but the specific HLA alleles that influenced viral DNA load varied across viruses (Supplementary Table 8). These virus-specific effects at the MHC locus probably contribute to the prominent variation of viral DNA load that we observed across genetic ancestries, which is likely to reflect the balancing selection that has shaped human genetic variation affecting the immune system; alleles that confer an advantage against a particular virus may increase susceptibility to other pathogens[61]. Beyond the MHC, polygenic influences on DNA load of different viruses were likewise largely distinct, sometimes surprisingly so: for example, variation at *MX2* and *PML*, which encode host factors thought to be broadly restrictive against herpesviruses[39,40],

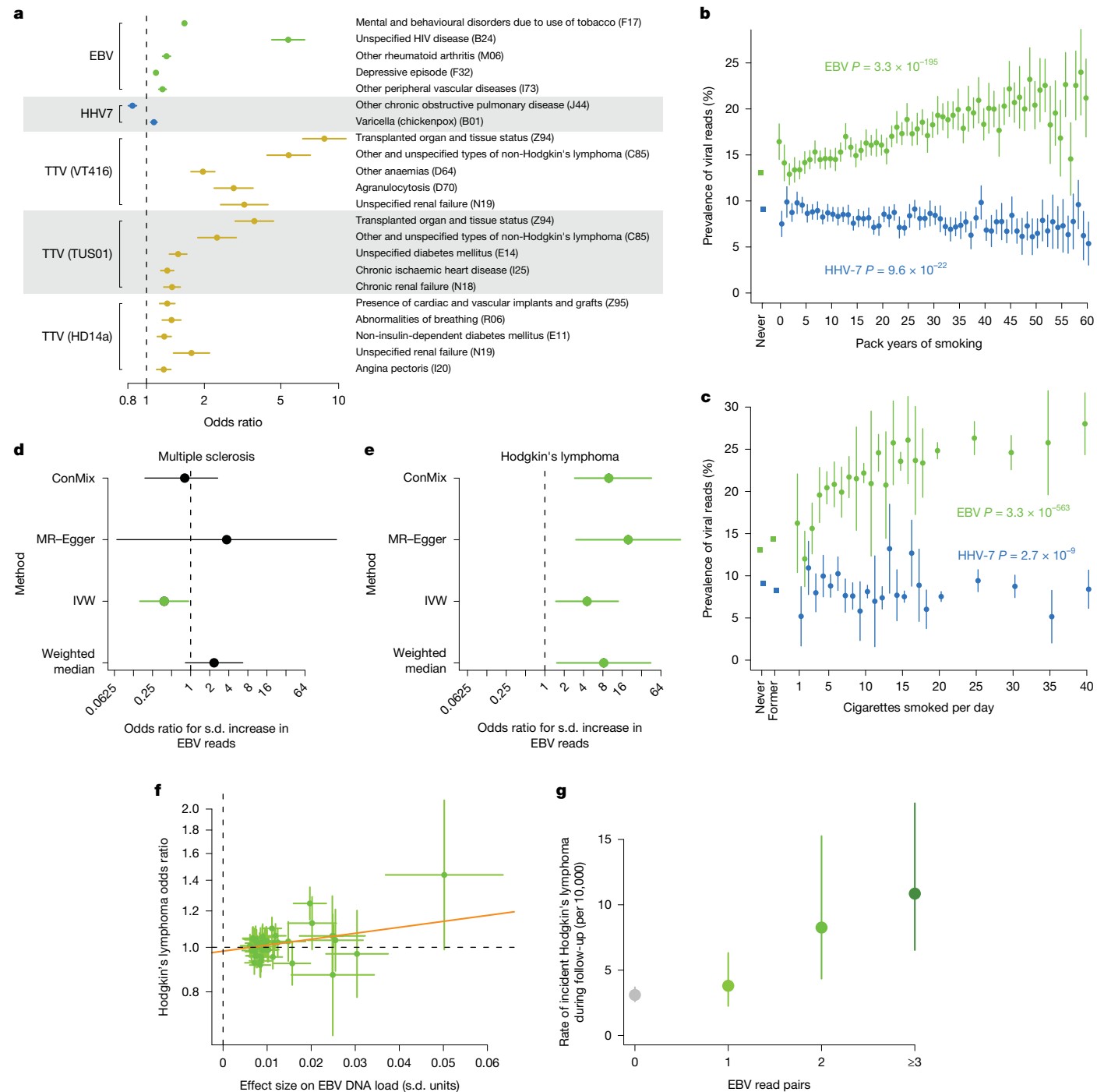

**Fig. 5 | Associations of viral DNA load with clinical phenotypes and smoking. a**, Associations of clinical conditions with detectable virus (presence of viral DNA sequences in blood WGS) in UKB participants with European ancestry (*n* = 453,770). Several of the strongest independent associations are shown; the full set of associations is provided in Supplementary Tables 10 and 11. **b**, Prevalence of detectable EBV positivity (green) and HHV-7 positivity (blue) by smoking pack-years in UKB (*n* = 379,233 participants with European ancestry). Never smokers are also shown. **c**, Prevalence of detectable EBV positivity (green) and HHV-7 positivity (blue) by cigarettes smoked per day in UKB (*n* = 438,101 participants with European ancestry). Never smokers and former smokers are also shown. **d**, Estimates of the causal effect of EBV DNA load (exposure; *n* = 638,825) on risk for multiple sclerosis (outcome; meta-analysis of 7,907 cases and 1,474,810 controls) using different Mendelian randomization approaches (*y* axis) with 44 non-MHC loci as instrument variables. An estimate is plotted in green if its 95% confidence interval does not overlap an odds ratio

of 1 (no effect). ConMix, contamination mixture; IVW, inverse-variance weighted; MR, Mendelian randomization. **e**, Estimates of the causal effect of EBV DNA load on risk for Hodgkin's lymphoma (meta-analysis of 2,529 cases and 1,159,394 controls). Mendelian randomization odds ratios are expected to be overestimated due to measurement noise (Methods). **f**, Effect sizes for 44 common genetic variants (at distinct non-MHC loci)–used as instrument variables in Mendelian randomization–for EBV DNA load (*x* axis) and risk for Hodgkin's lymphoma (*y* axis). The line shown is from the MR–Egger test; the *y* intercept is not significantly different from OR = 1, indicating low pleiotropy of the genetic instruments. **g**, Incidence rates of Hodgkin's lymphoma (during approximately 15 years of follow-up; *n* = 174 incident cases), stratified by the number of EBV DNA fragments in UKB blood WGS data, for individuals without a Hodgkin's lymphoma diagnosis at the time of sample collection. Error bars represent 95% confidence intervals. *P* values by two-sided linear regression (**b**,**c**).

associated with DNA load of HHV-7 but not EBV. We also highlight the utility of using pairs of related phenotypes, such as type 1 and type 2 EBV DNA load (Fig. 4) or EBV DNA load in blood and saliva (Extended Data Fig. 9i,j), as a mechanism to accomplish fine-mapping of a complex genetic association.

The large number of polygenic effects that we observed for EBV DNA load—45 human genomic loci—enabled Mendelian randomization to evaluate the causality of associations with clinical phenotypes (Fig. 5). These analyses suggested that multiple sclerosis risk from EBV infection may be mediated largely by adaptive immune response to an EBV infection[62], without an additional influence of subsequent lifetime viral DNA load. By contrast, EBV DNA load appears to be a causal risk factor for Hodgkin's lymphoma. This result contrasts with a Mendelian randomization analysis from a parallel study that did not observe an effect of EBV DNA load on Hodgkin's lymphoma[58]; this discrepancy might be explained by increased statistical power here (from Hodgkin's lymphoma GWAS summary statistics from a 2.6-fold larger meta-analysis, as well as more EBV DNA load GWAS hits available as genetic instruments) and inclusion versus exclusion of the MHC region. This result could potentially be validated in vitro by exposing germinal centre B cells to increasing titres of EBV and measuring the number of Hodgkin and Reed-Sternberg (HRS) cells produced[63]; alternatively, it could be studied in humanized mice[64] if HRS cell generation requires a tissue niche. An effect on Hodgkin's lymphoma is not unexpected given the risk of EBV-positive Hodgkin's lymphoma conferred by infectious mononucleosis[6], but it suggests that interindividual differences in lifetime management of DNA load affect health risks beyond infection itself. If so, interventions that reduce EBV DNA load could potentially decrease the risk of Hodgkin's lymphoma. Notably, smoking has previously been found to be a risk factor for EBV-positive Hodgkin's lymphoma (OR = 1.81 [1.27–2.56] for current smokers), but not for EBV-negative Hodgkin's lymphoma (OR = 1.02 [0.72–1.44])[65] and more weakly for non-Hodgkin's lymphoma (OR = 1.10 [1.00–1.20])[66], suggesting that the effect of smoking on EBV DNA load might be one mechanism that underlies Hodgkin's lymphoma subtype-specific risk. Similarly, antivirals such as acyclic nucleoside analogues that inhibit herpesvirus DNA polymerases and reduce EBV viral load in blood[67] may be worth evaluation for potential prophylactic benefit against Hodgkin's lymphoma.

Our analyses of human viruses that are commonly observable from population DNA sequencing data had several limitations that will need further efforts to resolve using analytical approaches that more completely capture viral genetic variation, additional cohorts of greater scale, and samples obtained from a wider range of body sites (Supplementary Note 10). We envision that biobank WGS data will continue to be a powerful resource for virology and epidemiology, particularly in geographic regions with a greater burden of certain DNA viruses such as hepatitis B and retroviruses such as HIV and human T lymphotropic virus.

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

## Methods

### Ethics

This research complies with all relevant ethical regulations. The study protocol (NHSR-8429) was determined to be not human subject research by the Broad Institute Office of Research Subject Protection as all data analysed were previously collected and de-identified. Use of SPARK data for this research was approved by SFARI (project 3350.2).

### UK Biobank, All of Us, and SPARK WGS data

All WGS data analysed in this work were generated in previous studies. For all cohorts, PCR-free methods were used in library preparation and sequencing was done on Illumina NovaSeq 6000 machines. WGS from UKB[23] was performed on libraries prepared with NEBNext Ultra II PCR-free kit (New England Biolabs) using blood-derived DNA from 490,401 individuals at the deCODE facility in Reykjavik, Iceland and the Wellcome Sanger Institute (Sanger), Cambridge, UK. Sequencing reads were aligned to human reference build GRCh38 graph genome with Illumina DRAGEN Bio-IT Platform Germline Pipeline v.3.7.8. Samples were sequenced to an average coverage of 32.5×.

WGS from the NIH All of Us v8 cohort[24] was performed on blood and saliva-derived DNA from 414,817 individuals (365,918 blood-derived and 48,899 saliva-derived samples). Sequencing reads from libraries prepared with PCR-Free Kapa HyperPrep library construction kit were aligned to human reference build GRCh38 with Illumina DRAGEN Bio-IT Platform Germline Pipeline v.3.4.12. Samples were sequenced to an average coverage of 37.9×.

WGS from the SPARK cohort of the Simons Foundation Autism Research Initiative (SFARI)[25] was performed on saliva-derived DNA from 12,519 individuals. Sequencing reads from libraries prepared with Illumina DNA PCR-Free Library Prep kit were aligned to human reference build GRCh38 with BWA-MEM by the New York Genome Center (NYGC) using Centers for Common Disease Genomics project standards. Samples were sequenced to an average coverage of 42×. Details of saliva sample collection, DNA extraction, and sequencing were described in ref. 68 (which analysed data from sequencing waves WGS1–3 of the SPARK integrated WGS (iWGS) v.1.1 dataset; here we analysed WGS1–5, which included additional samples included in subsequent sequencing waves).

Genotypes of genome-wide human genetic variants (SNPs and insertion–deletions) were previously generated for all three datasets. For UKB, we analysed genotypes previously imputed into UKB SNP-array data[55] using the TOPMed reference panel[69] (as the final UKB WGS genotype call set was not available at the time of analysis). For AoU and SPARK, we analysed genotypes previously called from WGS using DRAGEN (AoU[24]) and DeepVariant (SPARK[25]).

### Selection of viral reference genomes

We selected a panel of 31 viruses for which to profile viral DNA load (Supplementary Table 1) based on previous work that identified the most prevalent viruses observed in blood-derived DNA sequencing data from 8,240 individuals[9]. Additional viral reference genomes were included to more comprehensively represent viral families previously observed. For example, adenovirus was represented by Human adenovirus type 7 and human adenovirus E (type 4) reference genomes, as these are two of the most commonly observed in adults[70], and polyomavirus was represented by a set of common types (1/BK, 2/JC, 3/KI, 4/WU, 5/MC, 6, 7)[71,72]. Undetected herpesviruses (HSV-2 and VZV) were added given the high general prevalence of EBV and HHV-7. To maximize representation of the diversity of anellovirus genomes while limiting the size of the viral reference panel, we selected a representative from each of the five recently described anellovirus clades with available NCBI reference genomes[73] (Extended Data Fig. 1a).

The rationale for prioritizing a smaller set of viruses to profile (rather than attempting to comprehensively characterize viral diversity) was that our main goal was to identify effects of human genetics, age, sex and environmental exposures (such as smoking) on viral DNA load, and we only had statistical power to detect such effects for commonly observed viruses. Working with a smaller set of common viruses allowed us to perform careful QC on WGS-based quantifications of viral DNA load, which was important given the potential for read alignment artefacts.

### Measurement of viral DNA presence and abundance in WGS samples

In the UKB (blood), AoU (blood and saliva) and SPARK (saliva) WGS datasets, we first extracted unmapped reads (that is, reads that did not align to the GRCh38 human reference genome) and reads that aligned to the chrEBV decoy contig. We realigned these reads to the reference panel of 31 viral genomes (merged into a single reference for alignment) using BWA-MEM[74] (v.0.7.18) with 4 threads (-t 4). We took this read-mapping-based approach following Moustafa et al.[9], who observed minimal additional detection of viral sequences from blood-derived WGS upon using de novo assembly followed by protein-based search.

After realignment, each virus' genome was then scanned to identify regions with excessive numbers of alignments suggestive of accumulated misalignments originating from some other source of DNA (for example, mismapped human DNA) by first computing alignment coverage aggregated across all samples in each dataset (UKB, AoU blood, AoU saliva and SPARK, each analysed separately). To do so, for each sample, we used mosdepth[75] (v.0.3.9) to compute depth-of-coverage in each 500-bp window of each viral reference genome (--by 500); we skipped per-base depth output (-n) and mate overlap/CIGAR corrections (--fast-mode), restricted to reads passing default filters on SAM flags (-F 1796, which excludes duplicate reads), and filtered to reads with mapping quality ≥5 (-Q 5) for which their mates mapped to the same reference genome with an insert size in the range 100–1,000 bp (-l 100 -u 1000). The results of these filters were not sensitive to the choice of the mapping quality threshold; using a more stringent threshold (-Q 20) affected only a few percent of reads attributed to common viruses and had a negligible effect on downstream genetic association analyses.

Upon computing the 500 bp-resolution coverage profile of each viral genome of each sample, we computed the coverage profile of each virus (that is, for each 500-bp bin, what fraction of samples had non-zero coverage) within each cohort (UKB, AoU blood, AoU saliva and SPARK). For each virus, for each cohort, we flagged a subset of 500-bp regions for exclusion based on having coverage exceeding the following threshold:

$$4 \times (Q_3 - Q_1) + Q_3 + 5$$

where $Q_1$ and $Q_3$ are the first and third quartiles, respectively, of the distribution of alignment coverage across all 500 bp regions of that virus' genome in that cohort. This expression corresponds to a lenient 'Tukey fence', which is an outlier removal boundary defined based on adding a multiple of the interquartile range ($Q_3 - Q_1$) to the third quartile ($Q_3$). We used a lenient Tukey fence to retain regions with modest elevation of coverage (as such regions may still have a majority of alignments derived from the viral genome), and we added a constant offset of 5 to handle situations in which the first and third quartiles have the same value. Applying this bin-level filtering strategy per virus per cohort helped handle cohort-specific error modes of false positive viral alignments that might arise from heterogeneity in WGS data generation and processing (for example, details of how reads had previously been aligned to the human reference genome, which impacted which reads did and did not map to human chromosomes).

For association analyses of viral DNA load with biological and clinical phenotypes, these measures of viral DNA load were then converted into binary 'viral DNA positivity' indicators of presence or absence of reads from a given virus in each individual. For genetic association analyses

of viral DNA load in blood, the number of viral read pairs mapping to each genome (obtained by summing mosdepth 500-bp bin depth values across non-excluded bins and multiplying by 500/300 = (bin size)/(bases per read pair)) was inverse-normal transformed to capture quantitative information about viral abundance while limiting the influence of outlier samples. In saliva samples, in which viral reads were often much more abundant, two phenotypes were generated for genetic association analyses: viral DNA positivity (binary presence or absence of reads), and a quantitative abundance metric in which we applied inverse-normal transform to non-zero values (that is, masking individuals with no reads from a given virus) after normalizing read pair counts for library size. This allowed for the possibility of observing effects on viral prevalence but not abundance and vice versa. For associations with sex (Fig. 2c,d), the cohort used for analysis (UKB or AoU for blood; AoU or SPARK for saliva) was chosen to maximize prevalence and/or representation of an age range with a larger sex difference; for SPARK, which used a family design, analyses of sex effects were restricted to parents.

### Estimation of the number of EBV-derived reads expected to be present in blood WGS

To assess the reasonableness of the distribution of viral read counts observed in blood WGS (typically 0 or 1 read pair per sample, even for near-ubiquitous herpesviruses; Fig. 1b), we roughly estimated the number of EBV-derived reads expected per WGS sample as follows. On average, EBV is present in 1 out of 100,000 B cells[4] with roughly 100 episomes per infected cell[67]. Assuming B cells comprise roughly 5% of all white blood cells[76] yields an expected $5 \times 10^{-5}$ EBV genomes per white blood cell, or $2.5 \times 10^{-5}$ EBV genomes per haploid human genome in blood-derived WGS. The EBV reference genome is 171,823 bp, so WGS at 30x coverage of the human genome by 2x150bp read pairs should produce an expected 0.4 read pairs from the EBV genome per sample.

### Association of viral DNA prevalence with sample collection time

Collection time for blood samples in UKB was obtained from field 3166. For analyses of collection time for saliva-derived WGS in the AoU cohort, we excluded a large fraction of samples (62%) that we determined were likely to have been mailed; for such samples, recorded collection times corresponded to receipt of samples rather than time of saliva sampling. Specifically, samples were excluded if they lacked an in-person physical measurement for heart rate or had a heart rate measurement time separated by more than a day from WGS sample collection time. The recorded collection times for the 18,751 remaining samples were converted from UTC to local time by taking the modal time zone of the state containing the three-digit zip code for the corresponding individual.

*P* values for hour-of-day and month-of-year associations (Fig. 2e,f and Extended Data Fig. 3k–n) were calculated with ANOVA by comparing models with and without hour of the day or month of the year. Both hour of the day and month of the year were encoded as a series of indicator variables to allow for non-linear relationships with viral prevalence. All models included age, age squared, sex, assessment centre, and top genetic principal components as covariates (20 principal components for UKB; 16 principal components for AoU). Associations were confirmed with Kronos (v.1.0.0), which was run with default settings; to handle covariates, viral phenotypes were first adjusted for covariate effects estimated using harmonic regression.

### Association of viral DNA prevalence with genetic ancestry

For UKB, genetically inferred ancestry was determined as described previously[77]. In brief, 20 genome-wide ancestry principal components were used to identify groups of individuals within a Euclidean distance radius from the centre of individuals within each self-reported ethnicity category, with the distance threshold chosen to include a large fraction of individuals in that self-reported ethnicity category.

For AoU participants, previously generated genetically inferred ancestry[24] and ancestry admixture estimates from Rye[78] were used.

For SPARK participants, genetic ancestry was inferred using Euclidean distance on 10 genome-wide ancestry principal components to cluster centres determined from the subset of individuals who self-reported race. For each ancestry group, a cluster centre was chosen to have the median principal component coordinate for each principal component among individuals who self-reported a corresponding race. As in UKB, all individuals that fell within a Euclidean distance radius that enclosed a large majority of individuals self-reporting that race were then assigned to that genetically inferred ancestry. For European ancestry, the radius was set to include 90% of individuals who self-reported as 'white' ($n = 8,157$). For African ancestry, the radius was set to include 90% of individuals who self-reported as 'African American' ($n = 345$). For American ancestry, the radius was set to include 75% of individuals who self-reported as 'Hispanic' or 'Native American' ($n = 906$). For East Asian ancestry, the radius was set to include 75% of individuals who self-reported as 'Asian' and were separated from the majority of samples on PC2 ($n = 342$). For South Asian ancestry, the radius was set to include 75% of individuals who self-reported as 'Asian' and were separated from the majority of samples on PC4 ($n = 144$).

### HLA allele imputation in UK Biobank

The T1DGC reference panel for HLA allele imputation[79] ($n = 5,225$) was first converted to VCF format with variants lifted over to hg19 and merged into multiallelic sites where appropriate. A small number of individuals ($n = 136$) with >2 alleles for at least one multiallelic site were excluded from the reference panel. Imputation of HLA alleles onto phased SNP-array haplotypes[80] was done with BEAGLE[81] (v.5.4) using default parameters, after which imputed alleles were converted back to biallelic variants for genetic association analysis and lifted over to hg38.

### Genome-wide association analyses of viral DNA load phenotypes in UK Biobank

Abundances of reads aligning to reference genomes for EBV, HHV-6B, HHV-7 and anellovirus strains TUS01, VT416 and HD14a were inverse-normal transformed into quantitative phenotypes for GWAS. Individuals were excluded based on the following criteria: not having European genetic ancestry, not having available TOPMed-imputed genotypes (including for chromosome X), and/or having withdrawn, leaving 453,770 individuals for genetic association analyses (447,190 for HHV-6B after removal of individuals with endogenous HHV-6 integration). TOPMed-imputed variants for these individuals were filtered to require minor allele frequency >0.001 and INFO > 0.3. Linear mixed model association tests were performed with BOLT-LMM[82] (v.2.5) to account for relatedness, using the following covariates: age, age squared, sex, genotype array, assessment centre and 20 genetic principal components. SNP array genotypes were used for model fitting, and linkage disequilibrium scores derived from European-ancestry 1KGP samples were used for test statistic calibration. GWAS of germline-inherited endogenous HHV-6A and HHV-6B carrier status were performed using linear regression with the same covariates in individuals of European genetic ancestry, excluding one from each pair of relatives with second-degree or closer relatedness[83].

To identify index variants outside the MHC region (Supplementary Table 7), we first iteratively selected the strongest association and removed any variants within 1 Mb. Index variant pairs within 3 Mb were then evaluated to determine whether they represented independent associations using the following approximation of the association strength of the index variant $i$ conditional on the more strongly associated index variant $j$:

$$\chi_{i|j}^2 \approx \chi_i^2 \left( 1 - r_{ij}\text{sign}(\beta_i\beta_j)\sqrt{\frac{\chi_j^2}{\chi_i^2}} \right)^2$$

as previously described[84]. The less-strongly associated variant *i* was dropped if its approximate conditional association was no longer genome-wide significant ($P < 5 \times 10^{-8}$). Identified index variants were annotated with nearby genes using GENCODE 39 (ref. 85) definitions for protein-coding genes, long non-coding RNAs, and microRNAs. Index variants were annotated as expression quantitative trait loci (eQTLs) and splicing quantitative trait loci (sQTLs) using the v.10 release of GTEx[86]. Follow-up genetic association analyses of variants in the MHC region including both TOPMed-imputed variants and imputed HLA alleles were performed using linear regression with BOLT-LMM on unrelated individuals with European ancestry with the same covariates (Supplementary Table 8).

## Partitioning of heritability between MHC and non-MHC variation

To partition heritability between common variants within and outside the MHC region of the human genome, BOLT-REML[87] (v.2.5) was run on SNP-array genotypes for unrelated UKB participants with European ancestry with variants within the range chr. 6:24000000 to chr. 6:35000000 assigned to one component (MHC region) and all other variants assigned to a second component, with the --remlNoRefine option set. Age, age squared, sex, assessment centre, genotyping array, and the top 20 genetic ancestry principal components were included as covariates.

## Rare variant association analyses in UK Biobank

Gene-level burden masks were generated as previously described[88] using genotypes of rare protein-coding variants in the UKB DRAGEN WGS dataset and genotypes of copy number variants previously ascertained from UKB whole-exome sequencing data[89]. The specific burden masks analysed here for association with viral DNA load used a minor allele frequency threshold of MAF < 0.001 and included missense variants with PrimateAI-3D[90] scores >0.7 merged with loss-of-function SNPs, insertion–deletions and copy number variants, using only the MANE Select transcript[91]. Association tests were performed using linear regression (implemented in BOLT-LMM) on unrelated European-ancestry individuals with age, age squared, sex, assessment centre, genotyping array and the 20 genetic principal components included as covariates. Genes in the MHC region (chr. 6:24000000 to chr. 6:35000000) were excluded given the potential for linkage disequilibrium with strong common-variant associations, leaving 17,589 gene burden masks and a Bonferroni threshold of $5.7 \times 10^{-7}$ (adjusting for five viruses included in these analyses: EBV, HHV-7 and 3 TTVs).

## Genome-wide association analyses of viral DNA load phenotypes in All of Us

GWAS was performed on AoU participants with European ancestry for the following phenotypes: inverse-normal transformed EBV, HHV-6B, HHV-7 and anellovirus strains TUS01, VT416 and HD14a abundances in blood WGS (*n* = 201,168 for EBV, 198,059 for HHV-6B, and 200,091 for other viruses), EBV, HHV-6B, HHV-7 and MCPyV DNA positivity (that is, presence of any EBV reads) in saliva WGS (*n* = 33,164 for EBV, 32,711 for HHV-6B, and 33,050 for other viruses), and inverse-normal transformed EBV (*n* = 16,282) and HHV-7 (*n* = 29,989) abundances in DNA-positive saliva WGS samples. Variants from the allele count/allele frequency (ACAF) threshold call set present in the TOPMed-r2 imputation panel (to exclude those in regions of poor mappability) were filtered to those with minor allele frequency >0.1% and allele count ≥40 in the subset of European ancestry samples used. BOLT-LMM was run with SNP-array genotypes (minor allele frequency >1%, missingness <10% in European-ancestry samples) as model SNPs, with the following covariates: age, age squared, sex, sequencing site, and 16 genetic principal components. For EBV GWAS, samples without a sex call of XX or XY were assigned indicator variables, whereas they were excluded

from GWAS for other viruses due to a minor change in the analytical pipeline during the course of the project.

GWAS was also performed on AoU participants with African ancestry for inverse normal transformed EBV and HHV-7 abundances in blood WGS (*n* = 77,573). Variants from the ACAF threshold call set with minor allele frequency >0.5% in gnomAD v.4.1 African-ancestry samples were filtered to those with minor allele frequency >0.1% and allele count ≥40 in the subset of African-ancestry samples used. BOLT-LMM (using the --lmmInfOnly flag, as the non-infinitesimal mixed model provided a negligible increase in statistical power) was run with SNP-array genotypes (minor allele frequency >1%, missingness <10% in African-ancestry samples) as model SNPs, with the following covariates: age, age squared, sex, sequencing site and 16 genetic principal components.

## Genome-wide association analyses of viral DNA load phenotypes in SPARK

As an auxiliary analysis, we also performed GWAS of viral DNA load phenotypes from SPARK saliva WGS. Abundances of reads aligning to reference genomes for HHV-7, HHV-6B, and Merkel cell polyomavirus were transformed into up to two GWAS phenotypes for each virus: a binary phenotype encoding the presence/absence of any viral reads (HHV-6B, *n* = 9,081 after sample exclusions; MCPyV, *n* = 9,209) and a quantitative phenotype comprising inverse normal transformed non-zero values (HHV-6B, *n* = 6,258; HHV-7, *n* = 7,360). Individuals with non-European genetic ancestry were excluded using a more permissive ancestry definition based on genomic PC1 and PC2 to maximize power, leaving 9,209 individuals for genetic association analyses. For HHV-6B, carriers of eHHV-6B were also excluded. Variants called by DeepVariant were filtered and used to generate ancestry principal components as previously described[83]. Linear mixed model association tests were performed using BOLT-LMM to account for relatedness, using the following covariates: age, square root of age, age squared, sex, sequencing batch, percentage of mapped reads and ten genetic principal components.

We observed that GWAS power in SPARK saliva WGS was much lower than in AoU saliva WGS, as expected given the much smaller sample size. We therefore chose not to meta-analyse saliva viral DNA load GWAS results across AoU and SPARK because the potential power gain was modest (given the much smaller size of the SPARK cohort) and might be negated by the age heterogeneity of SPARK (mostly children) versus AoU (only adults).

## Meta-analysis of GWAS results from UK Biobank and All of Us

Associations with EBV, HHV-6B, HHV-7 and anellovirus strains TUS01, VT416, and HD14a DNA load in blood in AoU were meta-analysed with those from UKB using METAL[92] (v.2020-05-05) in standard error mode (SCHEME STDERR), restricting to variants with minor allele frequency greater than 0.1% (for EBV and HHV-7) or 1% (for HHV-6B and anelloviruses) in both cohorts (ADDFILTER MAF > 0.001 or 0.01), and applying genomic control correction within input studies (GENOMICCONTROL ON). One significantly associated locus in UKB that disagreed in direction of effect between cohorts (and between EBV presence phenotypes generated from left and right halves of the EBV genome; Supplementary Note 5) was filtered.

To compute genetic correlation between viral phenotypes in UKB, AoU, and SPARK, LDSC[93] (v.2.0.0) was run with standard settings and pairs of viral summary statistics as input.

## Generation of lymphocyte percentage phenotype in All of Us

A lymphocyte percentage phenotype was generated from the 'Lymphocytes/100 leukocytes in blood by automated count' phenotype (OMOP Concept Id: 3037511). Only entries with 'percent', 'percent of white blood cells', 'percent' and 'percentage unit' as units were kept, discarding entries with other or missing units as well as values outside the range [0,100]. For individuals with multiple valid measurements,

we took the median value. This left 170,196 people with lymphocyte percentage values, among whom 24,789 individuals with with African ancestry had blood-derived WGS available and were used to evaluate the extent to which the Duffy-null effects on EBV and HHV-7 DNA load were mediated by effects on lymphocyte percentage.

## Measurement of EBV type 1- and type 2-specific alignments

In both UKB and SPARK, unmapped reads and reads that aligned to the chrEBV decoy contig were realigned to the EBV type 1 (NC_007605.1) and EBV type 2 (NC_009334.1) reference genomes using BWA-MEM (providing both reference genomes simultaneously and using 4 computational threads). Read alignments were filtered to those for which both the read and its mate were mapped (samtools view -F 12) and were then collated within 500 bp bins of each of the two reference genomes using mosdepth with the same parameters as above, with the exception that the filter on insert size (-l 100 -u 1000) was dropped, as insert size was undefined in situations in which a read mapped to EBV type 1 and its mate to EBV type 2 or vice versa (for example, if only one read in the pair fell within a type-informative region, and its non-type-informative mate was mapped arbitrarily to type 1 or type 2). A 1 kb offset was added to alignment bin coordinates in the type 1 genome starting at position 85000 to account for differences between the two reference genomes in the *EBNA3A–EBNA3C* region. Alignments to 500 bp bins within the viral genomic regions 35501–38000 and 79001–92500 (corresponding to *EBNA2* and *EBNA3A–EBNA3C*) were considered to be type-specific, and the numbers of type 1-specific and type 2-specific reads for each sample were computed by summing 500 bp depths across these regions (separately for the two EBV genomes), multiplying by 500/150 = (bin size)/(bases per read), and rounding to the nearest integer. In UKB, samples with at least two type 1-specific reads were designated as positive for EBV type 1, and analogously for type 2.

To evaluate the risk of misclassification between type 1 and type 2, we examined the distribution of type 2 versus type 1 reads in SPARK saliva samples, making use of the fact that saliva samples frequently have hundreds of EBV reads that should typically come from only one EBV type (depending on whether the individual is infected with a type 1 or type 2 EBV strain). This analysis showed that as expected, nearly all samples had read counts heavily skewed to either type 1 or type 2 (typically >99% type 1 or >99% type 2), indicating a low rate of misclassification of reads.

## Variation in EBV type 1 versus type 2 frequency by birthplace of UK Biobank participants

For analyses of individuals born in the UK, geographic boundaries for nine regions in England, Wales, Scotland and Northern Ireland were obtained as a GeoJSON file corresponding to 'NUTS, level 1 (January 2018) Boundaries UK BFC' (for EBV type analyses) or 'NUTS, level 2 (January 2018) Boundaries UK BFC' (for total EBV analyses) from the Open Geography portal from the Office for National Statistics (ONS) (https://geoportal.statistics.gov.uk). Birth coordinates for individuals born in the UK (fields 129 and 130) were assigned to regions with these boundaries using the R package sf (v.1.0-20), and the proportion of participants positive for EBV type 2 (among EBV-positive individuals) was estimated as the ratio of the number of samples determined to be EBV type 2-positive to the total number of samples determined to be either EBV type 1-positive or type 2-positive. For country-level analyses, fields 1647 and 20115 were used.

## Genetic association analyses of MHC variants with EBV-type DNA load phenotypes

Variants in the MHC region of the human genome (including imputed HLA alleles) were tested for association with three binary EBV DNA load phenotypes derived from EBV type-specific read alignments. The first two phenotypes coded EBV type 1-positive individuals (respectively, type 2-positive individuals) as cases and all other individuals as controls.

The third phenotype, used in case-case association analyses of EBV type 1 versus type 2 positivity, coded individuals who were type 2-positive and lacked type 1-specific alignments as cases ($n = 1,366$), and those who were type 1-positive and lacked type 2-specific alignments as controls ($n = 9,817$); a small number of individuals determined to be positive for both type 1 and type 2 were excluded ($n = 126$). Association tests were performed using linear regression (implemented in BOLT-LMM) on unrelated individuals with European ancestry with age, age squared, sex, genotype array, assessment centre and 20 genetic principal components as covariates.

## Association analyses of viral DNA load with biological and clinical phenotypes in UK Biobank

Binarized viral DNA positivity phenotypes were tested for association with binary disease phenotypes in ICD-10 categories derived from UKB 'first occurrence' data fields (fields under category 1712, which merged data from electronic health records and self-report) and cancer registry data (field 40006). For each virus, we tested only binary disease phenotypes for which at least 5 cases were expected among viral DNA-positive individuals for that virus (comprising 493–1,413 tests for each of eight common viruses tested: EBV, HHV-6B, HHV-7, three TTVs and eHHV-6A and eHHV-6B). Bonferroni correction was applied to the full set of tests performed across all eight viruses. Association tests were performed using logistic regression with Firth correction using the Wald approximation (pl = FALSE) as implemented in the logistf R package (v.1.26.0) on participants with European ancestry with age, age squared, sex, assessment centre and the 20 genetic principal components as covariates.

Viral DNA positivity phenotypes were also tested for association with quantitative blood phenotypes in UKB: blood counts (category 100081), blood biochemistry (category 17518), and NMR metabolomics (category 220). Association tests were performed using linear regression on individuals with European ancestry with the above covariates, and Bonferroni correction was applied considering all pairwise tests of quantitative blood phenotypes with viral DNA positivity phenotypes.

Binary immunosuppressive drug phenotypes were generated by aggregating synonymous terms from self-reported medication data (category 100075). Specifically, methotrexate was generated from the union of individuals reporting 'methotrexate' or 'mtx – methotrexate'; cyclosporin from 'cya - cyclosporin', 'ciclosporin product', 'csa - cyclosporin a', 'cya - cyclosporin a', 'cyclosporin', 'cyclosporin product' or 'ciclosporin'; and corticosteroids from 'prednisone', 'prednisolone', 'methylprednisolone', 'prednisolone product', 'dexamethasone', 'fludrocortisone', 'hydrocortisone', 'cortisone product', 'hydrocortisone product' or 'cortisone'.

Quantitative smoking phenotypes in UKB were generated from the pack-years and cigarettes per day phenotypes by encoding never smokers (and, for cigarettes per day, former smokers) as 0 values. These smoking phenotypes were tested for association with viral DNA positivity phenotypes using linear regression on individuals with European ancestry with the above covariates.

## Mendelian randomization to identify causal effects of EBV DNA load on disease

Instrument variables for Mendelian randomization analyses were identified as the 44 lead variants at non-MHC loci from the GWAS meta-analysis of EBV DNA load in UKB and AoU. The MHC region was excluded from Mendelian randomization analyses given the likelihood of linkage disequilibrium generating pleiotropic effects of MHC haplotypes on many immune-related phenotypes. To minimize the impact of sample overlap between cohorts used in the exposure GWAS (EBV DNA load in UKB+AoU blood WGS) and outcome GWAS (disease phenotypes in FinnGen plus UKB plus MVP), we regenerated GWAS summary statistics for EBV DNA load in UKB after restricting to a control-only sub-cohort[94]. Specifically, we reran BOLT-LMM after removing UKB participants with the

following ICD-10 phenotypes: G35, C81, C82, C83, C84, C85, C91, M05, M06 and M32. GWAS results from this control-only UKB sub-cohort were then meta-analysed with AoU associations to generate effect sizes and standard errors for use in Mendelian randomization.

GWAS summary statistics for outcome phenotypes of interest were downloaded from https://mvp-ukbb.finngen.fi/ after applying for access. In these GWAS, FinnGen phenotypes had been harmonized over ICD-8, ICD-9 and ICD-10, cancer-specific ICD-O-3, (NOMESCO) procedure codes, Finnish-specific Social Insurance Institute (KELA) drug reimbursement codes and ATC-codes collected from various registries. MVP phenotypes had been defined by ICD-9 and ICD-10 codes from electronic health records grouped into corresponding phecodes, with case status defined as having two or more phecode-mapped ICD-9 or ICD-10 codes. Meta-analyses had been performed by identifying phenotypes with concordant endpoints, as described at https://finngen. gitbook.io/documentation/methods/meta-analysis. We used the MendelianRandomization R package[95] (v.0.10.0) with default settings to generate estimates of causal effect sizes of EBV DNA load on disease phenotypes using the weighted median, inverse-variance weighted (IVW), MR–Egger, and contamination mixture (ConMix) approaches. For weighted median, IVW, and MR–Egger, robust regression with penalized weights was used to account for invalid instrument variables[96].

We caution that effect size estimates from Mendelian randomization (here, the increase in log-odds of disease per s.d. increase in EBV reads; Fig. 5e) are expected to be overestimated when the exposure variable (here, EBV DNA load) is measured with high noise. For example, were the exposure phenotype to be randomly permuted in a subset of samples, this would leave the s.d. of the exposure unchanged, but it would shrink down the effect sizes (betas) of the instrumental variables. The betas of instrument variables are used as independent variables in the regression analysis used to estimate Mendelian randomization effect sizes ($x$ axis of Fig. 5f), such that shrinking the betas of the instrument variables would then increase the slope of the Mendelian randomization regression, causing the estimated effect size from Mendelian randomization to increase. This behaviour contrasts with how measurement noise in the exposure variable impacts direct analyses of association with the outcome variable (Fig. 5g): in such analyses, noise reduces the observed association.

### Reporting summary

Further information on research design is available in the Nature Portfolio Reporting Summary linked to this article.

## Data availability

The following data resources are available by application: UK Biobank (http://www.ukbiobank.ac.uk/), All of Us Research Program (https://allofus.nih.gov/), SFARI SPARK (https://www.sfari.org/resource/spark/), MVP-Finngen-UKBB meta-analysis summary statistics (https://mvp-ukbb.finngen.fi/) and T1DGC HLA imputation panel (https://repository.niddk.nih.gov/study/173). The following data resources are publicly available: human reference genome build GRCh38 (https://ftp.1000genomes.ebi.ac.uk/vol1/ftp/technical/reference/GRCh38_reference_genome/), TOPMed-r2 imputation panel variant list (https://imputation.biodatacatalyst.nhlbi.nih.gov/), gnomAD v.4.1 variant call set (https://gnomad.broadinstitute.org/), linkage disequilibrium score resources (https://alkesgroup.broadinstitute.org/LDSCORE/), NCBI Virus for reference sequences (https://www.ncbi.nlm.nih.gov/labs/virus/vssi/), PrimateAI-3D scores (https://primateai3d.basespace.illumina.com/), GENCODE 39 definitions (https://www.gencodegenes.org/) and GTEx expression and splice quantitative trait associations (https://gtexportal.org/home/). Full viral DNA load GWAS summary statistics are available from the GWAS Catalog under accessions GCST90809801 to GCST90809829. Viral DNA load phenotypes for AoU participants are available in the Controlled Tier

workspace (https://workbench.researchallofus.org/workspaces/aou-rw-77ec99c5/kamitakietalthednaviromevarieswithhumangenesandenvironments/data).

## Code availability

The following publicly available software resources were used: bwa (v.0.7.18, https://bio-bwa.sourceforge.net/), mosdepth (v.0.3.9, https://github.com/brentp/mosdepth), bcftools (v.1.14, http://www.htslib.org/), samtools (v.1.15.1, http://www.htslib.org/), plink (v.1.90b6.26 and v.2.00a3.7, https://www.cog-genomics.org/plink/), BEAGLE (v.5.4, https://faculty.washington.edu/browning/beagle/beagle.html), BOLT-LMM (v.2.5, https://alkesgroup.broadinstitute.org/BOLT-LMM/), METAL (v.2020-05-05, https://genome.sph.umich.edu/wiki/METAL), qqman R package (v.0.1.8, https://cran.r-project.org/web/packages/qqman/index.html), sf R package (v.1.0-20, https://cran.r-project.org/web/packages/sf/index.html), logistf R package (v.1.26.1, https://cran.r-project.org/web/packages/logistf/index.html) and MendelianRandomization R package (v.0.10.0, https://cran.r-project.org/web/packages/MendelianRandomization/index.html).

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

**Acknowledgements** We thank C. Usher for edits to text and figures. This research was conducted using the UK Biobank Resource under application number 40709. We are grateful to all of the families in SPARK, the SPARK clinical sites and SPARK staff. We acknowledge access granted to SPARK genetic data on SFARI Base. N.K. was supported by US NIH Fellowship F31 DE034283. S.A.M. was supported by US NIH grant R01 HG006855 and the Howard Hughes Medical Institute. P.-R.L. was supported by US NIH grants R56 HG012698 and R01 HG013110 and a Burroughs Wellcome Fund Career Award at the Scientific Interfaces. The funders had no role in study design, data collection and analysis, the decision to publish or the preparation of the manuscript. The content is solely the responsibility of the authors and does not necessarily represent the official views of the NIH. Computational analyses were performed on the O2 High Performance Compute Cluster supported by the Research Computing Group at Harvard Medical School (http://rc.hms.harvard.edu), the UKB Research Analysis Platform, and the All of Us Researcher Workbench. The All of Us Research Program is supported by the NIH, Office of the Director: Regional Medical Centers: 1 OT2 OD026549; 1 OT2 OD026554; 1 OT2 OD026557; 1 OT2 OD026556; 1 OT2 OD026550; 1 OT2 OD026552; 1 OT2 OD026553; 1 OT2 OD026548; 1 OT2 OD026551; 1 OT2 OD026555; IAA no. AOD 16037; Federally Qualified Health Centers: HHSN 263201600085U; Data and Research Center: 5 U2C OD023196; Biobank: 1 U24 OD023121; The Participant Center: U24 OD023176; Participant Technology Systems Center: 1 U24 OD023163; Communications and Engagement: 3 OT2 OD023205; 3 OT2 OD023206; and Community Partners: 1 OT2 OD025277; 3 OT2 OD025315; 1 OT2 OD025337; and 1 OT2 OD025276. In addition, the All of Us Research Program would not be possible without the partnership of its participants.

**Author contributions** N.K., S.A.M. and P.-R.L. conceived the study design. N.K. performed computational and statistical analyses. D.T. annotated functional coding variants and generated the gene burden masks for rare variant associations in UK Biobank. N.K., S.A.M. and P.-R.L. wrote the manuscript with contributions from all authors.

**Competing interests** The authors declare no competing interests.

**Additional information**
**Correspondence and requests for materials** should be addressed to Nolan Kamitaki, Steven A. McCarroll or Po-Ru Loh.

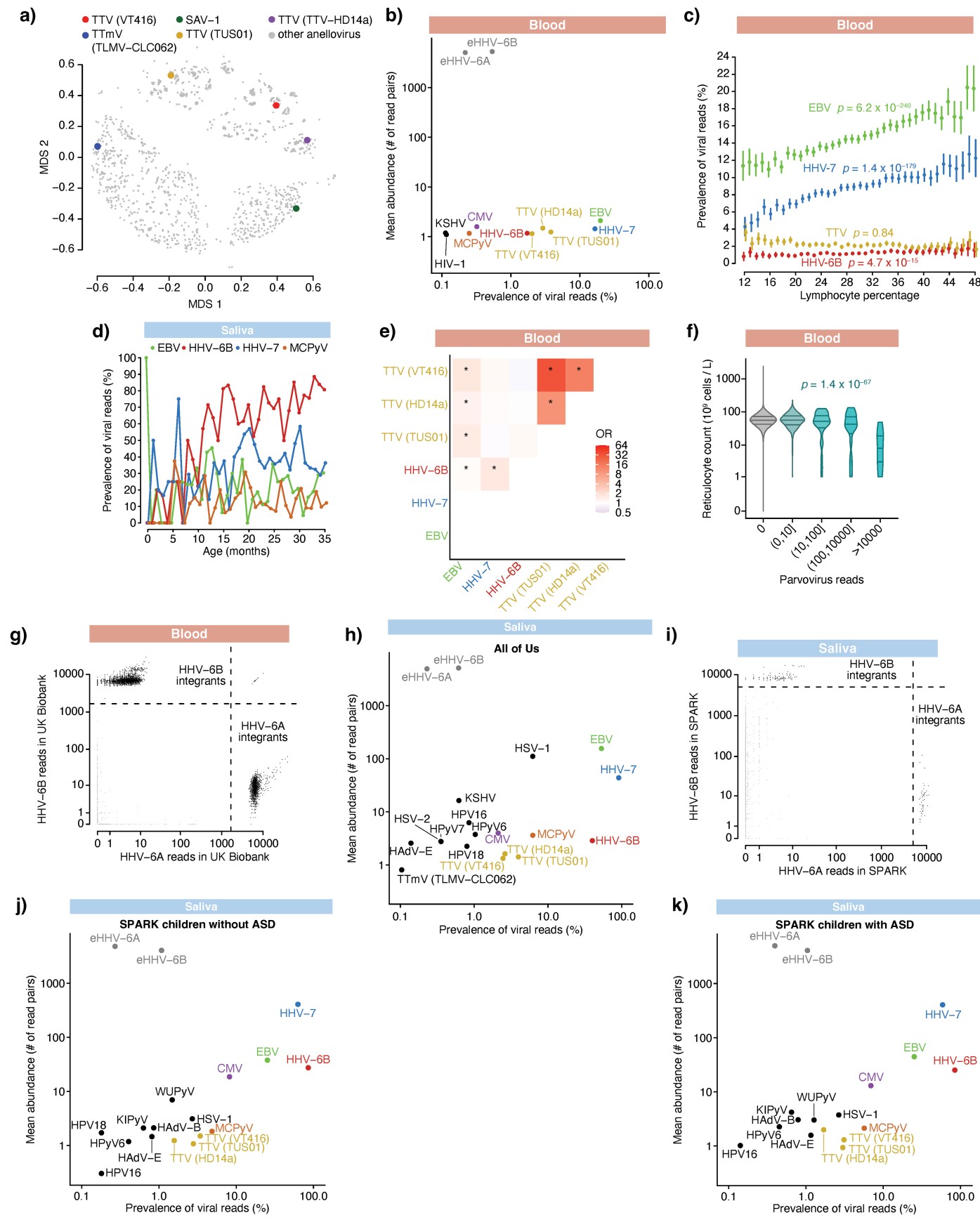

**Extended Data Fig. 1** | See next page for caption.

**Extended Data Fig. 1 | Viral DNA load quantification, associations with blood counts, and identification of endogenous HHV-6 chromosomal integrants.** a) Locations of the five representative anellovirus reference genomes included in the viral reference panel relative to other anellovirus genomes in the multidimensional scaling projection from ref. 73. b) Prevalence and abundance of viral DNA sequences in blood samples from AoU (n = 365,918). c) Prevalence of viral DNA sequences in blood by lymphocyte percentage (rounded to the nearest integer) in European-ancestry UKB participants (n = 441,026). P-values are from a model including sex, age, age-squared, assessment center, and 20 genetic principal components as covariates. Error bars, 95% CIs. d) Prevalence of viral DNA sequences in saliva-derived WGS by age (rounded to the nearest month) in SPARK for children under 3 years old (total n = 576; n = 1, 2, 5, 6, 4, 8, 4, 4 for the 0-month to 7-month bins; 7 ≤ n ≤ 45 for remaining bins). e) Pairwise enrichments of co-observing reads from two viruses in the same UKB sample. * for $p < 0.01$. f) Reticulocyte counts of European ancestry UKB participants (n = 434,573, y-axis) stratified by number of reads mapping to the parvovirus B19 genome (x-axis). P-value is from regression of log-transformed reticulocyte counts on log-transformed B19 reads with sex, age, age-squared, assessment center, and 20 genetic principal components as covariates. g) Numbers of reads aligned to the HHV-6A (x-axis) and HHV-6B (y-axis) genomes in blood-derived WGS from UKB. Threshold lines shown were used to determine carriers of endogenous HHV-6A or HHV-6B. A few individuals carrying both integrants are in the top right quadrant. h) Prevalence and abundance of viral DNA sequences in saliva samples from AoU (n = 48,899). i) Analogous to g, for saliva-derived WGS from SPARK (n = 12,519). j) Prevalence and abundance of viral DNA sequences in saliva samples from children without autism spectrum disorder in SPARK (n = 2,220). k) Prevalence and abundance of viral DNA sequences in saliva samples from children with autism spectrum disorder in SPARK (n = 3,540).

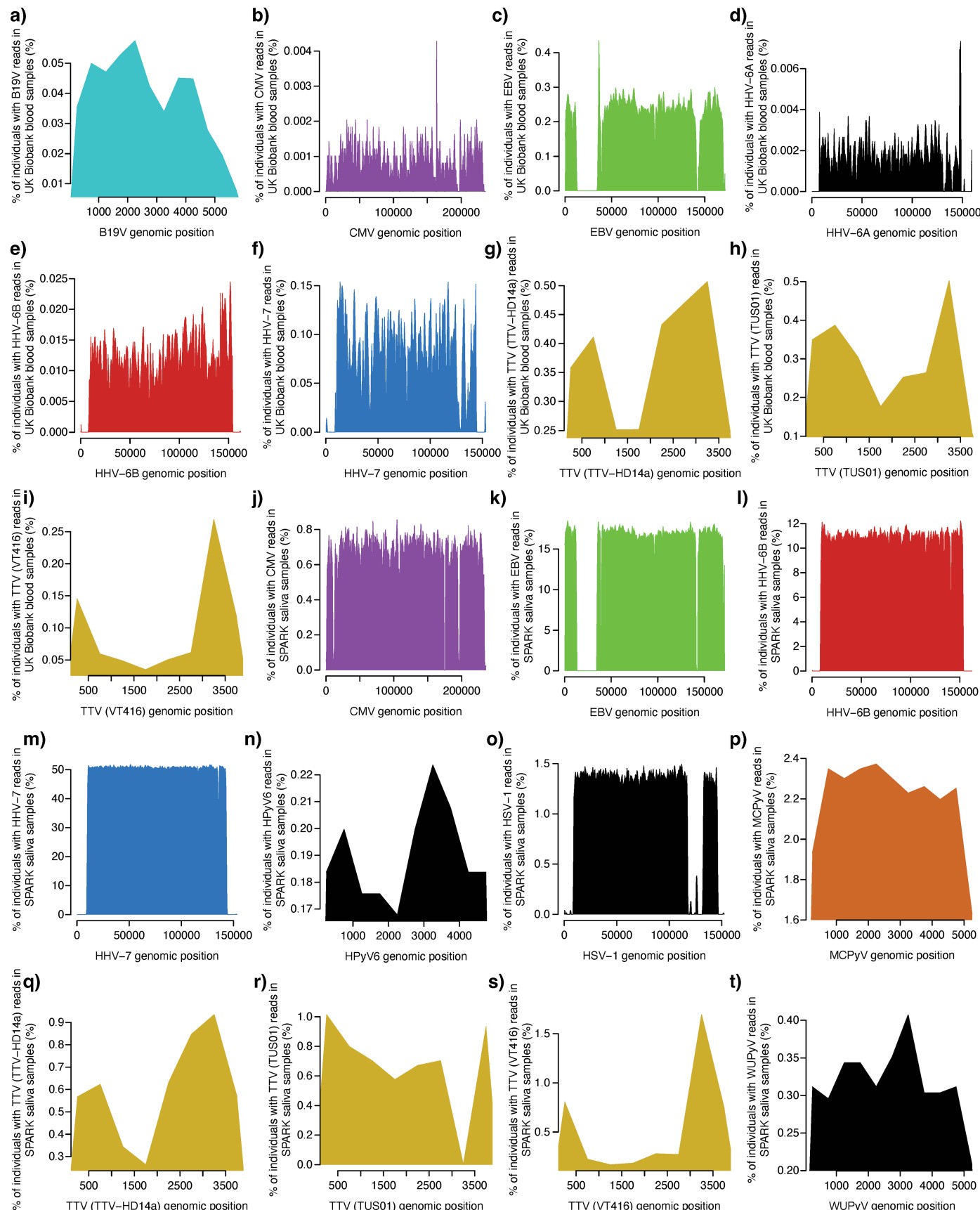

**Extended Data Fig. 2 | Coverage of viral genomes by aligned WGS reads.** For each virus commonly detected in UKB blood WGS or SPARK saliva WGS, the proportion of samples containing reads in 500 bp regions spanning the indicated virus' genome is plotted. Evenness of coverage indicates that reads are arising from sparse sampling of viral genomes across people; misalignments from homologous sequences in the human genome would appear as spikes in viral genomic coverage. Additional metrics on coverage in each cohort can be found in Supplementary Tables 2–5.

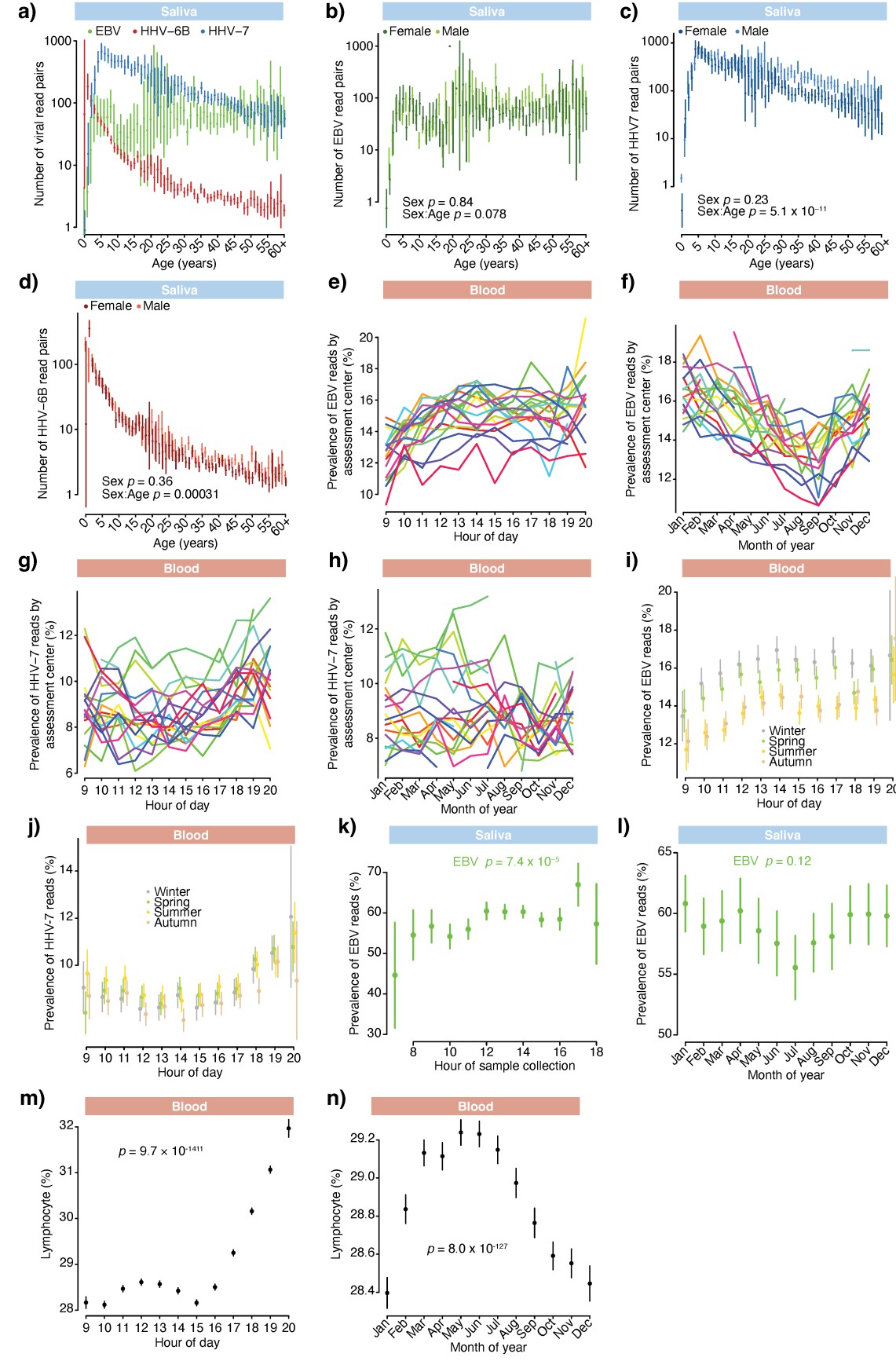

**Extended Data Fig. 3** | See next page for caption.

**Extended Data Fig. 3 | Additional data supporting age, sex, circadian, and seasonal associations with viral DNA load in saliva.** a) Abundance of viral DNA sequences in saliva-derived WGS by age (rounded to the nearest year) in SPARK. Sample sizes for each virus are in the following panel legends. Error bars, 95% CIs. b) Abundance of EBV DNA sequences by age in SPARK saliva WGS, stratified by sex. P-values are from a joint model containing both sex and an interaction term between sex and age (n = 4,753). Error bars, 95% CIs. c) Analogous to b, for HHV-7 DNA sequences (n = 9,978). d) Analogous to b, for HHV-6B DNA sequences (n = 8,528). e) Prevalence of EBV DNA sequences in UKB blood WGS by time of day of sample collection (rounded to the nearest hour) split by assessment center (n = 490,136). f) Prevalence of EBV DNA sequences in UKB blood WGS data by month of sample collection split by assessment center. g) Prevalence of HHV-7 DNA sequences in UKB blood WGS by time of day of sample collection (rounded to the nearest hour) split by assessment center. h) Prevalence of HHV-7 DNA sequences in UKB blood WGS data by month of sample collection split by assessment center. i) Prevalence of EBV DNA sequences in UKB blood WGS by time of day of sample collection (rounded to the nearest hour) split by season (winter, Dec-Feb; spring, Mar-May; summer, Jun-Aug; autumn, Sep-Nov). j) Prevalence of HHV-7 DNA sequences in UKB blood WGS by time of day of sample collection (rounded to the nearest hour) split by season. k) Prevalence of EBV DNA sequences by sample collection time of day (rounded to the nearest hour) in AoU saliva WGS (n = 18,751). P-values are from an ANOVA test between models with and without hour of the day. Error bars, 95% CIs. l) Prevalence of EBV DNA sequences by sample collection month of year in AoU saliva WGS. P-values are from an ANOVA test between models with and without collection month. Error bars, 95% CIs. m) Lymphocyte percentage in UKB blood samples by sample collection time of day (n = 472,115). Error bars, 95% CIs. n) Lymphocyte percentage in UKB blood samples by sample collection month. Error bars, 95% CIs.

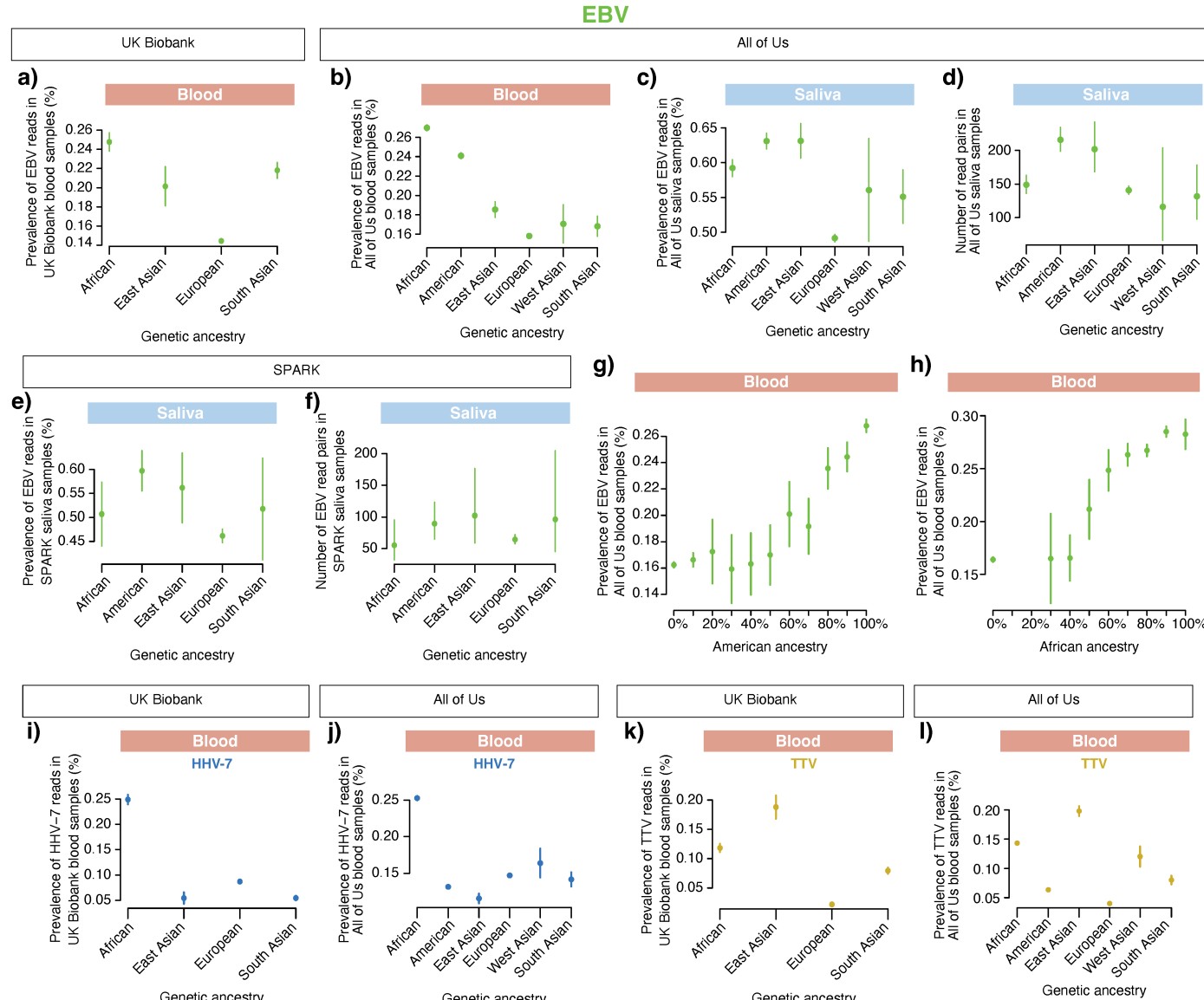

**Extended Data Fig. 4 | Variation in viral DNA load in blood and saliva by genetic ancestry.** a) Prevalence (population frequency) of detectable EBV DNA positivity (reads that align to the EBV genome) in blood-derived WGS data by ancestry in UKB (genetically inferred, n = 474,353). Error bars, 95% CIs. Per-ancestry prevalences and abundances can be found in Supplementary Table 6. b) Prevalence of EBV DNA positivity in blood-derived WGS data by genetic ancestry in AoU (n = 365,918). Error bars, 95% CIs. c) Prevalence of EBV DNA positivity in saliva-derived WGS data by genetic ancestry in AoU (n = 48,899). Error bars, 95% CIs. d) Abundance of EBV DNA sequences in saliva-derived WGS data by genetic ancestry in AoU (n = 25,915). Error bars, 95% CIs. e) Analogous to c, for saliva-derived WGS in adults (≥20 years old) from SPARK (n = 5,691). f) Analogous to d, for saliva-derived WGS in adults from SPARK (n = 2,730). g) Prevalence of EBV DNA positivity in blood-derived

WGS data from AoU by proportion of estimated American ancestry admixture (for individuals with estimated >90% European plus American ancestry, n = 184,514). Error bars, 95% CIs. h) Analogous to g, for proportion of estimated African ancestry admixture for AoU participants with estimated >90% European plus African ancestry (n = 168,651). Admixture proportions of 10–30% African ancestry were excluded due to low sample numbers (n < 500). i) Prevalence of HHV-7 DNA positivity in blood-derived WGS data by genetic ancestry in UKB (n = 474,353). Error bars, 95% CIs. j) Prevalence of HHV-7 DNA positivity in blood-derived WGS data by genetic ancestry in AoU (n = 365,918). Error bars, 95% CIs. k) Prevalence of anellovirus DNA positivity in blood-derived WGS data by genetic ancestry in UKB (n = 474,353). Error bars, 95% CIs. l) Prevalence of anellovirus DNA positivity in blood-derived WGS data by genetic ancestry in AoU (n = 365,918). Error bars, 95% CIs.

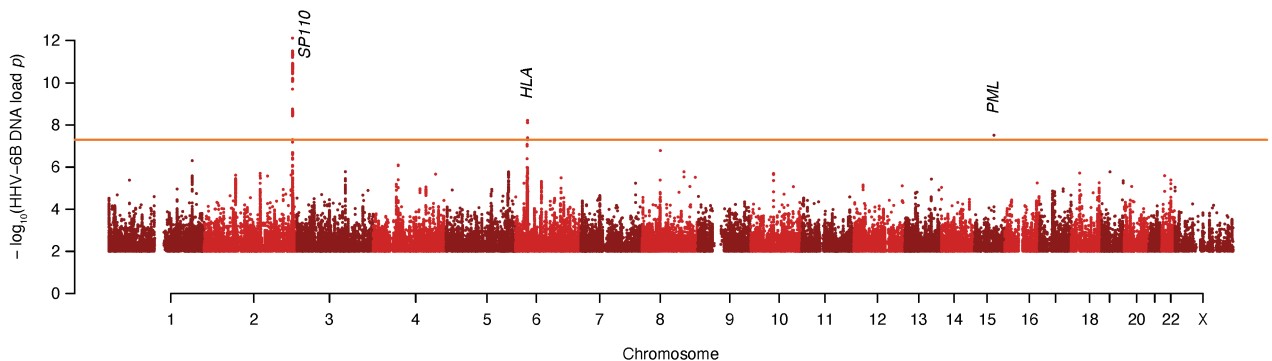

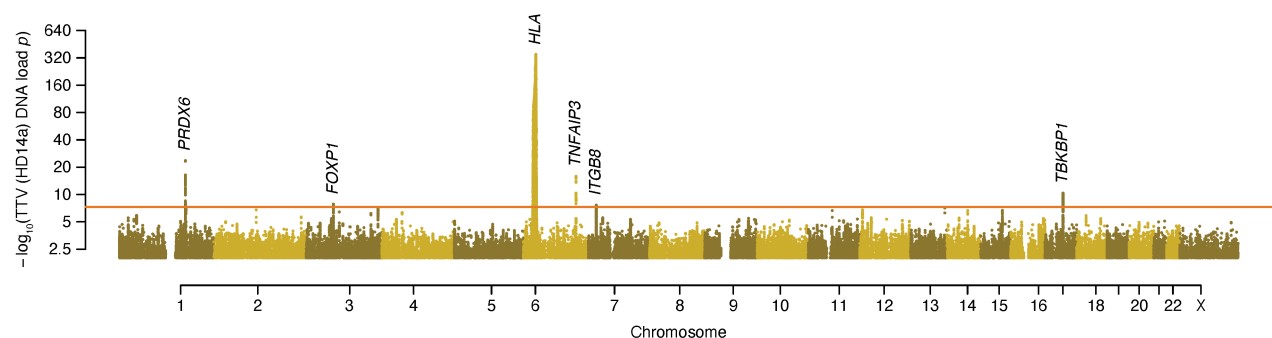

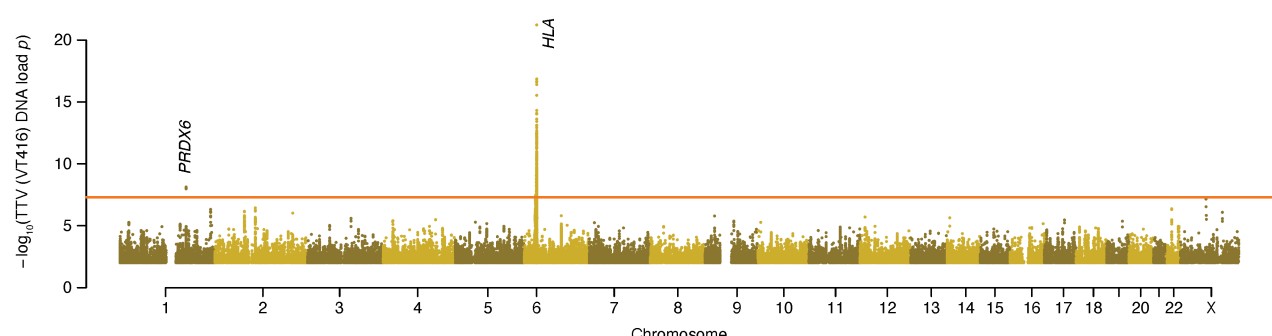

**Extended Data Fig. 5 | Genome-wide association analyses of HHV-6B and anellovirus (HD14a and VT416 TTV strains) DNA load in blood.** a) Genome-wide associations of human genetic variants with HHV-6B DNA load in blood (inverse-normal transformed abundance), meta-analyzed across UKB (n = 447,190) and AoU (n = 198,059) European-ancestry sub-cohorts. b) Analogous to a, for the anellovirus HD14a strain (UKB n = 453,770; AoU n = 200,091). c) Analogous to a, for the anellovirus VT416 strain (UKB n = 453,770; AoU n = 200,091).

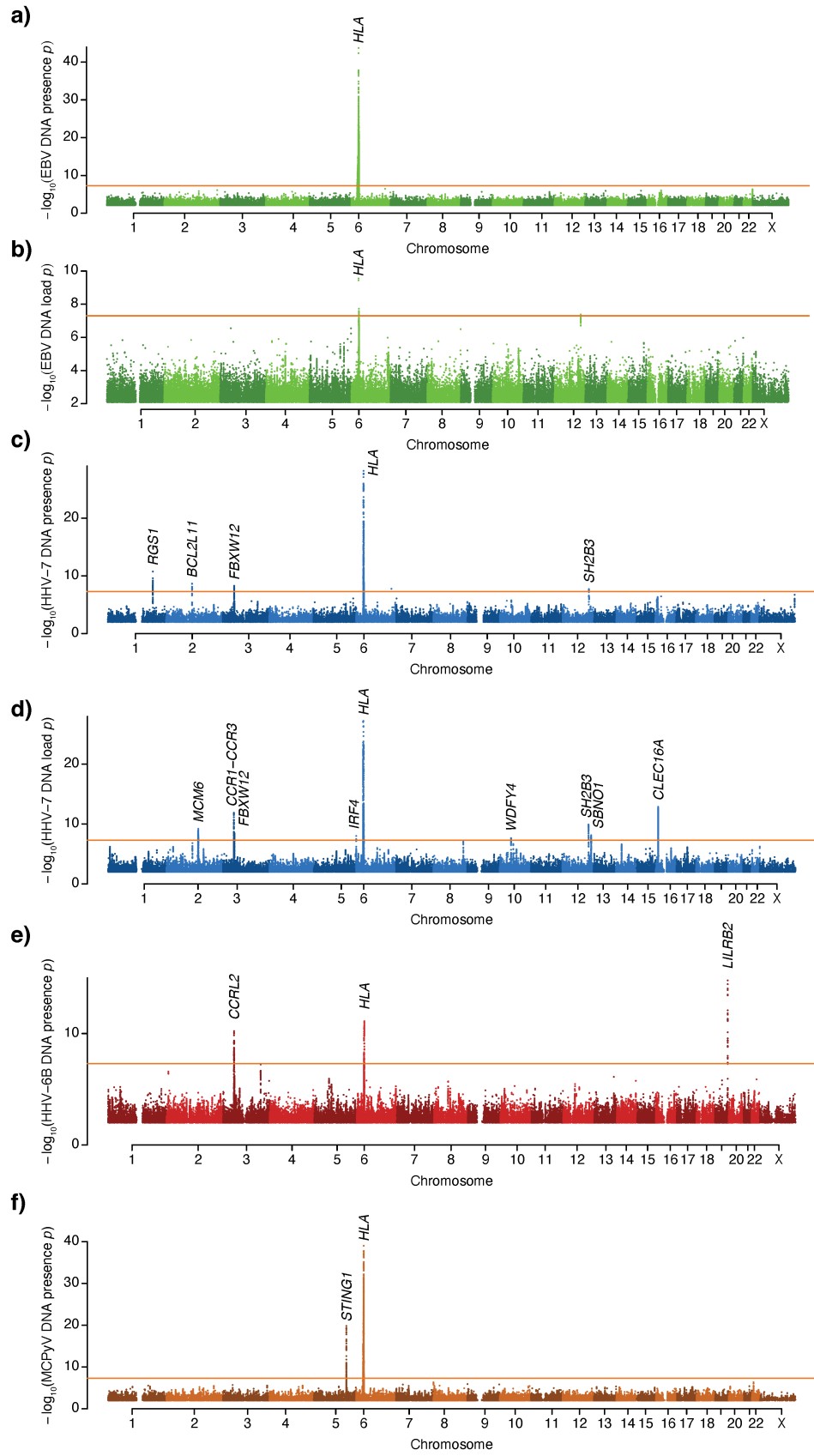

**Extended Data Fig. 6 |** See next page for caption.

**Extended Data Fig. 6 | Genome-wide association analyses of viral DNA load in saliva.** a) Genome-wide associations of human genetic variants with EBV DNA positivity in saliva-derived WGS from European-ancestry AoU participants (n = 33,164). Given the bimodal distribution of viral DNA load in saliva, in which samples containing virally-aligned reads often had hundreds or thousands of reads, we performed GWAS both on binarized viral DNA load (presence/absence of any reads from the virus) and quantitative viral DNA load phenotypes (inverse-normal transformed abundance, restricted to individuals whose saliva WGS had detectable viral DNA) to capture effects on the frequency and amount of secreted virus. b) Analogous to a, for EBV abundance (n = 16,282). c) Analogous to a, for HHV-7 positivity (n = 33,050). d) Analogous to a, for HHV-7 abundance (n = 29,989). e) Analogous to a, for HHV-6B positivity (n = 32,711). Carriers of endogenous HHV-6A or HHV-6B germline integrations were excluded. f) Analogous to a, for Merkel cell polyomavirus positivity (n = 33,050).

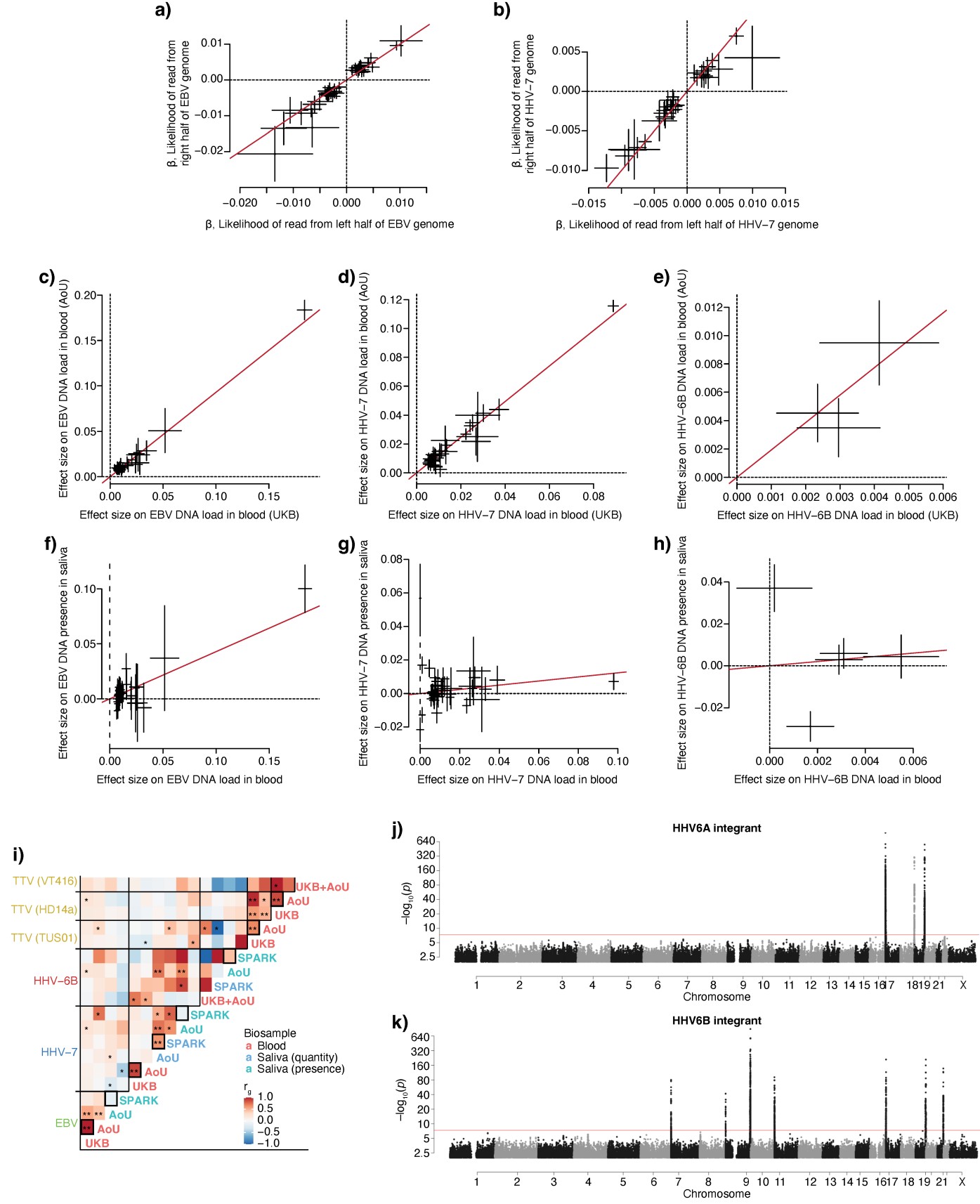

**Extended Data Fig. 7** | See next page for caption.

**Extended Data Fig. 7 | Robustness of genetic associations with EBV and HHV-7 DNA load and GWAS for HHV-6 integration sites.** a) Effect sizes for index variants at 44 independent non-MHC EBV DNA load-associated loci for association with presence of reads aligned to the left (x-axis) or right (y-axis) half of the EBV genome (n = 419,249 unrelated European ancestry participants). b) Analogous to a, for index variants at 34 independent non-MHC HHV-7 DNA load-associated loci. Two lower frequency variants of large effect size were excluded to facilitate visualization of the remaining variants. c) Effect sizes for index variants at 45 independent EBV blood DNA load-associated loci for association with EBV DNA load in blood samples from UKB (x-axis; n = 453,770) and AoU (y-axis; n = 201,168). Line shown is the median ratio of effect sizes between UKB and AoU. d) Analogous to c, for index variants at 35 independent HHV-7 blood DNA load-associated loci. Two lower frequency variants of large effect size were excluded to facilitate visualization of the remaining variants. e) Analogous to c, for index variants at 3 independent HHV-6B blood DNA load-associated loci. f) Effect sizes for index variants at 45 independent EBV blood or saliva DNA load-associated loci for association with EBV DNA load in blood samples from UKB+AoU (x-axis; n = 453,770 + 201,168) and EBV presence in saliva samples from AoU (y-axis; n = 33,164). Line shown is the median ratio of effect sizes between UKB+AoU blood and AoU saliva; technical variation (e.g., in sample processing or DNA sequencing for saliva vs. blood samples) would generally be expected to increase or decrease signal-to-noise to the same extent across different genetic effects, which would not explain the observed heterogeneity in saliva vs. blood effect size ratios. g) Analogous to f, for index variants at 40 independent HHV-7 blood or saliva DNA load-associated loci. Two lower frequency variants of large effect size were excluded to facilitate visualization of the remaining variants. h) Analogous to f, for index variants at 5 independent HHV-6B blood or saliva DNA load-associated loci. i) Genetic correlations between viral phenotypes in UKB, AoU, and SPARK. UKB and AoU were included separately when statistical power was sufficient within each cohort; for HHV6B and TTV (VT416), only the meta-analysis had sufficient power. Borders around cells indicate the same phenotype in different cohorts (e.g., UKB and AoU blood samples). * indicates $p < 0.1$ and ** for $p < 0.01$. j) Genome-wide associations of human genetic variants with carrier status for endogenous HHV-6A germline integration (eHHV-6A) in unrelated European-ancestry UKB participants (n = 419,249). Peritelomeric peaks indicate haplotypes in linkage disequilibrium with telomeric HHV-6A insertions. k) Analogous to c, for endogenous HHV-6B germline integration (eHHV-6B).

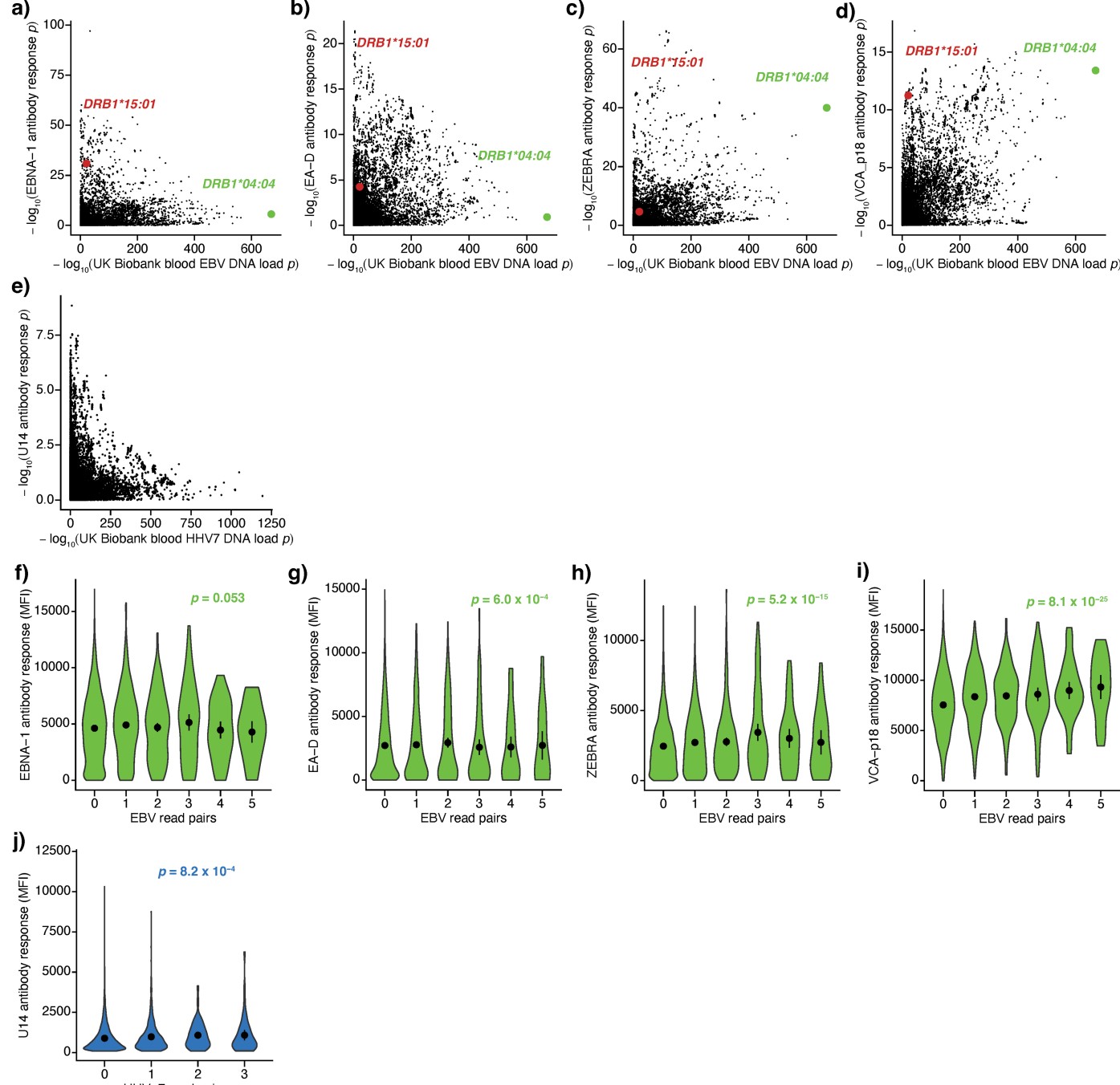

**Extended Data Fig. 8 | Analyses of serology data and comparison with viral DNA load in blood.** a) Associations of variants in the MHC region of the human genome with quantitative antibody response to EBV-encoded EBNA-1 in UKB serology measurements (y-axis, n = 8,100) and EBV DNA load in UKB blood WGS (x-axis; n = 453,770). Serology measurements were from unrelated European ancestry participants and associations were performed with BOLT-LMM using standard covariates. b) Analogous to a, for quantitative antibody response to EA-D. c) Analogous to a, for quantitative antibody response to ZEBRA. d) Analogous to a, for quantitative antibody response to VCA-p18. e) Associations to quantitative antibody response to HHV-7 encoded U14 (y-axis) and HHV-7 DNA load (x-axis; n = 453,770). f) Quantitative antibody response to EBV-encoded EBNA-1 (y-axis) stratified by EBV DNA fragment count in UKB blood

WGS (x-axis) in EBV seropositive individuals (n = 8,325). P-value is from a linear regression model with standard covariates and rank-based inverse normal transformed EBV read counts. MFI, median fluorescence intensity. g) Analogous to f, for quantitative antibody response to EBV-encoded EA-D. h) Analogous to f, for quantitative antibody response to EBV-encoded ZEBRA. i) Analogous to f, for quantitative antibody response to EBV-encoded VCA-p18. j) Quantitative antibody response to HHV-7-encoded U14 (y-axis) stratified by HHV-7 DNA fragment count in UKB blood WGS (x-axis) in HHV-7 seropositive individuals (n = 8,341). P-value is from a linear regression model including standard covariates and rank-based inverse normal transformed HHV-7 read counts.

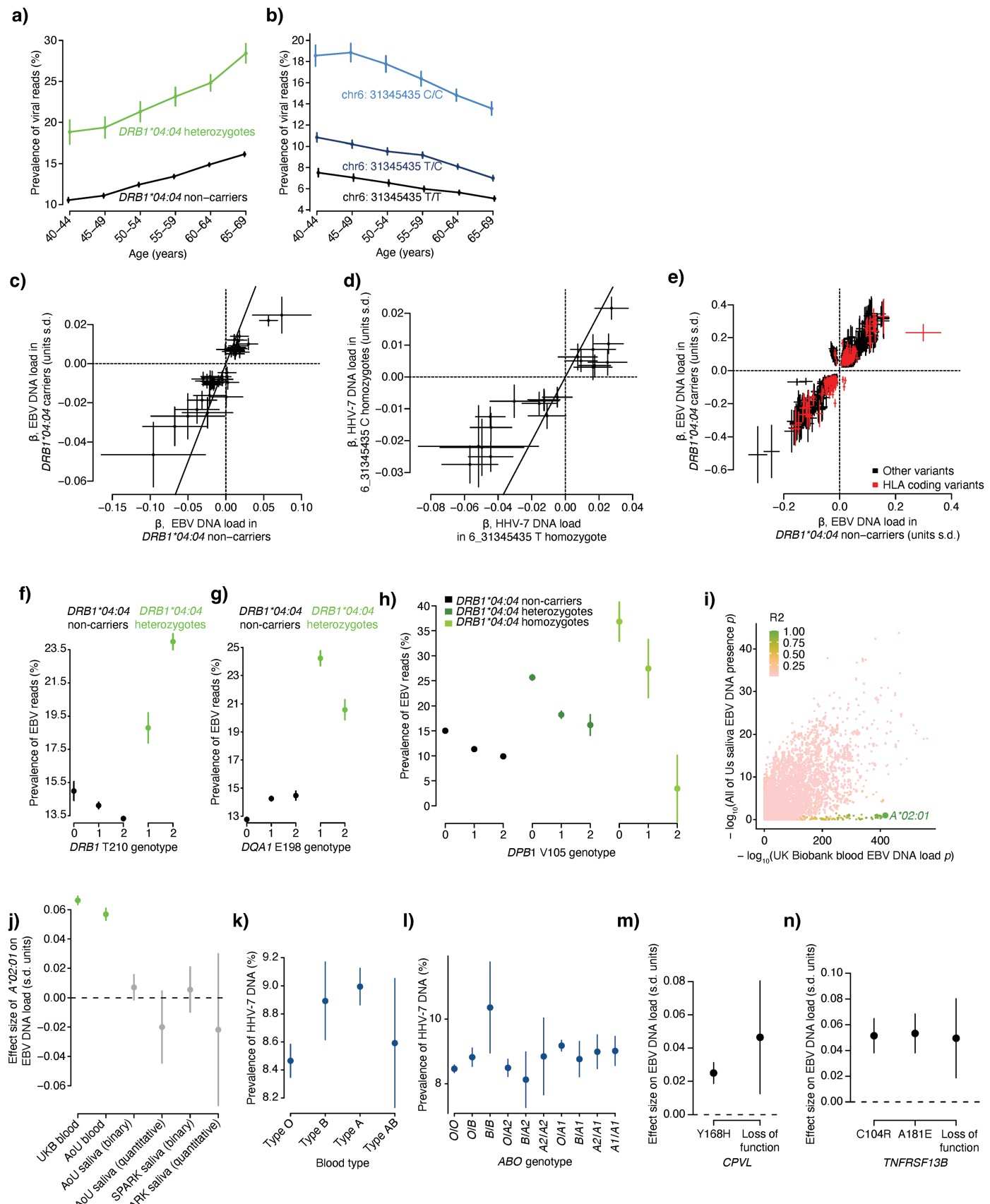

**Extended Data Fig. 9** | See next page for caption.

**Extended Data Fig. 9 | Non-additive genetic effects and associations of blood type and coding variants with viral DNA load in blood.** a) Prevalence of detectable EBV positivity by age in UKB blood WGS, stratified by *HLA-DRB1\*04:04* genotype (n = 452,318 European ancestry participants). Error bars, 95% CIs. b) Prevalence of detectable HHV-7 positivity by age in UKB blood WGS, stratified by genotype for a SNP in the MHC region at chr6:31345435. Error bars, 95% CIs. c) Effect sizes of index variants at 44 independent non-MHC loci associated with EBV DNA load in UKB blood WGS in *DRB1\*04:04* carriers (x-axis, n = 34,089) and *DRB1\*04:04* non-carriers (y-axis, n = 371,339). d) Effect sizes of index variants at 23 independent non-MHC loci associated with HHV-7 DNA load in UKB blood WGS in chr6:31345435 T/T homozygotes (x-axis, n = 167,251) and C/C homozygotes (y-axis, n = 57,034). Three lower frequency variants of large effect size were excluded to facilitate visualization of the remaining variants. e) Effect sizes of variants in the MHC region in *DRB1\*04:04* carriers (x-axis, n = 34,089) and *DRB1\*04:04* non-carriers (y-axis, n = 371,339) on EBV DNA load in UKB blood WGS. Only variants with associations that reached genome-wide significance in both sample subsets are shown. HLA coding variants and classical alleles are highlighted in red. f) Prevalence of detectable EBV positivity by *HLA-DRB1\*04:04* and *HLA-DRB1* T210 genotype combinations in UKB blood WGS (n = 454,545 European ancestry participants). Error bars, 95% CIs. g) Prevalence of detectable EBV positivity by *HLA-DRB1\*04:04* and *HLA-DQA1* E198 genotype combinations in UKB blood WGS (n = 454,545 European ancestry participants). Error bars, 95% CIs. h) Prevalence of EBV DNA positivity by *HLA-DRB1\*04:04* and *HLA-DPB1* V105 genotype combinations in UKB (n = 454,545 European ancestry participants). Error bars, 95% CIs. i) Associations of common genetic variants in the MHC region with EBV DNA load in blood (UKB, n = 453,770 European ancestry participants; x-axis) and saliva (AoU, n = 33,168 European ancestry participants; y-axis). Variants (dots in the scatter plot) are colored by linkage disequilibrium with rs12153924 (green-to-pink shading), which tags the *HLA-A\*02:01* allele ($r^2$ = 0.99). j) Effect sizes of *HLA-A\*02:01* on EBV DNA load in blood (green points; UKB, n = 453,770 European ancestry participants; AoU, n = 201,181 European ancestry participants) and saliva (gray points; AoU, n = 33,168 European ancestry participants; SPARK, n = 9,209 European ancestry participants). For saliva, effect sizes are shown for both a binary presence/absence phenotype and a quantitative phenotype (inverse-normal transformed abundance, restricted to individuals whose saliva WGS had detectable EBV DNA). Error bars, 95% CIs. k) Prevalence of HHV-7 DNA positivity by genetically determined blood type in UKB (n = 456,988 European ancestry participants). Blood type was determined as in ref. 83. l) Prevalence of HHV-7 DNA positivity by genotype combinations of common blood type alleles (*O*, *B*, *A1*, and *A2*) in UKB. m) Effect size on EBV DNA load for the *CPVL* Y168H missense variant and *CPVL* loss-of-function variants (aggregated across such variants in a burden test; n = 453,770). n) Effect size on EBV DNA load for the *TNFRSF13B* C104R and A181E missense variants and *TNFRSF13B* loss-of-function variants. *TNFRSF13B* coding variants (C104R, $p = 2.3 \times 10^{-14}$; A181E, $p = 3.5 \times 10^{-12}$) lead to impaired ligand binding and downstream signaling[97].

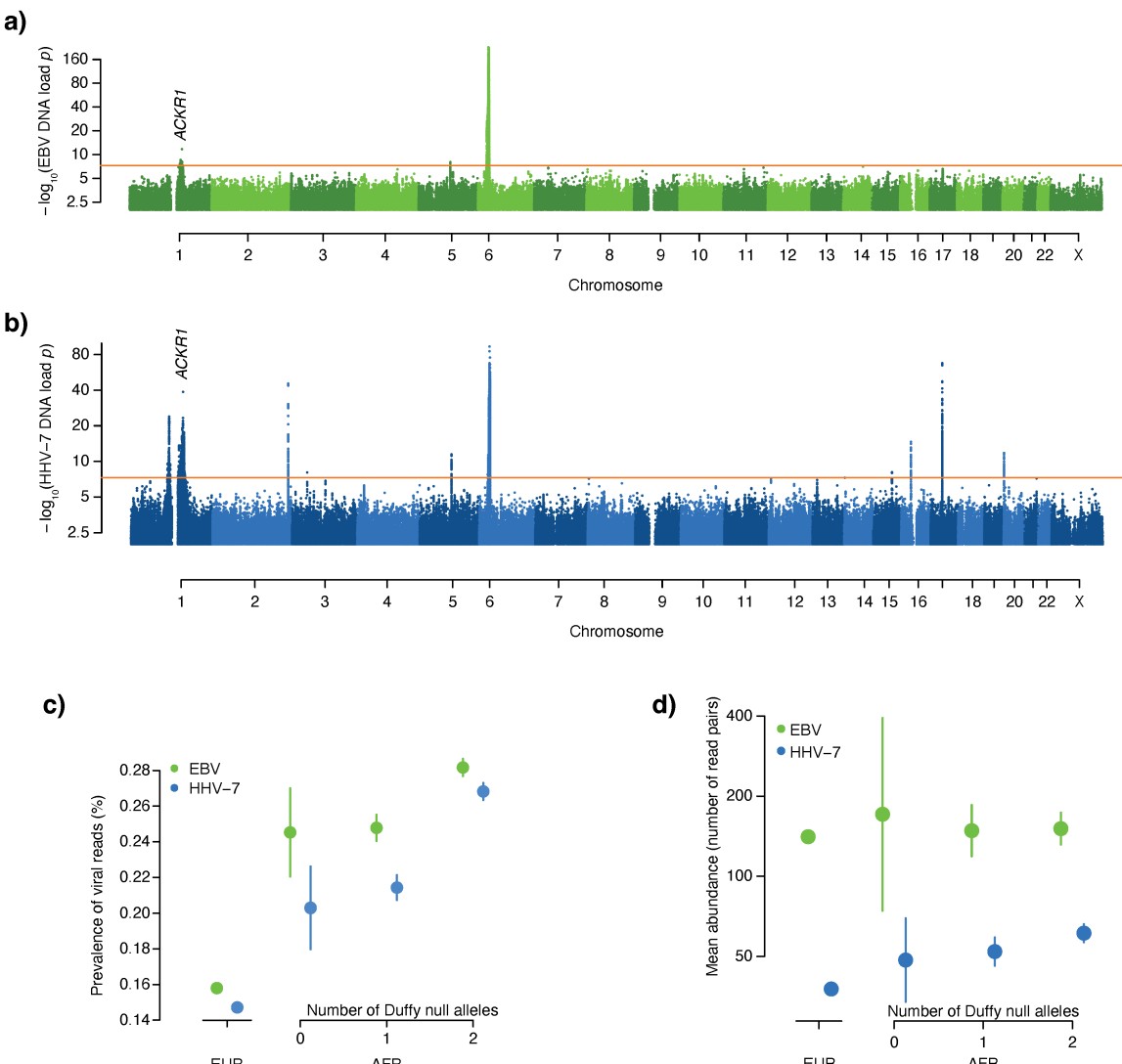

**Extended Data Fig. 10 | Genome-wide association analyses of viral DNA load in blood samples from African ancestry participants.** a) Genome-wide associations of human genetic variants with EBV DNA positivity in blood-derived WGS from African-ancestry AoU participants (n = 77,573). b) Genome-wide associations of human genetic variants with HHV-7 DNA positivity in blood-derived WGS from African-ancestry AoU participants (n = 77,573). c) Prevalence of detectable EBV and HHV-7 positivity (reads mapping to each viral genome) by genetic ancestry and Duffy-null genotype in AoU blood WGS (n = 251,209 participants). The AFR sub-cohort analyzed here consisted of individuals with 70–90% African ancestry (n = 51,052 participants) to minimize correlation of Duffy-null genotype with admixture proportion. Error bars, 95% CIs. d) Number of reads mapping to the EBV (n = 18,723 participants) and HHV-7 (n = 33,823 participants) genomes by genetic ancestry and Duffy-null genotype in AoU saliva WGS. The AFR sub-cohort analyzed here consisted of individuals with 70–90% African ancestry (n = 51,052 participants) to minimize correlation of Duffy-null genotype with admixture proportion. Error bars, 95% CIs.

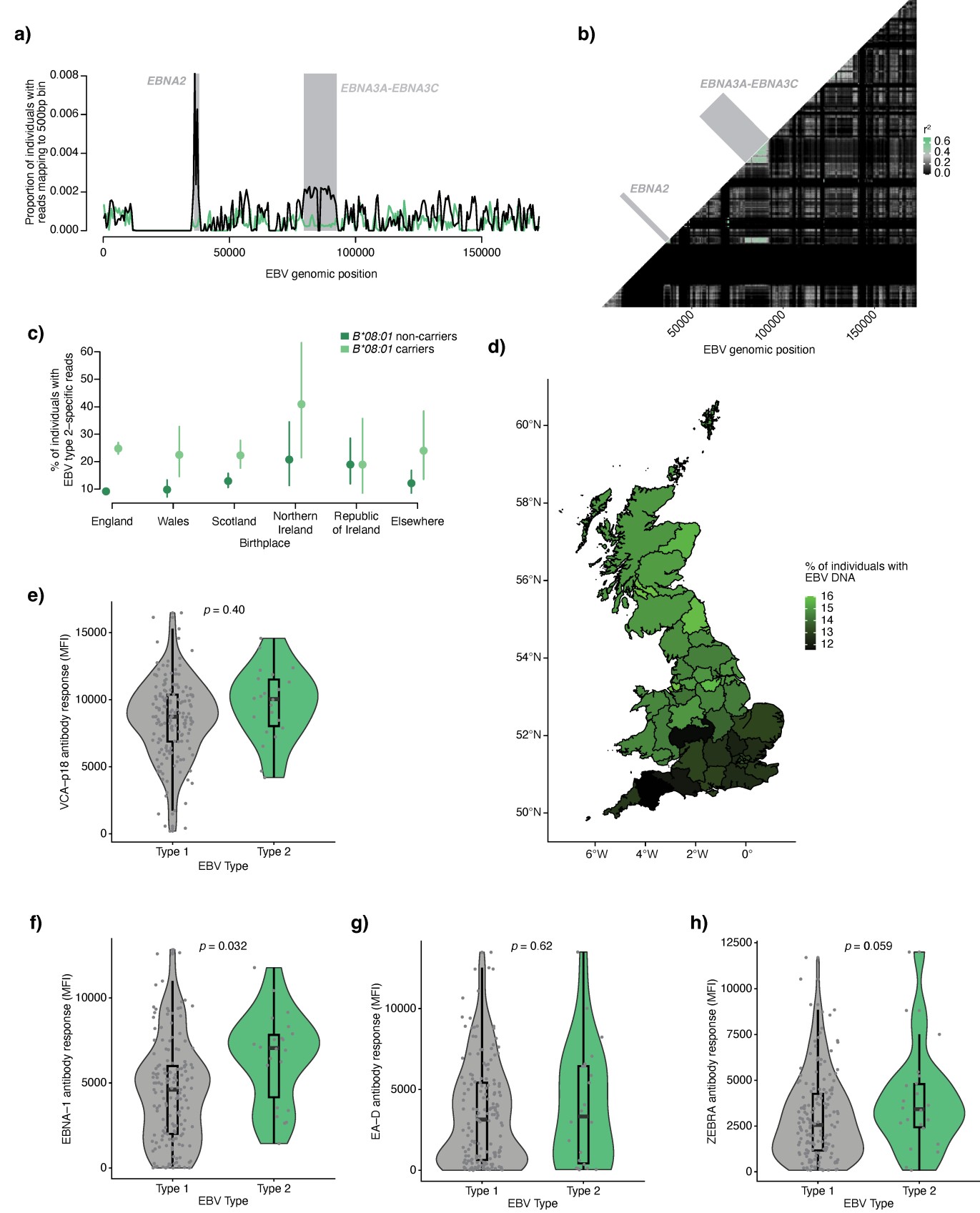

**Extended Data Fig. 11** | See next page for caption.

**Extended Data Fig. 11 | Validation of type-informative EBV genome regions and assessment of quantitative antibody responses for EBV type 1 and type 2.** a) Frequencies of observing WGS reads mapping to each 500 bp segment of the EBV type 1 (black line) and type 2 (green line) reference genomes in blood WGS samples from UKB (n = 490,401). Highlighted in gray are two regions used to distinguish EBV types, both containing genes with known type-specific sequences, *EBNA2* and *EBNA3A–EBNA3C*. Read alignments in these two regions show consistent ratios of type 1-aligned to type 2-aligned reads, whereas other EBV genomic regions (which are not type-informative) do not. b) Pairwise correlations in the abundances of DNA sequences mapping to 500 bp segments of the EBV type 2 genome in saliva WGS samples from SPARK with >1 type-specific alignment (n = 4,757). For each sample, read counts for each 500 bp bin were first normalized by the maximum number of reads in any 500 bp region across the EBV type 1 and type 2 genomes. Regions used for downstream EBV type analyses (containing *EBNA2* and *EBNA3A–EBNA3C*) are highlighted. The correlation heatmap shows that reads aligning to the EBV type 2 genome in these two regions tended to occur in the same individuals. c) Proportion of UKB participants positive for EBV type 2 reads (as a fraction of those positive for either type 1 or type 2) by country of birth within the United Kingdom and Republic of Ireland, stratified by *HLA-B*08:01* allele carrier status (n = 456,923). The effect of *B*08:01* on EBV type 1 versus type 2 (decreasing the proportion of individuals with EBV type 1 relative to type 2) appeared to be present across all geographic regions of the United Kingdom, which vary in rates of EBV type 2 prevalence (Fig. 5c). Error bars, 95% CIs. d) Proportion of UKB participants positive for EBV DNA by region of birth within the United Kingdom. Map source: Office for National Statistics, licensed under the Open Government Licence v.3.0 and Contains OS data. Crown copyright and database right 2017–2026. e) Distribution of quantitative antibody response to VCA-p18 for EBV seropositive UKB participants with type-informative EBV DNA sequences exclusively from the EBV type 1 (black) or type 2 (green) genomes (n = 335 with available serology measurements). P-values are from linear regression including age, age-squared, sex, assessment center, and 20 ancestry PCs as covariates. MFI, median fluorescence intensity. f) Analogous to c, for quantitative antibody response to EBNA-1. g) Analogous to c, for quantitative antibody response to EA-D. h) Analogous to c, for quantitative antibody response to ZEBRA.

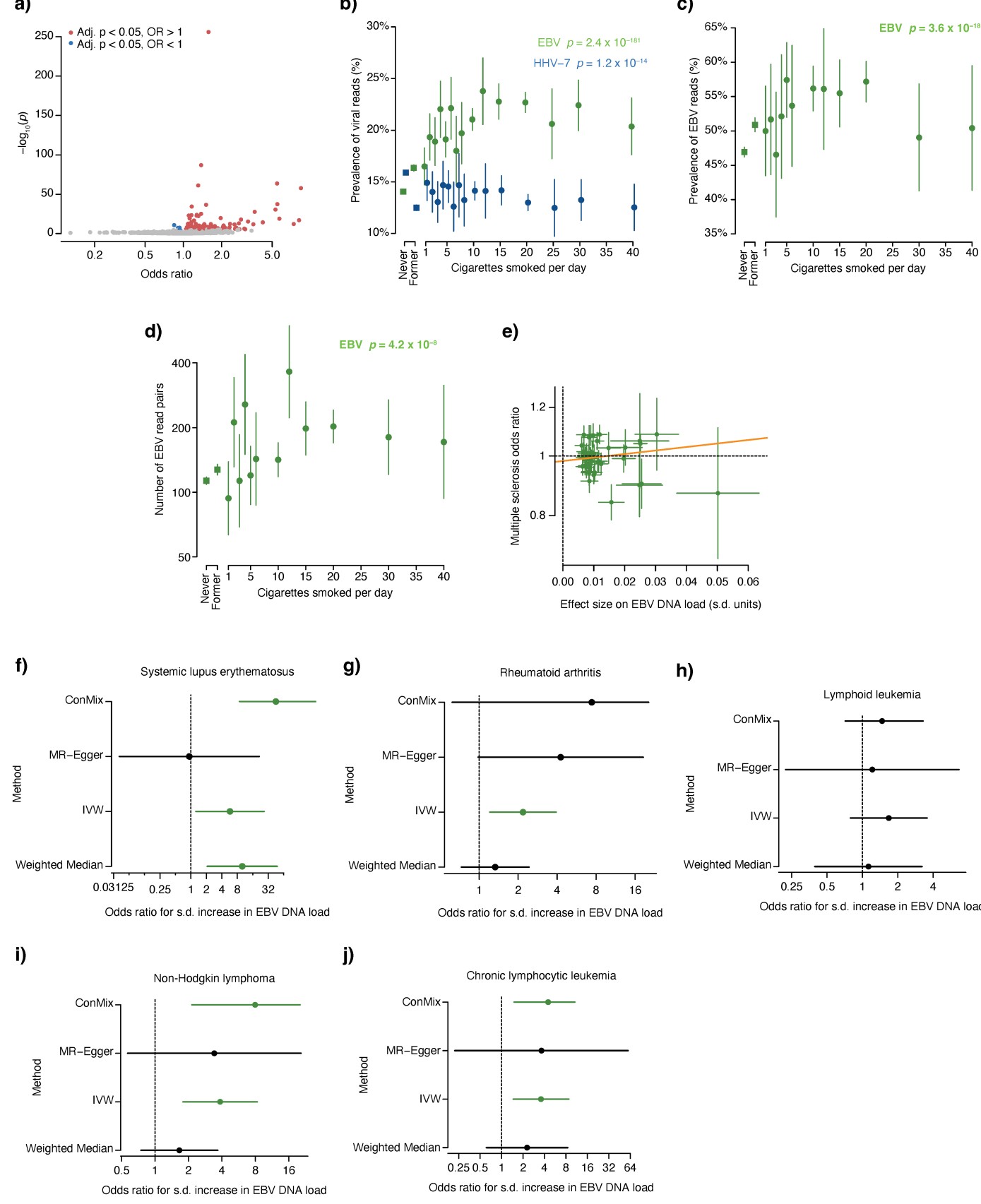

**Extended Data Fig. 12** | See next page for caption.

**Extended Data Fig. 12 | Additional data regarding associations of viral DNA load with clinical phenotypes and smoking.** a) Associations of clinical conditions with detectable viral positivity (presence of viral DNA sequences in blood WGS) in European-ancestry UKB participants (n = 453,770). Significant associations (Bonferroni-corrected $p < 0.05$) are colored by direction of effect. b) Prevalence of detectable EBV positivity (in green) and HHV-7 positivity (in blue) by cigarettes smoked per day in AoU blood samples (n = 365,918). Prevalences among individuals who have never smoked or formerly smoked are indicated by the square points on the left. Error bars, 95% CIs. c) Prevalence of detectable EBV positivity in AoU saliva WGS (n = 48,899) by cigarettes smoked per day. Prevalences among individuals who have never smoked or formerly smoked are indicated by the square points to the left. Error bars, 95% CIs. d) Analogous to b, for mean EBV abundance (geometric mean among samples with at least one read pair). e) Effect sizes for 44 common genetic variants (at distinct non-MHC loci)—used as instrument variables in Mendelian randomization—for EBV DNA load (x-axis; n = 638,825) and risk for multiple sclerosis (y-axis; meta-analysis of 7,907 cases and 1,474,810 controls). The line shown is from the MR-Egger test; the y-intercept is not significantly different from OR = 1, indicating low pleiotropy of the genetic instruments. f) Estimates of the causal effect of EBV DNA load (exposure) on risk for systemic lupus erythematosus (outcome; meta-analysis of 3,491 cases and 1,361,247 controls) using different Mendelian randomization approaches (y-axis) with 44 non-MHC loci as instrument variables. An estimate is plotted in green if its 95% CI does not overlap an odds ratio of 1 (no effect). Error bars, 95% CIs. However, as these results replicated inconsistently across MR approaches with different assumptions[98], we do not conclude causality, but rather that viral DNA load is more likely to affect the risk of systemic lupus erythematosus compared to MS. g) Estimates of the causal effect of EBV DNA load on risk for rheumatoid arthritis (meta-analysis of 35,875 cases and 1,296,933 controls). Error bars, 95% CIs. h) Estimates of the causal effect of EBV DNA load on risk for lymphoid leukemia (meta-analysis of 5,966 cases and 1,248,154 controls). Error bars, 95% CIs. i) Estimates of the causal effect of EBV DNA load on risk for non-Hodgkin lymphoma (meta-analysis of 13,198 cases and 1,273,062 controls). Error bars, 95% CIs. j) Estimates of the causal effect of EBV DNA load on risk for chronic lymphocytic leukemia (meta-analysis of 4,467 cases and 1,248,724 controls). Error bars, 95% CIs.

# Reporting Summary

## Statistics

For all statistical analyses, confirm that the following items are present in the figure legend, table legend, main text, or Methods section.

| n/a | Confirmed | |
|---|---|---|
| ☐ | ☒ | The exact sample size (*n*) for each experimental group/condition, given as a discrete number and unit of measurement |
| ☐ | ☒ | A statement on whether measurements were taken from distinct samples or whether the same sample was measured repeatedly |
| ☐ | ☒ | The statistical test(s) used AND whether they are one- or two-sided *Only common tests should be described solely by name; describe more complex techniques in the Methods section.* |
| ☐ | ☒ | A description of all covariates tested |
| ☐ | ☒ | A description of any assumptions or corrections, such as tests of normality and adjustment for multiple comparisons |
| ☐ | ☒ | A full description of the statistical parameters including central tendency (e.g. means) or other basic estimates (e.g. regression coefficient) AND variation (e.g. standard deviation) or associated estimates of uncertainty (e.g. confidence intervals) |
| ☐ | ☒ | For null hypothesis testing, the test statistic (e.g. *F*, *t*, *r*) with confidence intervals, effect sizes, degrees of freedom and *P* value noted *Give P values as exact values whenever suitable.* |
| ☒ | ☐ | For Bayesian analysis, information on the choice of priors and Markov chain Monte Carlo settings |
| ☒ | ☐ | For hierarchical and complex designs, identification of the appropriate level for tests and full reporting of outcomes |
| ☐ | ☒ | Estimates of effect sizes (e.g. Cohen's *d*, Pearson's *r*), indicating how they were calculated |

*Our web collection on statistics for biologists contains articles on many of the points above.*

## Software and code

Policy information about availability of computer code

| Data collection | No software was used for data collection. |
|---|---|
| Data analysis | The following publicly available software resources were used: bwa (v0.7.18, https://bio-bwa.sourceforge.net/), mosdepth (v0.3.9, https://github.com/brentp/mosdepth), bcftools (v1.14, http://www.htslib.org/), samtools (v1.15.1, http://www.htslib.org/), plink (v1.90b6.26 and v2.00a3.7, https://www.cog-genomics.org/plink/), BEAGLE (v5.4, https://faculty.washington.edu/browning/beagle/beagle.html), BOLT-LMM (v2.5, https://alkesgroup.broadinstitute.org/BOLT-LMM/), METAL (v2020-05-05, https://genome.sph.umich.edu/wiki/METAL), qqman R package (v0.1.8, https://cran.r-project.org/web/packages/qqman/index.html), sf R package (v1.0-20, https://cran.r-project.org/web/packages/sf/index.html), logistf R package (v1.26.1, https://cran.r-project.org/web/packages/logistf/index.html), and MendelianRandomization R package (v0.10.0, https://cran.r-project.org/web/packages/MendelianRandomization/index.html). |

For manuscripts utilizing custom algorithms or software that are central to the research but not yet described in published literature, software must be made available to editors and reviewers. We strongly encourage code deposition in a community repository (e.g. GitHub). See the Nature Portfolio guidelines for submitting code & software for further information.

## Data

Policy information about availability of data

All manuscripts must include a data availability statement. This statement should provide the following information, where applicable:

- Accession codes, unique identifiers, or web links for publicly available datasets
- A description of any restrictions on data availability
- For clinical datasets or third party data, please ensure that the statement adheres to our policy

The following data resources are available by application: UK Biobank (http://www.ukbiobank.ac.uk/), All of Us Research Program (https://allofus.nih.gov/), SFARI SPARK (https://www.sfari.org/resource/spark/), MVP-Finngen-UKBB meta-analysis summary statistics (https://mvp-ukbb.finngen.fi/), and T1DGC HLA imputation panel (https://repository.niddk.nih.gov/study/173). The following data resources are publicly available: human reference genome build GRCh38 (https://ftp.1000genomes.ebi.ac.uk/vol1/ftp/technical/reference/GRCh38_reference_genome/), TOPMed-r2 imputation panel variant list (https://imputation.biodatacatalyst.nhlbi.nih.gov/), gnomAD v4.1 variant call set (https://gnomad.broadinstitute.org/), LD score resources https://alkesgroup.broadinstitute.org/LDSCORE/), NCBI Virus for reference sequences (https://www.ncbi.nlm.nih.gov/labs/virus/vssi/), PrimateAI-3D scores (https://primateai3d.basespace.illumina.com/), GENCODE 39 definitions (https://www.gencodegenes.org/), and GTEx expression and splice quantitative trait associations (https://gtexportal.org/home/). Full viral DNA load GWAS summary statistics are available from the GWAS Catalog under accessions GCST90809801 to GCST90809829.

## Research involving human participants, their data, or biological material

Policy information about studies with human participants or human data. See also policy information about sex, gender (identity/presentation), and sexual orientation and race, ethnicity and racism.

| | |
|---|---|
| Reporting on sex and gender | For UK Biobank (recorded 222,094 males, 263,132 females), sex was acquired from NHS central registry at recruitment, but in some cases updated by the participant. For All of Us (recorded 161,253 males, 252,074 females), sex was genetically determined by copy number and presence of X and Y chromosomes. For SFARI SPARK (recorded 7284 males, 5235 females), sex was self-reported. Sex was used as a covariate in most analyses (ex. human genetic associations with viral load), where no values directly pertaining to sex are reported for these analyses. Values pertaining to sex were reported for analyses that demonstrated higher viral load in men (Fig. 2a-d, EDF 3c-e). |
| Reporting on race, ethnicity, or other socially relevant groupings | UK Biobank, using the top 20 ancestry principal components, a subset of individuals that fell within a Euclidean distance (centered at the mean values of each PC for individuals who self-identified as "white") capturing 99% of individuals who self-identified as "white" were used for genotype and phenotype associations. For All of Us, analyses were restricted to individuals with previously released genetically-predicted European ancestry for genotype and phenotype associations. For SPARK, using the top 10 ancestry principal components, a subset of individuals that fell within a Euclidean distance (centered at the mean values of each PC for individuals who self-identified as "white") capturing 90% of individuals who self-identified as "white" were used for genotype and phenotype associations. For all analyses, ancestry principal components were included as covariates in genetic associations. For associations of viral DNA with ancestry, these genetically-predicted ancestries were used as independent variables for association with viral DNA. |
| Population characteristics | UK Biobank is a cohort of approximately 500,000 individuals across the United Kingdom between 40 and 69 years of age at time of recruitment (Sudlow et al. 2015 PLOS Medicine). For viral DNA associations in the UK Biobank cohort, age, age squared, sex, genotype array, assessment center, and top 20 genetic ancestry PCs were used as covariates. All of Us is a cohort of 414,817 individuals with WGS available (at time of analysis) across the United States older than 18 years of age at time of recruitment (The All of Us Research Program Investigators 2019 N Engl J Med). For viral DNA associations in the All of Us cohort, age, age squared, sex, and the top 16 genetic ancestry principal components (from ancestry_preds.tsv) were used as covariates. SFARI SPARK is a cohort of approximately 160,000 families with at least one child with autism spectrum disorder, where 12,519 individuals (at time of analysis) have WGS from saliva available (SPARK Consortium 2018 Neuron). Children range in age from 0-50 years of age (mean 9 years) and parents from 19-90 years of age (mean 41 years). For viral DNA associations in the SPARK SFARI cohort, sequencing batch, age, age squared, square root of age, sex, percent of mapped reads, and the top 10 genetic ancestry principal components were used as covariates. |
| Recruitment | Individuals and biosamples were not obtained for this study and their recruitment is as described in prior publications (cited in current work). |
| Ethics oversight | Individuals and biosamples were not obtained for this study and local IRBs at each institution approved the collections and patient-consent materials, as described in the earlier papers on these cohorts (cited in current work). Datasets were used as approved for research plans as stated in applications to each, including UK Biobank Resource application #40709. |

Note that full information on the approval of the study protocol must also be provided in the manuscript.

# Field-specific reporting

Please select the one below that is the best fit for your research. If you are not sure, read the appropriate sections before making your selection.

☒ Life sciences    ☐ Behavioural & social sciences    ☐ Ecological, evolutionary & environmental sciences

For a reference copy of the document with all sections, see nature.com/documents/nr-reporting-summary-flat.pdf

# Life sciences study design

All studies must disclose on these points even when the disclosure is negative.

| | |
|---|---|
| Sample size | For associations in the UK Biobank WGS data set, individuals were excluded based on the following criteria: removed for not having European genetic ancestry; removed for not having available TOPMed-imputed genotypes (including for chromosome X); and removed for having withdrawn, leaving 453,770 available individuals for genetic association analyses. For some associations using linear regression rather than linear mixed models that would account for relatedness (ex. HLA local associations), additional samples were removed to drop one relative within pairs of close relatives with kinship coefficient > 0.0884.<br><br>For associations in the All of Us cohort, 414,817 samples with available WGS were first separated into those where sample DNA was either blood (n=365,918) or saliva-derived (n=48,899). Individuals were then excluded for not having European genetic ancestry, leaving 201,181 blood-derived samples and 33,164 saliva-derived samples. For analyses of HHV-6B viral load, individuals with endogenous HHV-6A or HHV-6B were excluded, leaving 199,133 blood-derived samples and 32,826 saliva samples. For saliva-derived samples, some associations (such as those associating quantity rather than presence of viral DNA) were then performed on the subset of samples with at least 1 viral DNA sequence, leaving a subset of samples as noted in the text (ex. 16,282 samples for EBV).<br><br>For associations in the SFARI SPARK cohort, among 12,519 samples with available WGS data some samples were removed for not having European genetic ancestry, leaving 9,209 individuals. For analyses of HHV-6B viral load, individuals with endogenous HHV-6A or HHV-6B were excluded, leaving 9,081 individuals. Some associations associating quantity rather than presence of viral DNA were then performed on the subset of samples with at least 1 viral DNA sequence, leaving a reduced sample number as noted in the text.<br><br>In all cases, no sample-size calculation was done to predetermine sample size and the maximum number of available samples were used. For viral DNA genetic associations, we expected that although the phenotype was sparser with less dynamic range than previous work measuring viral load by PCR (ex. HIV viral load), that the greatly increased sample size (2-3 orders of magnitude) would allow for observing expected associations in the MHC region of the human genome. Previous work has also suggested EBV viral load to be heritable. |
| Data exclusions | Established QC metrics were used to exclude some samples, genotypes, or sequencing data for analysis as described in previously published studies (cited in the current work). In brief: For UK Biobank WGS, the sequencing of 914 participants failed due to either insufficient or poor-quality DNA. For UK Biobank array genotypes used for imputation, ~3% of samples had insufficient DNA to use as input for genotyping and 968 samples with genotypes were excluded for high heterozygosity or >5% missing rate. For All of Us WGS, samples were excluded for low quality or insufficient DNA for library production and 987 samples were at least eight median absolute deviations from median residual in at least one type of variant metric (ex. number of SNPs, number of indels). For SPARK WGS, no samples were excluded by QC criteria. Samples from individuals in UK Biobank, All of Us, and SFARI SPARK that requested to be withdrawn at the time of analysis were excluded. |
| Replication | For each viral association observed in UK Biobank (blood) and SPARK (saliva), we were generally successful in replicating those tested in the subset of All of Us samples with DNA derived from the same source (blood or saliva) such that each had a single attempt at replication. This included associations with human genetics (besides one EBV-associated locus that had discordant direction of effect in blood samples from All of Us relative to UK Biobank), smoking, age, sex, time of day, and month of the year. We did not attempt replication of binary ICD-10 code phenotypes in All of Us.<br><br>Additionally, we attempted to verify all genetic associations to EBV and HHV-7 load in UK Biobank by generating additional phenotypes from reads aligning to the left and right halves of each genome and evaluating whether associations to all virally-aligned reads replicated in these phenotypes. All loci but 1 successfully replicated (the same locus which failed replication in All of Us), such that each association had a single attempt at replication with this approach. |
| Randomization | For UK Biobank, samples were collected in batches at different assessment centers at locations across the United Kingdom and these were encoded as indicator covariates in phenotype-genotype associations. For All of Us, samples were collected in batches at different sequencing centers, and these were encoded as indicator variables in phenotype-genotype associations. For SFARI SPARK, samples were collected in different batches of sequencing cohorts (WGS1 through WGS5) and these were encoded as indicator covariates in phenotype-genotype associations. No further randomization was done in each cohort as all samples were used for each analysis. |
| Blinding | For data collection, blinding was not relevant as data was collected by other research groups (cited in the current work) and re-analyzed in this work. For all computational analyses blinding was always done, as samples were listed with a randomized ID where association of measured genotype with viral DNA phenotype was only done at the point of final statistical analysis. |

# Reporting for specific materials, systems and methods

We require information from authors about some types of materials, experimental systems and methods used in many studies. Here, indicate whether each material, system or method listed is relevant to your study. If you are not sure if a list item applies to your research, read the appropriate section before selecting a response.

nature portfolio | reporting summary

## Materials & experimental systems

| n/a | Involved in the study |
|-----|----------------------|
| ☒ | Antibodies |
| ☒ | Eukaryotic cell lines |
| ☒ | Palaeontology and archaeology |
| ☒ | Animals and other organisms |
| ☒ | Clinical data |
| ☒ | Dual use research of concern |
| ☒ | Plants |

## Methods

| n/a | Involved in the study |
|-----|----------------------|
| ☒ | ChIP-seq |
| ☒ | Flow cytometry |
| ☒ | MRI-based neuroimaging |

## Plants

| | |
|---|---|
| Seed stocks | *Report on the source of all seed stocks or other plant material used. If applicable, state the seed stock centre and catalogue number. If plant specimens were collected from the field, describe the collection location, date and sampling procedures.* |
| Novel plant genotypes | *Describe the methods by which all novel plant genotypes were produced. This includes those generated by transgenic approaches, gene editing, chemical/radiation-based mutagenesis and hybridization. For transgenic lines, describe the transformation method, the number of independent lines analyzed and the generation upon which experiments were performed. For gene-edited lines, describe the editor used, the endogenous sequence targeted for editing, the targeting guide RNA sequence (if applicable) and how the editor was applied.* |
| Authentication | *Describe any authentication procedures for each seed stock used or novel genotype generated. Describe any experiments used to assess the effect of a mutation and, where applicable, how potential secondary effects (e.g. second site T-DNA insertions, mosiacism, off-target gene editing) were examined.* |

