## [Peer Review File · Nature]

The DNA virome varies with human genes and environments

Corresponding Author: Dr Nolan Kamitaki

Version 0:

Reviewer comments:

Referee #1

(Remarks to the Author)

Kamitaki et al. describe an in-depth analysis of viral load for 31 DNA viruses that persistently or recurrently infect humans. Viral genomic reads were identified in the previously generated, high-coverage WGS data from >900k participants in UK Biobank, All of Us, and SPARK (samples from blood or saliva origin). The authors report associations between viral load of some viruses and multiple factors, including age, sex, ancestry, as well as circadian and seasonal effects. They also report dozens of host genetic associations dominated by the HLA region. Finally, they describe EBV-host type-specific effects (e.g., HLA-B*08:01 and EBV type 1 vs 2), and use Mendelian randomization to demonstrate that EBV DNA load is causal for Hodgkin lymphoma risk but not for multiple sclerosis.

This is a very solid study, performed using state-of-the-art bioinformatic and statistical approaches, which uses large-scale existing data to address an important biomedical question. The findings are important and can be useful to suggest functional follow-up to better understand the intricate relationships between genetic background, viral infections and the long-term development of human diseases. The paper is well written.

Major comments:

1. The novelty of the main results is compromised by a preprint posted in July 2025 by Schmidt et al. (<https://doi.org/10.1101/2025.07.19.25331823>), which describes a very similar study, even if mostly focused on EBV: largely overlapping study population and data sources, and – reassuringly – mostly comparable main results. This work should obviously be cited. Also, some discrepancies merit to be discussed, e.g., the conflicting results of the Mendelian randomization analyses (no causal effect of EBV load on Hodgkin lymphoma in Schmidt et al., as opposed to a clear causal role for EBV load suggested by all four MR methods here)
2. Validity of viral load estimation: because many samples only have a very small number of sequencing reads mapping to viral genomes (especially in blood), how confident can we be that these reflect true viral signals rather than noise? Is it possible to estimate the false positive rate (e.g. via simulation) or quantify this uncertainty? Also, the authors use a threshold of 5 for alignment mapping quality, which is low. As this is a Phred scale metric, a mapping quality of 5 implies a 31.6% chance of alignment error, which is quite high. Value of 20 or 30 are most commonly used. Would the main findings survive a stricter filtering (e.g., mapping quality >20)?
3. EBV type 1 vs type 2 classification: the identification of EBV type relies on reads in EBNA2/EBNA3 regions that have mostly been sequenced at very low depth. It would be good to quantify the risk of misclassification based on this shallow data, or at least to show that the HLA-B*08:01 result persists when restricting to individuals infected with EBV that have been typed with high confidence.
4. Temporal claims (circadian/seasonal): the reported circadian and seasonal effects on viral load are interesting, but visit times might be scheduled based on individual availability, and may thus correlate with age, sex, study site, or factors such as fasting status and recent illness. To help rule out these potential sources of bias, it would be helpful to conduct sensitivity analyses, for example by 1) including clinic or day-of-week in the models, 2) excluding samples from outlier clinics, 3) replicating circadian patterns within individual centers and seasons.
5. Ancestry associations: the differences between ancestries are large, which might also reflect complex socio-economic impacts. Beyond principal components, it would be good also adjust for socioeconomic indices and smoking behavior, and present ancestry-stratified effect estimates.
6. Cross-tissue consistency of genetic associations: for the main associations identified, it would be helpful to get a clear

summary of the results across cohorts and tissues (e.g. UKB blood compared with AoU blood, and SPARK saliva compared with AoU saliva) along with concordance of effect directions. For tissue-specific associations, a discussion of whether these reflect true biological differences or technical variation is needed.

Minor comments:

1. The terminology around "viral load" should be clarified. The authors use "prevalence" (any reads) and "abundance" (read counts), which is good. However, "viral load" is used for both. I suggest using "load" for the quantitative measure only, and "positivity" for the binary variable.
2. For the reported effect of age, sex and ancestry on viral load, statistical testing should be performed.
3. Which multiple-testing correction approach was used for the phenome-wide analyses of the 31 viruses?
4. The association between smoking and decreased prevalence of HHV-7 is intriguing: any potential explanation (or previous reports)?
5. What's the rationale for choosing "4(Q3-Q1)+Q3+5" as the coverage threshold?
6. In the Methods, indicate whether duplicate reads were removed before counting viral pairs.
7. The last sentence of the Discussion is wrong: HIV, HCV and HTLV-1 are not DNA viruses.

Referee #2

(Remarks to the Author)

I co-reviewed this manuscript with one of the reviewers who provided the listed reports.

Referee #3

(Remarks to the Author)

Review of Nature submission "Genes and environment profoundly affect the human virome"

This is an ambitious and interesting study pairing large-scale human genetic data with viral genomic signatures across multiple major cohorts. The authors analyse whole-genome sequencing (WGS) datasets from the UK Biobank (n = 490,401), SPARK (n = 12,519), and All of Us (n = 414,817) for the presence of 29 DNA viruses. I think that this represents the largest dataset of its kind to date, encompassing genomic data from over 900,000 individuals - I haven't seen a study on this scale before.

The analysis of temporal, age-associated, and sex-defined variation in viral load is particularly compelling. The demonstration of genetic associations with viral load further supports the idea of host genetic control of viral persistence, echoing previous findings for HCV and HIV.

I have a few comments and questions about the content and about the analysis used.

1. The observed link between EBV DNA positivity and multiple sclerosis (MS) aligns with recent findings showing that exposure, rather than viral load, drives MS risk, presumably strengthening the case for an immune-mediated phenomenon. In contrast, viral load was strongly associated with Hodgkin lymphoma. How many lymphoma cases were included, and could the measured viral load reflect treatment effects (e.g. chemotherapy-related immune suppression)? I think important to consider this.

2. Terminology – "Human virome"

The term "human virome" is somewhat overstated, as RNA viruses were not assessed (except possibly HIV provirus in the supplementary data). The title would be more accurate as "Genes and environment profoundly affect the human DNA virome."

3. The study searched for 29 DNA viruses, including HHV-1–8, HAdV-A, HPV-18, and the proviral retroviruses HTLV-1/2 and HIV-1/2. One reference sequence per species was mostly used. I'm not sure why this approach was used rather than de novo assembly followed by mapping to the relevant viruses - as there might be other virus families present in the samples. e.g. I am curious about adeno-associated viruses (AAVs), PARV4, and TTV midi viruses?

4. The number and percentage of viral reads were not presented (ideally as reads per million sequenced). Validation of read counts against known viral loads would strengthen the quantitative interpretation.

5. Read mapping strategy. Using single paired-read hits to infer virus presence is acceptable, but mapping such reads to a single viral genome can be misleading. DNA viruses contain conserved regions that cross-map between related species. For example, reads from PARV4 and B19 could map to both genomes. I think a better approach would be to take the raw reads, run blastx (e.g. diamond blast x or a kmer approach) and then do mapping to the closest reference genomes with a clear e-score for blastn confirmation of positive hits. In summary - a more comprehensive approach would include human DNA depletion followed by (a) k-mer-based or diamond blastx screening of all reads with blastn confirmation and (b) de novo assembly (e.g. IDBA or SPAdes) to identify the closest reference sequence prior to re-mapping. (c) ML phylogenetic analysis. Mapping to a single reference risks missing divergent viral variants.

6. The total numbers and prevalence of each detected virus across the dataset should be clearly reported as well as being

plotted in a graph. i.e. the raw data are missing and should be provided as a dataset (along with clear metrics, e.g. reads per million)

7. Sequencing methods. The methods for generating viral reads require clarification—specifically, the wet-lab preparation protocol, any depletion or enrichment steps, and the sequencing platform(s) used. If human-targeted enrichment was employed, viral reads may be underestimated, and this limitation should be acknowledged as a potential bias

8. Clinical metadata. The reliability and coding of clinical conditions should be clarified. How systematically were diagnoses collected across cohorts—through medical records (e.g. “Do you have MS? Y/N”) or self-reporting? Were definitions and data completeness consistent between datasets? I'd have liked to see absolute numbers here also e.g. how many cases of MS were actually present in the dataset and how many of Hodgkins.

9. Was medication information available? Again, if not, it will be important to acknowledge that immune suppressive drugs in particular could cause reactivation - and might bias interpretation of the data.

10. Coinfections Were any associations or interactions observed between coinfections?

11. The statement that the study “revealed a causal association between EBV viral load and increased risk of Hodgkin lymphoma” would be better phrased as “confirmed,” as causality cannot be inferred directly from this analysis.

12. Prevalence in figures S1c and S1d. How many individuals were virus-positive? The total cohort size is 441,026, but the numerator for each virus is unclear. Figure S1c (prevalence of viral reads vs. lymphocyte %) interestingly shows increasing lymphocyte percentage with higher HHV-7 and EBV reads, but not with TTV or HHV-6B. Figure S1d (reticulocyte count vs. parvovirus reads) appears to show a non-linear decline in reticulocyte count; clarification would be helpful.

13. Coverage plots (S2). For each virus shown and within each patient, how many individuals had detectable reads, and what was the range?

14. HHV-6B and telomere loss. This is a naive question but could the decline in HHV-6B reads by age reflect loss of telomeric integration?

15. Figure S3 (and elsewhere) Figure legends should state the absolute numbers of individuals positive for each virus.

16. Seasonal variation The temporal distribution of EBV reads by month (S3f) was interesting but not statistically significant. Were participants all from temperate regions with distinct seasonal variation?

17. Diurnal and seasonal lymphocyte variation The variation in lymphocyte percentage by month and hour (S3g, S3h) is intriguing and I think perhaps has been previously observed in other settings e.g. during exercise. The datasets used and absolute numbers should be specified in the legends.

18 Ethnicity (S4) For each figure, the number of samples included and the number testing positive should be provided.

19. Nomenclature ICTV virus nomenclature should be included (minor point).

20. Supplementary Figure S8 Absolute numbers appear to be missing and should be added.

21. Supplementary Figure S9 The association with class II HLA alleles is interesting—does this relate to receptor differences or to CD4⁺ T-cell-mediated immunity?

22. Supplementary Figure S10d The geographic variation in EBV DNA prevalence is notable. Does this also correlate with MS prevalence or with viral load by longitude?

Summary and recommendation

This is a highly valuable and potentially landmark study linking host genetics with viral persistence across population-scale cohorts. The findings on host–virus interactions and genetic associations are compelling and will likely influence future studies of viral latency and immune control. However, the manuscript would benefit from additional methodological detail, clearer reporting of quantitative data (including prevalence and read depth), and improved consistency in figure legends. With these clarifications, the paper would represent a major contribution to our understanding of the human DNA virome.

Referee #4

(Remarks to the Author)

This study presents a comprehensive survey of the viral load of 31 DNA viruses in human blood and saliva, leveraging whole-genome sequencing data from the UK Biobank (n=490,401), SPARK (n=12,519), and All of Us (n=414,817) cohorts. The authors observed marked variations in viral load across age, time of day, and season, with higher loads generally found in men than in women for most viruses. They also identified genetic variants associated with viral load for certain viruses. Furthermore, the study provided evidence for a strong causal effect of Epstein–Barr virus (EBV) viral load on the risk of Hodgkin lymphoma. Overall, the principal strength of this work lies in its use of an unprecedented sample size to elucidate

the influence of human genetic and environmental factors on a range of human DNA viruses.

We note that the methodologies applied for virus detection and the subsequent genetic (GWAS and mendelian randomization analyses) analyses, while appropriate, are not novel. The results are largely reported in a descriptive fashion, with limited interpretation of the underlying biological or clinical implications. Additionally, we have concerns regarding several conceptual, statistical, and methodological aspects of the work, which are outlined in the following points:

Major concerns:

1. The scope of the viral analysis substantially limits the impact and interpretation of this study.

The profiling of only 31 DNA viruses does not fully represent the "human virome", a term that encompasses a far greater diversity of viruses, including the entirely uninvestigated RNA viruses. This oversight, coupled with the well-established composition of the human virome (which includes both DNA and RNA viruses), makes the current title misleading. Readers are likely to anticipate findings relevant to established niches like the gut virome. Therefore, we strongly recommend revising the title to accurately reflect the analytical focus of the study. Furthermore, the manuscript must provide a clear and thorough rationale for the selection of these specific 31 DNA viruses.

2. Statistical support of significance needs to be provided.

The presentation of the results is largely descriptive and lacks essential P values. Key sections, such as "The DNA virome in 856,319 blood and 61,418 saliva samples" and "Viral load is shaped by age, sex, circadian and seasonal effects," report findings without providing associated P-values. To allow for a rigorous assessment of the claimed associations, it is imperative that P-values or adjusted P-values (e.g., for multiple testing) be provided for all statistical comparisons throughout the manuscript.

3. Circadian analysis

The observation of circadian and seasonal fluctuations in viral load, as suggested by the figures, is intriguing. However, the statistical methodology used to support these claims could be more robust. We recommend applying established methods specifically designed for circadian rhythm analysis, such as those implemented in the Kronos package (PMID: 39504963) or other referenced tools (e.g., PMID: 38569545), to validate these findings with greater rigor.

4. Single or meta-analyses of the cohorts

The strategy for handling the multiple cohorts is inconsistent and unclear. The authors should justify the reasoning behind sometimes combining cohorts (lines 205-206) and sometimes analyzing them separately. Providing a justification for these choices is essential. Furthermore, we note that a formal meta-analysis was not performed across all cohorts, please give the reasons?

5. Validation of results in vivo or vitro models is desirable.

The authors reported that EBV viral load exhibited a strong causal effect on increased risk of Hodgkin lymphoma with Mendelian randomization analysis. We encourage the authors to discuss the possibility of validating this association in in vitro or in vivo models in future work, as such data would greatly strengthen the conclusion.

6. Heterogeneity between cohorts in data processing

In the method, the samples from different cohorts were sequenced in various coverage, and then the sequencing data were processed by different bioinformatic tools. How did the author handle these variations in the current manuscript?

Minor concerns:

- Line 172: it is better to give the exact time of morning and evening.
- The parameters of the bioinformatic tools that the authors used should be given, e.g., line 1121-1122.
- Line 385: provide the full name of MR analyses
- Line 443: the human virome need to be rephrased as only 31 viruses were analyzed.
- The number of samples need to be given in figures, such figure 2 for each covariate and time points, as well as the P values as mentioned above.

Referee #5

(Remarks to the Author)

I co-reviewed this manuscript with one of the reviewers who provided the listed reports.

Version 1:

Reviewer comments:

Referee #1

(Remarks to the Author)

I have reviewed the revised manuscript and commend the authors for their thorough and thoughtful response to the previous round of comments. The vast majority of my earlier concerns have been addressed. However, a few minor issues remain, detailed below:

1. Regarding the two overlapping studies: they are now cited, the discrepancy in MR results is discussed, including the key point about excluding the MHC region. Still, the paper would benefit from explicitly positioning what is new relative to these studies beyond a general "broader scope" statement: several novelty points emphasized in the response letter are not transparently laid out in the manuscript itself (e.g., inclusion of saliva, breadth across 31 viruses, and any EBV-specific findings the authors consider uniquely supported here).

2. Validity of viral load estimation: in their response, the authors convincingly show that the viral signal is not dominated by noise. The detailed numerical robustness results (fractions of affected individuals per virus and near-identical association p-values) are not clearly documented in the manuscript, so these points currently read as asserted rather than directly supported in the paper. Clearer definitions and documentation in the text would be helpful, e.g., define what "false positive rate" means here (per-read vs per-individual positivity), clarify the logic behind interpreting EBV seronegative-with-reads cases, and point explicitly to where the MAPQ sensitivity outcomes are shown (or briefly summarize the key numbers in Supplement).

3. Impact of circadian and seasonal effects on viral load: The response is satisfactory but the key elements have not been incorporated into the revised manuscript (or supplement): (i) day-of-week sensitivity analysis, and (ii) assessment-center "outlier" screen - even if the conclusion is that no exclusions were warranted. I would appreciate to see 1) a brief description of these sensitivity analyses in the Methods or Supplement and 2) a sentence acknowledging that confounding by unmeasured visit-related factors cannot be fully excluded, even with the above sensitivity analyses.

Referee #2

(Remarks to the Author)

I co-reviewed this manuscript with one of the reviewers who provided the listed reports.

Referee #3

(Remarks to the Author)

I am satisfied that the authors have answered the majority of my questions very comprehensively and those of the other reviewers.

There is only one comment that I would make which is that my suggestion for further analysis of viral reads was misunderstood. The suggestion was not to look at de novo assembly as per the author rebuttal, but in fact to look at raw fastq reads using diamond blastx (suggested because it is highly sensitive, substantially faster and less computationally intensive than blastn) or kmer analysis of those reads followed by mapping. I agree fully that de novo assembly is associated with a far lower sensitivity than mapping and that de novo assembly followed by blast would not add to the analysis.

However, I am persuaded that this request would be expensive and time-consuming, and while the authors may have missed some viruses in their analysis (which can be >30% divergent between genotypes), it does not reduce the validity of the findings as they currently stand. This issue is more problematic with viral genomes than mammalian (or indeed bacterial) genomes due to far higher levels of genetic diversity.

There were also some gaps in viruses selected for mapping - eg AAVs.

I would be satisfied if the authors were to acknowledge these limitations.

Referee #4

(Remarks to the Author)

Thank you for the opportunity to review the revised version of this manuscript. I have thoroughly examined the manuscript again and am pleased to find that all of my previous concerns have been fully addressed. I have no further major comments on the study. Below are only some minor formatting suggestions for your consideration.

In figures where sample numbers are presented as a range (e.g., the age bins in Figure 2), could the exact sample size (n) be indicated in the figure labels? For instance, labeling as "0-1 year (n=xxx)" would enhance clarity.

Line 90: high-coverage need to be specified

Line 159-160: figure 2b shows prevalence not abundance, right?

Line 178: P value should be added in figure 2a and figure 2b for each time point.

Referee #5

(Remarks to the Author)

I co-reviewed this manuscript with one of the reviewers who provided the listed reports.

Referee #1 (Remarks to the Author):

Kamitaki et al. describe an in-depth analysis of viral load for 31 DNA viruses that persistently or recurrently infect humans. Viral genomic reads were identified in the previously generated, high-coverage WGS data from >900k participants in UK Biobank, All of Us, and SPARK (samples from blood or saliva origin). The authors report associations between viral load of some viruses and multiple factors, including age, sex, ancestry, as well as circadian and seasonal effects. They also report dozens of host genetic associations dominated by the HLA region. Finally, they describe EBV-host type-specific effects (e.g., HLA-B*08:01 and EBV type 1 vs 2), and use Mendelian randomization to demonstrate that EBV DNA load is causal for Hodgkin lymphoma risk but not for multiple sclerosis.

This is a very solid study, performed using state-of-the-art bioinformatic and statistical approaches, which uses large-scale existing data to address an important biomedical question. The findings are important and can be useful to suggest functional follow-up to better understand the intricate relationships between genetic background, viral infections and the long-term development of human diseases. The paper is well written.

We appreciate the thoughtful review of our manuscript and the helpful suggestions below, which led us to undertake several further analyses that strengthened results.

Major comments:

1. The novelty of the main results is compromised by a preprint posted in July 2025 by Schmidt et al. (<https://doi.org/10.1101/2025.07.19.25331823>), which describes a very similar study, even if mostly focused on EBV: largely overlapping study population and data sources, and – reassuringly – mostly comparable main results. This work should obviously be cited. Also, some discrepancies merit to be discussed, e.g., the conflicting results of the Mendelian randomization analyses (no causal effect of EBV load on Hodgkin lymphoma in Schmidt et al., as opposed to a clear causal role for EBV load suggested by all four MR methods here)

We have added citations to this preprint (now ref. 73) and another similar preprint on EBV that was posted the same week (Nyeo et al., <https://doi.org/10.1101/2025.07.18.665549>; ref. 74). While the scope of our manuscript is much broader (studying 31 viruses in both blood and saliva), we agree that our results on EBV are reassuringly broadly consistent—though even for EBV, several key results in our manuscript are novel relative to the two preprints:

- We identify nearly twice as many GWAS loci for EBV (45 loci in our study vs. 28 or 21 loci in the other two studies).
- We observe a strikingly bimodal distribution of EBV read counts in saliva samples, suggesting frequent bursts of active (lytic) viral replication and shedding.
- We identify multiple effects of *HLA* alleles on EBV load, including an effect of *HLA-A*02:01* specific to EBV load in blood (and not saliva) and a host-virus genetic interaction of *HLA-B*08:01* with EBV subtype.

Regarding the conflicting results of our Mendelian randomization analyses compared to Schmidt et al., we agree that this discrepancy is important to discuss. We believe that the discrepancy probably arises from the following two factors (now explained in Discussion, p. 11):

1. Limited statistical power in Schmidt et al.'s MR analysis of EBV and Hodgkin lymphoma. The Hodgkin lymphoma GWAS summary statistics that Schmidt et al. analyzed (from the MVP cohort) were based on 975 HL cases and 449,793 controls, whereas we analyzed GWAS summary statistics from a 2.6-fold larger meta-analysis (across FinnGen+MVP+UKB) of 2,529 HL cases and 1,159,394 controls. Additionally, our MR analysis benefited from many more genetic instruments available from our higher-powered GWAS of EBV load.
2. Inclusion versus exclusion of the MHC region from the set of instrument variables. We excluded the MHC region from our MR analyses, reasoning that restricting to 44 non-MHC instrumental variables would reduce the potential for bias (as noted in Methods; p. 51): *HLA* alleles generate the largest effects on EBV load but also have strong linkage disequilibrium with other variants in the MHC that could separately influence risk of Hodgkin lymphoma, violating the assumptions of MR.

2. Validity of viral load estimation: because many samples only have a very small number of sequencing reads mapping to viral genomes (especially in blood), how confident can we be that these reflect true viral signals rather than noise? Is it possible to estimate the false positive rate (e.g. via simulation) or quantify this uncertainty? Also, the authors use a threshold of 5 for alignment mapping quality, which is low. As this is a Phred scale metric, a mapping quality of 5 implies a 31.6% chance of alignment error, which is quite high. Value of 20 or 30 are most commonly used. Would the main findings survive a stricter filtering (e.g., mapping quality >20)?

We appreciate these pertinent questions and undertook several additional analyses to investigate them.

First, regarding the false positive rate among reads attributed to viral genomes, we identified three independent lines of evidence indicating that the false positive rate is low:

1. For EBV, we directly estimated the false positive rate using high-quality EBV serostatus data available in UKB. Among seronegative individuals, only 4 out of 494 individuals (0.8%) had EBV reads, and moreover, 2 of these individuals had three read pairs that mapped to EBV, suggesting that these reads were correctly attributed to EBV and that these 2 individuals had false negative serostatus. In contrast, 1,323 out of 8,857 seropositive individuals (14.9%) had EBV reads. This suggests that the false positive rate was well-controlled at roughly ~3% ($= 2 / 492 / (1,323 / 8,857)$). We have now noted this information in Results (p. 3).

- For each virus, reads attributed to the virus covered the viral reference genome reasonably evenly (aggregating all reads observed to align to the virus across all individuals; **Supplementary Fig. 2**). Most reads attributed to a virus did not arise from any localized spike in read coverage, which would be a diagnostic sign of false positive reads arising from mapping artifacts.
- For HHV-7 and HHV-6B, which we observed in $\geq 90\%$ of saliva DNA samples from adults (for HHV-7) or children age 5–9 (for HHV-6B) (**Fig. 2a**, copied below-right), viral reads were observed in only a small minority of the youngest SPARK participants (infants) (below-left, now included as **Supplementary Fig. 3a**). This observation is consistent with infants being unlikely to have been infected and indicates that the false positive rate was well-controlled. Similarly, MCPyV, which was observed in $\sim 10\text{--}20\%$ of young children in SPARK, was not observed in any infants under 5 months of age.

Regarding the sensitivity of the analyses to the mapping quality threshold of 5, we repeated the analyses of UK Biobank using a more stringent filter as suggested ($\text{MAPQ} \geq 20$) and found that this affected very few reads (and very few individuals). For example, for the 6 most prevalent viruses, the affected individuals (i.e., those who had alignments with MAPQ between 5 and 19) represented only a modest fraction of the number of individuals who we had determined to be viral DNA-positive based on the $\text{MAPQ} \geq 5$ threshold:

- EBV: 0.5% (367/72157)
- HHV-7: 1.8% (783/42987)
- HHV-6B: 1.8% (106/5675)
- TTV (TUS01): 2.5% (143/5582)

- TTV (HD14a): 0.3% (22/6264)
- TTV (VT416): 7.9% (171/1990)

The relatively larger (but still small) fractions of affected individuals for two anelloviruses (TUS01 and VT416) probably reflect the sequence diversity of anelloviruses, which results in more low-quality alignments due to mutations relative to the viral reference genomes.

We further verified that the choice of mapping quality threshold had a negligible impact on genetic associations. For example, the top HLA associations for each of the above viruses (not including HHV-6B, which lacks an association in the MHC region) were nearly unchanged upon switching from $\text{MAPQ} \geq 20$ to $\text{MAPQ} \geq 5$:

- EBV: $p = 2.6 \times 10^{-670} \rightarrow 9.2 \times 10^{-669}$
- HHV-7: $p = 6.9 \times 10^{-603} \rightarrow 1.5 \times 10^{-608}$
- TTV (TUS01): $p = 2.0 \times 10^{-80} \rightarrow 9.4 \times 10^{-81}$
- TTV (HD14a): $p = 5.8 \times 10^{-251} \rightarrow 4.0 \times 10^{-252}$
- TTV (VT416): $p = 1.1 \times 10^{-16} \rightarrow 1.1 \times 10^{-14}$

Overall, these data indicate that the choice of mapping quality filter ultimately had little impact on the results, perhaps because requiring both a read and its mate to align to the same viral reference genome ended up dropping spurious reads with low mapping quality. We have summarized this discussion in Methods (p. 43).

3. EBV type 1 vs type 2 classification: the identification of EBV type relies on reads in EBNA2/EBNA3 regions that have mostly been sequenced at very low depth. It would be good to quantify the risk of misclassification based on this shallow data, or at least to show that the HLA-B*08:01 result persists when restricting to individuals infected with EBV that have been typed with high confidence.

This was a great suggestion that led us to identify a source of false positives in our original determination of EBV type-specific reads. Correcting this issue substantially strengthened the *HLA-B*08:01* result ($p = 9.4 \times 10^{-48}$ before; $p = 7.4 \times 10^{-70}$ now).

Briefly, the issue was that in the EBV type 1 vs. type 2 analysis, we realigned all unmapped or chrEBV-mapped reads to the EBV type 1 and type 2 reference genomes (providing both reference genomes simultaneously to *bwa*), after which we filtered to reads for which both the read and its mate had been aligned to one of the two reference genomes. However, whereas our main analysis pipeline included a filter on insert size (100–1000bp), the type 1 vs. type 2 pipeline did not include such a filter, as insert size was undefined in situations in which a read mapped to EBV type 1 and its mate to EBV type 2 or vice versa (e.g., if only one read in the pair fell within a type-informative region, and its non-type-informative mate was mapped arbitrarily to type 1 or type 2). This resulted in an elevated false positive rate in the original type 1 vs. type 2 analysis, such that ~25% of type-1-positive calls and ~7% of type-2-positive calls had been made in people who had been determined to be EBV DNA-negative in the main analysis.

To correct this issue, we restricted the EBV type 1 vs. type 2 analysis to a subset of individuals who had been typed with high confidence, as suggested. Specifically, we restricted to individuals with two or more reads aligned to the same EBV reference genome (i.e., either 2+ type-1 reads or 2+ type-2 reads, usually corresponding to a read and its mate). This reduced the number of type-1 calls from 16,291 to 10,032 and the number of type-2 calls from 1,842 to 1,396, with the excluded individuals largely comprising those who had been EBV DNA-negative in the main analysis. In this high-confidence subset, we observed that the association of EBV type with *HLA-B*08:01* strengthened from $p = 9.4 \times 10^{-48}$ to $p = 7.4 \times 10^{-70}$.

We additionally evaluated the risk of misclassification, as suggested. To do so, we examined the distribution of type 2 vs. type 1 reads in SPARK saliva samples, making use of the fact that saliva samples frequently have hundreds of EBV reads that should typically come from only one EBV type (depending on whether the individual is infected with a type 1 or type 2 EBV strain). This analysis showed that as expected, nearly all samples had read counts heavily skewed to either type 1 or type 2, indicating a low rate of misclassification of reads:

The low but nonzero misclassification rate is reasonable given that not all differences between the type 1 and type 2 reference genomes are expected to be truly type-specific, even within the type-informative genomic regions we selected.

We have now updated all type 1 vs. type 2 analyses presented (in Results (p. 8), **Fig. 5**, and **Supplementary Fig. 10**) to restrict to individuals with 2+ type-specific reads, and we have updated Methods accordingly.

4. Temporal claims (circadian/seasonal): the reported circadian and seasonal effects on viral load are interesting, but visit times might be scheduled based on individual

availability, and may thus correlate with age, sex, study site, or factors such as fasting status and recent illness. To help rule out these potential sources of bias, it would be helpful to conduct sensitivity analyses, for example by 1) including clinic or day-of-week in the models, 2) excluding samples from outlier clinics, 3) replicating circadian patterns within individual centers and seasons.

This is a good point, and we appreciate these suggestions of sensitivity analyses to rule out potential sources of bias in the temporal analyses. We have undertaken all of these suggestions and found that the observed temporal patterns are robust to conditioning on each of these variables. Specifically:

- 1) Day-of-week (encoded as indicator variables) had no effect on either EBV or HHV-7 ($p > 0.05$) when included as a covariate in both the circadian and seasonal analyses. We had already included assessment center as a categorical covariate in these analyses.
- 2) No assessment center had an outlier rate of EBV or HHV-7 positivity: EBV prevalence ranged from -0.080 to $+0.042$ s.d. and HHV-7 prevalence ranged from -0.047 to $+0.11$ s.d., so we concluded that no center needed to be excluded.
- 3) Circadian and seasonal patterns were broadly consistent within each assessment center (different colored lines in the new **Supplementary Fig. 3f-i**):

Likewise, circadian patterns were consistent within each season, now shown in **Supplementary Fig. 3j,k**:

5. Ancestry associations: the differences between ancestries are large, which might also reflect complex socio-economic impacts. Beyond principal components, it would be good also adjust for socioeconomic indices and smoking behavior, and present ancestry-stratified effect estimates.

This is a good point as well. We evaluated the effect of additionally adjusting for Townsend deprivation index, years of education, and smoking status, and we observed a negligible impact on the ancestry differences: differences between ancestries that ranged from 0.09 to 0.32 s.d. changed by at most 0.01 s.d. upon including these additional covariates. Detailed data are provided in the new **Supplementary Note 1**.

The relatively minor contribution of socio-economic effects suggests that the large differences in viral loads between ancestries may be primarily genetic. To more deeply evaluate this possibility, we conducted GWAS of EBV and HHV-7 load in African ancestry AoU participants ($n=77,573$), finding that the variant that causes the Duffy-null phenotype appears to explain $\sim 1/4$ of the African- vs. European-ancestry difference in EBV load and $\sim 1/2$ of the difference in HHV-7 load. We have now reported these results in Results (p. 7), **Fig. 4a**, **Supplementary Fig. 9a-c**, and **Supplementary Note 5**. We restricted all genetic association analyses throughout the manuscript to single genetic ancestries.

6. Cross-tissue consistency of genetic associations: for the main associations identified, it would be helpful to get a clear summary of the results across cohorts and tissues (e.g. UKB blood compared with AoU blood, and SPARK saliva compared with AoU saliva) along with concordance of effect directions. For tissue-specific associations, a discussion of whether these reflect true biological differences or technical variation is needed.

We agree that a clear summary of cross-cohort consistency and cross-tissue consistency (or heterogeneity) of genetic effects was needed. We have now performed these comparisons for EBV, HHV-6B, and HHV-7, which were the most commonly observed sources of viral DNA and were the three viruses for which we identified genetic effects on load in both blood and saliva. These comparisons were helpful, and the cross-cohort consistency that we observed across UKB and AoU motivated us to harmonize our GWAS analyses to consistently perform meta-analysis of blood DNA load across the UKB and AoU European-ancestry cohorts (see response to Referee #4, Comment 4).

In detail: to evaluate cross-cohort consistency of genetic effects, we compared effect size estimates in AoU vs. UKB for index variants identified by our GWAS of EBV, HHV-6B, and HHV-7 load in blood. This analysis showed strong cross-cohort concordance of effect sizes, now illustrated in **Supplementary Fig. 7c-e**:

We considered performing an analogous comparison of effect sizes between SPARK saliva and AoU saliva but determined that this would not be informative due to the smaller sample size of SPARK ($n=12,519$), its age heterogeneity relative to AoU (with SPARK being comprised mostly of children), and the limited number of saliva GWAS hits available for comparison.

Next, to evaluate cross-tissue consistency of genetic effects, we compared the effect sizes of index variants (from both the blood GWAS and saliva GWAS) in the UKB+AoU blood meta-analysis versus the AoU saliva association analysis. This showed moderate concordance for EBV but general discordance for HHV-6B and HHV-7, with several genetic effects clearly specific to blood or saliva (**Supplementary Fig. 7f-h**):

These differences in saliva vs. blood effect sizes appear likely to reflect true biological differences: the differences are well outside the range of statistical noise, and technical variation (e.g., in sample processing or DNA sequencing for saliva vs. blood samples) would be expected to increase or decrease signal-to-noise to the same extent across different genetic effects, which would not explain the observed heterogeneity in saliva vs. blood effect size ratios (beta/beta). We have noted this in Results (p. 6) and the legend of Supplementary Fig. 7f-h.

These comparisons of index variants made us realize that a more comprehensive evaluation of effect concordance genome-wide could be accomplished by measuring genetic correlation between summary statistics for pairs of viral phenotypes. We have now undertaken this analysis as well, now reported in **Supplementary Fig. 7i**:

This genetic correlation analysis revealed a few additional interesting properties and confirmed the patterns observed above by comparing effect sizes of index variants:

- 1) For each virus, the genetic correlation between UKB blood and AoU blood was high. Genetic correlations between AoU saliva and SPARK saliva were noisy, as expected.
- 2) EBV had a statistically significant, moderate genetic correlation between blood and saliva, whereas HHV-7 did not have significant genetic correlation between blood and saliva, and likewise for HHV-6B.
- 3) HHV-7 and HHV-6B had significant between-virus genetic correlation in blood and likewise in saliva, suggesting that HHV-7 and HHV-6B have shared biology in a compartment-specific manner.
- 4) The anelloviruses had strong genetic correlation with each other, consistent with their sharing of GWAS loci (**Fig. 3c** and **Supplementary Fig. 5b,c**).

Minor comments:

1. The terminology around “viral load” should be clarified. The authors use "prevalence" (any reads) and "abundance" (read counts), which is good. However, "viral load" is used for both. I suggest using "load" for the quantitative measure only, and "positivity" for the binary variable.

We appreciate this suggestion and likewise wish to make the exposition as clear and precise as possible. However, after considering the possible options for terminology, we worry that using “positivity” for the binary variable would raise a risk of misinterpretation, since readers might associate binary “positivity” with seropositivity. Here, positivity for the observation of any reads is more an indication of higher-than-usual viral load: e.g., for EBV, ~95% of UKB participants are infected based on seropositivity, but only 15% were positive for EBV reads, reflecting a subset of the population with higher EBV load (along with a contribution of sampling noise). As such, we believe “load” is actually a better conceptual description of the binary variable as well. For this reason, we have refrained from using “positivity” to describe the binary variable in the main text.

In light of these considerations, we decided to retain the use of “viral load” as a broader term that we use to describe either measurement in the manuscript. To avoid ambiguity, we have carefully checked through the manuscript to ensure that the relevant variable (quantitative or binary) is always clear from the context in which “viral load” is used (e.g., based on a referenced figure panel, in which the variable being analyzed is clearly defined).

2. For the reported effect of age, sex and ancestry on viral load, statistical testing should be performed.

We have added p-values for each of these points.

3. Which multiple-testing correction approach was used for the phenome-wide analyses of the 31 viruses?

We have clarified in Methods (p. 50) that we applied Bonferroni correction to account for running phenome-wide analyses on 8 viruses (6 acquired viruses: EBV, HHV-6B, HHV-7, and three anelloviruses; and the 2 germline integrated HHV-6 (eHHV-6A and eHHV-6B)). Other viruses had low prevalence so we did not include them in the phenome-wide analyses to reduce multiple testing burden. We note there was a labeling error in **Supplementary Fig. 11a** that has been corrected from “FDR < 0.01” to “Adj. p < 0.05”; the colored points were those passing Bonferroni correction as indicated correctly in the legend, not FDR.

4. The association between smoking and decreased prevalence of HHV-7 is intriguing: any potential explanation (or previous reports)?

This result also intrigued us but we are not confident in any particular explanation. We did not find prior reports of this result; this was unsurprising given that while the association was clear in the large UKB cohort ($p=9.6 \times 10^{-22}$ in a sample size of $n \sim 450k$; **Fig. 6b**), it would not have been visible in the smaller sample sizes in which HHV-7 has previously been studied.

5. What’s the rationale for choosing “ $4(Q3-Q1)+Q3+5$ ” as the coverage threshold?

This expression corresponds to a lenient “Tukey fence,” which is an outlier removal boundary defined based on adding a multiple of the interquartile range ($Q3-Q1$) to the third quartile ($Q3$). Here, we used a lenient Tukey fence to retain regions with modest elevation of coverage (as such regions may still have a majority of alignments derived from the viral genome), and we added a constant offset of 5 to handle situations in which the first and third quartiles have the same value. We have now explained this in Methods (p. 44).

6. In the Methods, indicate whether duplicate reads were removed before counting viral pairs.

Thank you for pointing out that this was unclear. Duplicate reads were indeed filtered (by mosdepth, which we used for read-counting). We have clarified this in Methods (p. 43).

7. The last sentence of the Discussion is wrong: HIV, HCV and HTLV-1 are not DNA viruses.

Thank you for catching this mistake. We have amended the statement as follows:

“... certain DNA viruses such as hepatitis B and retroviruses such as HIV and HTLV.”

Additionally, we have amended the manuscript text to no longer refer to the 31 viruses considered in our analyses as “31 DNA viruses.”

Referee #3 (Remarks to the Author):

Review of Nature submission “Genes and environment profoundly affect the human virome”

This is an ambitious and interesting study pairing large-scale human genetic data with viral genomic signatures across multiple major cohorts. The authors analyse whole-genome sequencing (WGS) datasets from the UK Biobank (n = 490,401), SPARK (n = 12,519), and All of Us (n = 414,817) for the presence of 29 DNA viruses. I think that this represents the largest dataset of its kind to date, encompassing genomic data from over 900,000 individuals - I haven't seen a study on this scale before.

The analysis of temporal, age-associated, and sex-defined variation in viral load is particularly compelling. The demonstration of genetic associations with viral load further supports the idea of host genetic control of viral persistence, echoing previous findings for HCV and HIV.

I have a few comments and questions about the content and about the analysis used.

We appreciate the thorough review of our work and many helpful suggestions below, which have strengthened the revised manuscript.

1. The observed link between EBV DNA positivity and multiple sclerosis (MS) aligns with recent findings showing that exposure, rather than viral load, drives MS risk, presumably strengthening the case for an immune-mediated phenomenon. In contrast, viral load was strongly associated with Hodgkin lymphoma. How many lymphoma cases were included, and could the measured viral load reflect treatment effects (e.g. chemotherapy-related immune suppression)? I think important to consider this.

Thank you for raising this question. The association with Hodgkin lymphoma presented in **Fig. 6g** is actually with **incident** cases of HL, so treatment effects should not be an issue. We have now further verified that the association is robust to the time interval used to define “incident” (>0, 1, or 2 years after DNA acquisition):

This is concordant with the results from Mendelian randomization (supporting a causal effect of EBV load on risk of Hodgkin lymphoma but not MS), which should be robust from treatment effects as Mendelian randomization relies on inherited alleles present from birth.

We have included the number of incident Hodgkin lymphoma cases (n=174) in the legend of **Fig. 6g**.

2. Terminology – “Human virome”

The term “human virome” is somewhat overstated, as RNA viruses were not assessed (except possibly HIV provirus in the supplementary data). The title would be more accurate as “Genes and environment profoundly affect the human DNA virome.”

This is a good point. We have modified the title as suggested.

3. The study searched for 29 DNA viruses, including HHV-1–8, HAdV-A, HPV-18, and the proviral retroviruses HTLV-1/2 and HIV-1/2. One reference sequence per species was mostly used. I'm not sure why this approach was used rather than de novo assembly followed by mapping to the relevant viruses - as there might be other virus families present in the samples. e.g. I am curious about adeno-associated viruses (AAVs), PARV4, and TTV midi viruses?

Thank you for pointing out that the rationale for the read-mapping-based approach used to identify and quantify viral DNA was not explained. We took this approach following Moustafa et al. 2017 (ref. 13), which evaluated both possible approaches—(i) de novo assembly followed by protein-based search; and (ii) nucleotide-based search against viral reference genomes—and found that blood-derived whole-genome sequencing data rarely contained enough viral reads for de novo assembly to be feasible. Specifically, among n=8,240 blood WGS samples analyzed by Moustafa et al., viruses were detected only by read mapping in 3,342 samples, by both read mapping and de novo assembly in 137 samples, and only by de novo assembly in 13 samples. Among the 13 examples detected only by de novo assembly, 2 represented contaminant plasmid sequences and the remaining 11 corresponded to highly variable anelloviruses. We have now explained this in Methods (p. 43).

We chose to use a limited reference panel of 31 viral genomes, mostly one per species, because the focus of our work was on identifying effects of human genetics, age, sex, and exposures such as smoking on DNA load of common viruses. As such, for each common virus of interest, we ultimately needed to obtain a single quantification of the load of that virus (collapsed across strains of the virus) for GWAS and phenome-wide association analyses. We did not have power to perform GWAS or PheWAS of rarer viruses, so we did not attempt to comprehensively characterize viral diversity observed infrequently in human populations. AAVs and PARV4 in particular were not observed at appreciable rates in Moustafa et al.'s analysis of 8,240 samples (on which we primarily based our selection of the 31 reference genomes). We have now clarified this in Methods as well (p. 43).

Regarding TTV midi viruses (Gammatorquevirus), these were actually represented in the reference panel (by the SAV-1 reference genome); likewise, TTV mini viruses (Betatorquevirus) were represented by the TTmV (TLMV-CLC062) reference genome. We apologize that this was not clear in the initial submission and have now clarified this by including ICTV nomenclature in **Supplementary Table 1** as suggested below (Comment 19).

Finally, thank you for pointing out that our reference panel of 31 viruses included a few RNA retroviruses. We have edited the manuscript text to no longer refer to the 31 viruses as “31 DNA viruses.”

4. The number and percentage of viral reads were not presented (ideally as reads per million sequenced). Validation of read counts against known viral loads would strengthen the quantitative interpretation.

These are good points. We have now reported this information in four new **Supplementary Tables 2-5** (one per cohort: UKB blood, AoU blood, SPARK saliva, AoU saliva), providing the following statistics computed among individuals DNA-positive for each virus (i.e., having >0 read alignments that passed filtering): mean, minimum, and maximum number of read pairs; and mean reads per million sequenced reads. When reporting means, we used geometric means to better represent wide ranges in scale.

Validating read counts against known viral loads was a sensible suggestion, but after exploring prior literature, we found that previous quantifications of viral load in healthy individuals were performed in small sample sizes using qPCR techniques that were too noisy for absolute quantifications to provide an informative comparison. For example, we found two studies that attempted to quantify abundances of multiple herpesviruses in 100–200 blood donors (Hudnall et al. 2008, PMID: 18422852; Geraudie et al. 2012, PMID: 22133730). These data initially seemed promising, but the estimated viral loads spanned several orders of magnitude across samples. Moreover, each study identified one participant with endogenous HHV-6B, but the first study estimated viral load to be 11 copies per cell and the second study 2.23–3.21 copies per cell. These estimates point to high error in qPCR-based quantification of viral load: in our WGS-based analysis, eHHV-6A and eHHV-6B are almost always present in 1 copy per cell:

In light of these issues with qPCR data, we decided not to pursue further comparison with previous viral load quantifications (beyond the short discussion we had included in Methods of the number of EBV reads expected to be observed in WGS data given previous data on EBV; Methods, p. 44).

5. Read mapping strategy. Using single paired-read hits to infer virus presence is acceptable, but mapping such reads to a single viral genome can be misleading. DNA viruses contain conserved regions that cross-map between related species. For example, reads from PARV4 and B19 could map to both genomes. I think a better approach would be to take the raw reads, run blastx (e.g. diamond blast x or a kmer approach) and then do mapping to the closest reference genomes with a clear e-score for blastn confirmation of positive hits. In summary - a more comprehensive approach would include human DNA depletion followed by (a) k-mer-based or diamond blastx screening of all reads with blastn confirmation and (b) de novo assembly (e.g. IDBA or SPAdes) to identify the closest reference sequence prior to re-mapping. (c) ML phylogenetic analysis. Mapping to a single reference risks missing divergent viral variants.

While we agree that using a single reference risks missing divergent viral variants, our main goal was to capture the majority of virally-derived reads in a computationally efficient way (scalable to hundreds of thousands of samples) for downstream GWAS and phenome-wide association analyses (PheWAS). For these purposes, missing a small fraction of divergent reads or incurring a small amount of mis-mapping in regions conserved across species was acceptable. Two properties of the viral abundance measurements produced by our pipeline suggest that it accomplished the goal of capturing most of the relevant reads:

- 1) The overall results were very concordant with the more comprehensive pipeline in Moustafa et al. using BLASTN to align against a much larger set of reference sequences. We observed similar prevalence and abundances for each of the detected viruses.
- 2) The results were largely insensitive to the choice of mapping quality threshold (<2% difference between using MAPQ \geq 5 versus MAPQ \geq 20 for most viruses; see Referee #1, Comment 2). This suggests that few reads are sufficiently diverged at the nucleotide level to create alignment ambiguity that could be resolved by amino acid-based alignment.

We appreciate the suggestion of an alternative pipeline using amino acid-based screening followed by de novo assembly, but after considering it, we determined that it while such a pipeline might be best practice for other applications (e.g., metagenomic or gut microbiome analyses, perhaps), it was not practical to implement here for the following reasons:

- 1) De novo assembly of viral reads from blood WGS data—which comprised the bulk of the data we analyzed—is rarely possible due to the sparsity of viral reads, as previously observed by Moustafa et al. (see response to Comment 3 above).

- 2) Reanalyzing all unmapped reads from ~900,000 genomes would incur substantial computational cost: in addition to CPU and RAM costs, the cost to re-access sequencing reads in AoU (which requires data retrieval from nearline storage) would alone be several thousand dollars.
- 3) An amino acid-based search would lose viral DNA sequences that originate in intergenic or noncoding transcribed sequences. This loss of reads would offset and potentially exceed gains from more completely capturing viral genetic heterogeneity.

6. The total numbers and prevalence of each detected virus across the dataset should be clearly reported as well as being plotted in a graph. i.e. the raw data are missing and should be provided as a dataset (along with clear metrics, e.g. reads per million)

We agree that the raw data underlying **Fig. 1c,e** and **Supplementary Fig. 1b** (prevalence and abundance) are of interest, and we have now reported these numbers for each cohort in the new **Supplementary Tables 2-5**.

7. Sequencing methods. The methods for generating viral reads require clarification—specifically, the wet-lab preparation protocol, any depletion or enrichment steps, and the sequencing platform(s) used. If human-targeted enrichment was employed, viral reads may be underestimated, and this limitation should be acknowledged as a potential bias

We agree that this information is important and have provided it in Methods (“UK Biobank, SPARK, and All of Us WGS data”; p. 42). No depletion or enrichment was done, and all extracted DNA was directly used in PCR-free WGS library preparation.

8. Clinical metadata. The reliability and coding of clinical conditions should be clarified. How systematically were diagnoses collected across cohorts—through medical records (e.g. “Do you have MS? Y/N”) or self-reporting? Were definitions and data completeness consistent between datasets? I'd have liked to see absolute numbers here also e.g. how many cases of MS were actually present in the dataset and how many of Hodgkins.

For UKB (used in PheWAS), we analyzed “first occurrence” clinical phenotypes that were generated by UK Biobank by aggregating: 1) Read code information in the Primary Care data (Category 3000), 2) ICD-9 and ICD-10 codes in the Hospital inpatient data (Category 2000), 3) ICD-10 codes in Death Register records (Field 40001, Field 40002), and 4) Self-reported medical condition codes (Field 20002) reported at the baseline or subsequent UK Biobank assessment centre visit. We also analyzed cancer registry data comprising ICD codes from national cancer registries that centralize information from separate regional cancer centres around the UK. We have summarized this information in Methods (p. 50), indicating the relevant UK Biobank Category or Field (from which precise documentation can easily be found in the UKB Data Showcase).

For the FinnGen+MVP+UKB meta-analyzed GWAS summary statistics (used in Mendelian randomization), FinnGen phenotypes were harmonized over ICD-8, -9 and -10, cancer-specific

ICD-O-3, (NOMESCO) procedure codes, Finnish-specific Social Insurance Institute (KELA) drug reimbursement codes and ATC-codes collected from various registries. Data from these registries were aggregated using national personal identification numbers assigned to all Finnish citizens and residents. MVP phenotypes were defined using ICD-9 and -10 codes from electronic health records grouped into corresponding phecodes, with case status defined as having two or more phecode-mapped ICD-9 or -10 codes. Meta-analyses were done by identifying phenotypes with concordant endpoints, as described at <https://finngen.gitbook.io/documentation/methods/meta-analysis>. We have added this information to Methods (p. 51).

We have added case numbers for all analyzed UKB clinical phenotypes in **Supplementary Tables 10** and **11**, and we have added case numbers from the FinnGen+MVP+UKB meta-analysis in **Supplementary Table 15**.

9. Was medication information available? Again, if not, it will be important to acknowledge that immune suppressive drugs in particular could cause reactivation - and might bias interpretation of the data.

Self-reported medication data was available in UKB, so we tested viral load phenotypes for association with use of corticosteroids, methotrexate, and cyclosporin A (each defined by aggregating common drug names for the medication). We observed significant or near-significant associations of all three medications with increased load of EBV and anelloviruses. We have now reported these associations in the new **Supplementary Table 14**, and we have noted this potential explanation for associations with other phenotypes (Results, p. 9) given the difficulty of controlling for medication use, which is likely to be incompletely represented by self-report.

10. Coinfections Were any associations or interactions observed between coinfections?

This was an interesting question that we had not explored. To investigate, we first asked which pairs of viruses had correlated DNA positivity (i.e., tended to co-occur more often than by chance). We had power to evaluate this for the most commonly observed viruses (EBV, HHV-6B, HHV-7, and anelloviruses), among which we observed strong co-occurrence of anelloviruses with each other and mildly enriched co-occurrences of EBV with anelloviruses and HHV-6B with other herpesviruses (now displayed in **Supplementary Fig. 1j**):

We next searched for associations of clinical phenotypes with co-occurring pairs of viruses. This analysis was again very limited in power by the small numbers of samples with viral reads observed from two viruses, so we considered the pair of most prevalent DNA viruses (EBV and HHV-7) and a pair with stronger co-occurrence of viral DNA (HHV-7 and HHV-6B), and we asked whether simultaneous DNA positivity for both viruses was associated with any of the clinical phenotypes. The only significant association that emerged with was “Mental and behavioural disorders due to use of tobacco (F17)” with EBV and HHV-7 (OR = 1.36, log₁₀(p) = 12.59), but this seemed most likely to reflect a strong positive association with EBV (OR=1.58, log₁₀(p) = 255.98) that remained significant after dilution from a weaker protective effect on HHV-7 presence (OR=0.90, log₁₀(p) = 7.07). In general, we think this lack of observed associations mostly reflects insufficient statistical power.

As a side note, these observations of co-occurring viruses are perhaps best thought of not as “coinfections” per se, as nearly all individuals have been latently infected with both EBV and HHV-7 (90%+ each) and so most individuals are likely to be coinfecting; being positive for DNA from each virus is more an indicator of having high viral load for each.

11. The statement that the study “revealed a causal association between EBV viral load and increased risk of Hodgkin lymphoma” would be better phrased as “confirmed,” as causality cannot be inferred directly from this analysis.

We have amended our use of “causal” in the Abstract (p. 1):

“In contrast, Mendelian randomization supported a strong causal effect on increased risk of Hodgkin lymphoma”

and Introduction (p. 2):

“Analyses also identified polygenic effects of host genetic variation across the human genome on load of several viruses and supported a causal association between EBV viral load and increased risk of Hodgkin lymphoma.”

12. Prevalence in figures S1c and S1d. How many individuals were virus-positive? The total cohort size is 441,026, but the numerator for each virus is unclear. Figure S1c (prevalence of viral reads vs. lymphocyte %) interestingly shows increasing lymphocyte percentage with higher HHV-7 and EBV reads, but not with TTV or HHV-6B. Figure S1d (reticulocyte count vs. parvovirus reads) appears to show a non-linear decline in reticulocyte count; clarification would be helpful.

We agree that information about the number of virus-positive individuals for each cohort was needed, and we have now provided these numbers in **Supplementary Tables 2-5**.

Yes, the relationship between reticulocyte count and parvovirus reads appears to be non-linear, presumably reflecting an effect present during active B19V infection. We have explained this in Results (p. 3) and noted in the figure legend that the p-value is from a model in which both reticulocyte count and parvovirus reads were log-transformed.

13. Coverage plots (S2). For each virus shown and within each patient, how many individuals had detectable reads, and what was the range?

This information can now be found in **Supplementary Tables 2-5**.

14. HHV-6B and telomere loss. This is a naive question but could the decline in HHV-6B reads by age reflect loss of telomeric integration?

This was an interesting hypothesis: telomere shortening during aging could in theory either (1) cause loss of previously-integrated copies of HHV-6B or (2) reduce efficiency of new integration into shorter telomeres.

To evaluate (1), we examined carriers of endogenous HHV-6, in whom the inherited copy of HHV-6 would be subject to such an effect in all cells throughout life, and for whom we could precisely quantify HHV-6 abundance (see response to Comment 4 above). Among individuals who inherited a single copy of either HHV-6A or HHV-6B, we observed no effect of age on HHV-6 abundance ($p=0.83$ and $p=0.36$), suggesting that (1) is unlikely.

We could not directly evaluate (2), but we found a recent experimental study that evaluated HHV-6A integration efficiency into HeLa cells having short or long telomeres and observed that HHV-6A could efficiently integrate into telomeres independent of their length (Wight et al. 2022; PMID: 36146670). This suggests (2) might also be unlikely.

Finally, we also analyzed leukocyte telomere length measurements previously generated in UKB (Codd et al. 2021; PMID: 34611362). We observed a modest association of telomere length with HHV-6B load ($\beta=0.0025$; $p=1.67 \times 10^{-8}$, controlling for sex, assessment center, and genetic principal components), but this association was attenuated upon adjusting for age and

age squared ($\beta=0.0017$; $p=0.00019$), suggesting that the residual association may reflect some other aspect of shared biology.

15. Figure S3 (and elsewhere) Figure legends should state the absolute numbers of individuals positive for each virus.

We have added this information to the legend of **Supplementary Fig. 3** and the other figures (in addition to providing these numbers for all cohorts in **Supplementary Tables 2-5**).

16. Seasonal variation The temporal distribution of EBV reads by month (S3f) was interesting but not statistically significant. Were participants all from temperate regions with distinct seasonal variation?

This supplementary figure (now **Supplementary Fig. 3m**) presented data for saliva ($n=18,751$); the seasonal variation is consistent and much clearer in the blood-derived WGS data (**Fig. 2ef**; $n=490,136$), such that the lack of statistical significance in saliva probably just reflects insufficient power. We apologize that the plots in **Supplementary Fig. 3** were not clearly labeled as blood vs. saliva and have now clarified this.

Regarding the question of whether participants were from temperate regions with distinct seasonal variation, the vast majority of AoU participants live in a temperate climate zone with seasonal variation (e.g., 99.9% live above 25°N).

17. Diurnal and seasonal lymphocyte variation The variation in lymphocyte percentage by month and hour (S3g, S3h) is intriguing and I think perhaps has been previously observed in other settings e.g. during exercise. The datasets used and absolute numbers should be specified in the legends.

We have clarified in the main text referencing these panels (now **Supplementary Fig. 3n,o**) that these measurements of lymphocyte percentage were from UK Biobank blood samples, and we provided the sample size ($n=472,115$) in the legend as suggested.

18 Ethnicity (S4) For each figure, the number of samples included and the number testing positive should be provided.

We have revised each figure legend to state the sample sizes, and we have provided absolute numbers of positive individuals for each virus for each cohort in **Supplementary Tables 2-5** as well as subdivided by ancestry in **Supplementary Table 6**.

19. Nomenclature ICTV virus nomenclature should be included (minor point).

Thank you for this suggestion. We have added this information in **Supplementary Table 1**.

20. Supplementary Figure S8 Absolute numbers appear to be missing and should be added.

We have added sample numbers for each virus to the figure legend.

21. Supplementary Figure S9 The association with class II HLA alleles is interesting—does this relate to receptor differences or to CD4⁺ T-cell-mediated immunity?

Although we hypothesize that *DRB1*04:04* might be associated with EBV load due to a receptor difference that generates higher affinity for EBV (given that *DRB1*04:04* is a strong risk factor for increased EBV load, whereas effects from CD4⁺ T-cell-mediated immunity would probably manifest more as protective alleles), we think the interactions between HLA alleles shown in **Supplementary Fig. 9** could be through either mechanism. We are not sure of a way to disentangle the two possibilities without in vitro/in vivo experiments.

22. Supplementary Figure S10d The geographic variation in EBV DNA prevalence is notable. Does this also correlate with MS prevalence or with viral load by longitude?

These were interesting questions to explore, but we were underpowered to draw any conclusions. For MS, we did not observe evidence of analogous geographic variation ($p = 0.91$ for association with latitude), but power was limited due to the small number of MS cases in UKB ($n=2,419$; **Supplementary Table 10**). For EBV viral load, we likewise did not observe an association with EBV load among EBV DNA-positive individuals ($p = 0.24$), but this was probably also an issue of power (as only 5.9% of the cohort had >1 EBV read pair). Given the high infection prevalence of EBV (90%+), the geographic variation of DNA positivity (~15% prevalence) is probably a proxy for viral load.

Summary and recommendation

This is a highly valuable and potentially landmark study linking host genetics with viral persistence across population-scale cohorts. The findings on host–virus interactions and genetic associations are compelling and will likely influence future studies of viral latency and immune control. However, the manuscript would benefit from additional methodological detail, clearer reporting of quantitative data (including prevalence and read depth), and improved consistency in figure legends. With these clarifications, the paper would represent a major contribution to our understanding of the human DNA virome.

We appreciate the very thorough review and agree that the suggested revisions have substantially improved the manuscript.

Referee #4 (Remarks to the Author):

This study presents a comprehensive survey of the viral load of 31 DNA viruses in human blood and saliva, leveraging whole-genome sequencing data from the UK Biobank (n=490,401), SPARK (n=12,519), and All of Us (n=414,817) cohorts. The authors observed marked variations in viral load across age, time of day, and season, with higher loads generally found in men than in women for most viruses. They also identified genetic variants associated with viral load for certain viruses. Furthermore, the study provided evidence for a strong causal effect of Epstein–Barr virus (EBV) viral load on the risk of Hodgkin lymphoma. Overall, the principal strength of this work lies in its use of an unprecedented sample size to elucidate the influence of human genetic and environmental factors on a range of human DNA viruses.

We note that the methodologies applied for virus detection and the subsequent genetic (GWAS and mendelian randomization analyses) analyses, while appropriate, are not novel. The results are largely reported in a descriptive fashion, with limited interpretation of the underlying biological or clinical implications. Additionally, we have concerns regarding several conceptual, statistical, and methodological aspects of the work, which are outlined in the following points:

We appreciate the balanced review of our manuscript, which we agree derives its principal strength from the unprecedented size of the data sets analyzed. We also appreciate the specific conceptual, statistical, and methodological points noted below, and we have revised the manuscript accordingly to address them.

Additionally, to obtain further biological insights into the observed differences in viral load across ancestries, which we agree were reported in a largely descriptive fashion, we undertook an additional set of GWAS for EBV and HHV-7 viral load in African-ancestry AoU participants (n=77,573). These analyses uncovered a surprising new result: the *ACKR1* promoter variant that causes the Duffy-null phenotype appears to explain ~1/4 of the African- vs. European-ancestry difference in EBV load and ~1/2 of the difference in HHV-7 load:

This effect, which was recessive as expected, appeared to be partially but not completely due to the neutropenia that has previously been observed in Duffy-null individuals. We have now reported these results in Results (p. 7), **Fig. 4a**, **Supplementary Fig. 9a-c**, and **Supplementary Note 5**).

Major concerns:

1. The scope of the viral analysis substantially limits the impact and interpretation of this study.

The profiling of only 31 DNA viruses does not fully represent the "human virome", a term that encompasses a far greater diversity of viruses, including the entirely uninvestigated RNA viruses. This oversight, coupled with the well-established composition of the human virome (which includes both DNA and RNA viruses), makes the current title misleading. Readers are likely to anticipate findings relevant to established niches like the gut virome. Therefore, we strongly recommend revising the title to accurately reflect the analytical focus of the study. Furthermore, the manuscript must provide a clear and thorough rationale for the selection of these specific 31 DNA viruses.

Thank you for pointing out that the title was too broad. We agree and have revised the title to "Human genes and environments profoundly affect the DNA virome" as suggested by Referee #1, who also pointed out this issue.

We have also revised the Methods (p. 43) to explain the rationale for selecting the 31 profiled viruses, which we agree needed a clear explanation. The reason for this approach was that our main goal was to identify effects of human genetics, age, sex, and environmental exposures (such as smoking) on viral load, and we only had statistical power to detect such effects for commonly observable viruses. We therefore prioritized a smaller set of common viruses to profile (primarily comprising the viruses commonly detected in a previous analysis of 8,240 individuals (Moustafa et al.; ref. 13)). Working with a smaller set of common viruses allowed us to perform careful QC on WGS-based quantifications of viral load (**Supplementary Fig. 2**), which was important given the potential for read alignment artifacts.

2. Statistical support of significance needs to be provided.

The presentation of the results is largely descriptive and lacks essential P values. Key sections, such as "The DNA virome in 856,319 blood and 61,418 saliva samples" and "Viral load is shaped by age, sex, circadian and seasonal effects," report findings without providing associated P-values. To allow for a rigorous assessment of the claimed associations, it is imperative that P-values or adjusted P-values (e.g., for multiple testing) be provided for all statistical comparisons throughout the manuscript.

We agree that p-values were needed in many places and have now added p-values for all statistical comparisons throughout the manuscript (>20 places).

3. Circadian analysis

The observation of circadian and seasonal fluctuations in viral load, as suggested by the figures, is intriguing. However, the statistical methodology used to support these claims could be more robust. We recommend applying established methods specifically designed for circadian rhythm analysis, such as those implemented in the Kronos package (PMID: 39504963) or other referenced tools (e.g., PMID: 38569545), to validate these findings with greater rigor.

We have more rigorously evaluated the circadian and seasonal fluctuations with the Kronos package as suggested and confirmed the robustness of these claims (time of day, $p = 1.2 \times 10^{-9}$ and 3.3×10^{-23} for EBV and HHV-7; month of year, $p = 2.9 \times 10^{-101}$ and 9.8×10^{-12} for EBV and HHV-7). We have now reported these confirmatory findings in Results (p. 5).

4. Single or meta-analyses of the cohorts

The strategy for handling the multiple cohorts is inconsistent and unclear. The authors should justify the reasoning behind sometimes combining cohorts (lines 205-206) and sometimes analyzing them separately. Providing a justification for these choices is essential. Furthermore, we note that a formal meta-analysis was not performed across all cohorts, please give the reasons?

This was a good point that led us to re-evaluate our approach to handling the multiple cohorts and ultimately improve our GWAS power. We settled on the following harmonized approach (now explained in Results, p. 5):

- For GWAS of viral load in blood, we used the METAL software (Willer et al. 2010) to perform meta-analysis across UKB and AoU for all viruses (rather than just EBV). This substantially increased GWAS discovery (e.g., 37 loci for HHV-7 compared to 24 in our previous analysis of only UKB).
- For GWAS of viral load in saliva, we analyzed just AoU. We did not meta-analyze AoU with SPARK because the potential power gain was modest (given the much smaller size of the SPARK cohort) and might be negated by the age heterogeneity of SPARK (mostly children) versus AoU (only adults).

5. Validation of results in vivo or vitro models is desirable.

The authors reported that EBV viral load exhibited a strong causal effect on increased risk of Hodgkin lymphoma with Mendelian randomization analysis. We encourage the authors to discuss the possibility of validating this association in in vitro or in vivo models in future work, as such data would greatly strengthen the conclusion.

We appreciate this suggestion and have added a discussion of *in vitro* or *in vivo* models that could be used to confirm the proposed effect of EBV load on HL (Discussion, p. 11):

“This result could potentially be validated *in vitro* by exposing germinal center B-cells to increasing titers of EBV and measuring the number of Hodgkin and Reed-Sternberg (HRS) cells produced⁷⁸. Alternatively, this could be studied in humanized mice⁷⁹ if HRS generation requires a tissue niche.”

6. Heterogeneity between cohorts in data processing

In the method, the samples from different cohorts were sequenced in various coverage, and then the sequencing data were processed by different bioinformatic tools. How did the author handle these variations in the current manuscript?

This is a good question that needed to be clarified. We handled the heterogeneity in WGS data generation and processing (e.g., alignment to the human reference genome, which impacted which reads did and did not map to human chromosomes) in two ways:

- First, we performed QC for mapping artifacts across each viral genome in each cohort separately (UKB blood, AoU blood, SPARK saliva, AoU saliva). That is, for each virus, for each cohort, we identified and excluded regions of the viral genome with excessive coverage relative to the rest of its genome (indicating specific regions of the viral genome in which mismapped reads “pile up”). This procedure handled cohort-specific error modes of false positive viral alignments: for example, we observed certain 500bp bins in the EBV genome that were excessively covered in the AoU blood-derived samples, but not UKB blood-derived samples nor saliva-derived samples in AoU or SPARK. These regions were then masked in AoU blood samples, as presumably something specific to processing of blood samples in AoU generated a higher rate of false positive alignments to these regions.
- Second, we inverse-normal transformed viral load measurements (based on counts of aligned read pairs) within each cohort before performing GWAS. This helped calibrate the scale of measurements in cohorts with different sequencing coverage (though heterogeneity in sequencing coverage was ultimately a minor consideration, as mean coverage was not dramatically different across cohorts: 32.5x to 42x).

We have explained these procedures in Methods (p. 44), and we have also confirmed that effect sizes of GWAS hits for EBV, HHV-6B, and HHV-7 were highly concordant in UKB and AoU, indicating effective handling of cross-cohort heterogeneity (**Supplementary Fig. 7c-e**):

Minor concerns:

- **Line 172: it is better to give the exact time of morning and evening.**

We have specified the exact times (9AM to 8PM).

- **The parameters of the bioinformatic tools that the authors used should be given, e.g., line 1121-1122.**

We have revised Methods to include any additional flags or parameters used when running each of the bioinformatic tools.

- **Line 385: provide the full name of MR analyses**

We have provided the full names of the MR analyses in the legend of **Fig. 6**.

- **Line 443: the human virome need to be rephrased as only 31 viruses were analyzed.**

We have rewritten this sentence as follows:

“Our analyses of human viruses commonly observable from population DNA sequencing data had several limitations.”

- **The number of samples need to be given in figures, such figure 2 for each covariate and time points, as well as the P values as mentioned above.**

We have added sample numbers for **Figure 2** and all other figures along with p-values as mentioned above.

Referee #1 (Remarks to the Author):

I have reviewed the revised manuscript and commend the authors for their thorough and thoughtful response to the previous round of comments. The vast majority of my earlier concerns have been addressed. However, a few minor issues remain, detailed below:

1. Regarding the two overlapping studies: they are now cited, the discrepancy in MR results is discussed, including the key point about excluding the MHC region. Still, the paper would benefit from explicitly positioning what is new relative to these studies beyond a general “broader scope” statement: several novelty points emphasized in the response letter are not transparently laid out in the manuscript itself (e.g., inclusion of saliva, breadth across 31 viruses, and any EBV-specific findings the authors consider uniquely supported here).

We have added text to the Discussion to clarify what our study has done uniquely compared to these studies (p. 9).

2. Validity of viral load estimation: in their response, the authors convincingly show that the viral signal is not dominated by noise. The detailed numerical robustness results (fractions of affected individuals per virus and near-identical association p-values) are not clearly documented in the manuscript, so these points currently read as asserted rather than directly supported in the paper. Clearer definitions and documentation in the text would be helpful, e.g., define what “false positive rate” means here (per-read vs per-individual positivity), clarify the logic behind interpreting EBV seronegative-with-reads cases, and point explicitly to where the MAPQ sensitivity outcomes are shown (or briefly summarize the key numbers in Supplement).

We agree that these results would be useful to include to demonstrate direct support and have added new Supplementary Notes 1 and 2. We have also clarified what “false positive rate” means in the first Supplementary Note.

3. Impact of circadian and seasonal effects on viral load: The response is satisfactory but the key elements have not been incorporated into the revised manuscript (or supplement): (i) day-of-week sensitivity analysis, and (ii) assessment-center “outlier” screen - even if the conclusion is that no exclusions were warranted. I would appreciate to see 1) a brief description of these sensitivity analyses in the Methods or Supplement and 2) a sentence acknowledging that confounding by unmeasured visit-related factors cannot be fully excluded, even with the above sensitivity analyses.

We have added these results to a new Supplementary Note 3 and added a statement acknowledging that these do not fully exclude the possibility of confounding by unmeasured factors (Results, p. 4).

Referee #3 (Remarks to the Author):

I am satisfied that the authors have answered the majority of my questions very comprehensively and those of the other reviewers.

There is only one comment that I would make which is that my suggestion for further analysis of viral reads was misunderstood. The suggestion was not to look at de novo assembly as per the author rebuttal, but in fact to look at raw fastq reads using diamond blastx (suggested because it is highly sensitive, substantially faster and less computationally intensive than blastn) or kmer analysis of those reads followed by mapping. I agree fully that de novo assembly is associated with a far lower sensitivity than mapping and that de novo assembly followed by blast would not add to the analysis.

However, I am persuaded that this request would be expensive and time-consuming, and while the authors may have missed some viruses in their analysis (which can be >30% divergent between genotypes), it does not reduce the validity of the findings as they currently stand. This issue is more problematic with viral genomes than mammalian (or indeed bacterial) genomes due to far higher levels of genetic diversity.

There were also some gaps in viruses selected for mapping - eg AAVs.

I would be satisfied if the authors were to acknowledge these limitations.

We apologize for the misunderstanding and have added both of these limitations of the present study to the Discussion (p. 10) and explain in more detail what could be done in future work in a new Supplementary Note 10.

Referee #4 (Remarks to the Author):

Thank you for the opportunity to review the revised version of this manuscript. I have thoroughly examined the manuscript again and am pleased to find that all of my previous concerns have been fully addressed. I have no further major comments on the study. Below are only some minor formatting suggestions for your consideration.

In figures where sample numbers are presented as a range (e.g., the age bins in Figure 2), could the exact sample size (n) be indicated in the figure labels? For instance, labeling as "0-1 year (n=xxx)" would enhance clarity.

We considered incorporating per-bin sample sizes into the figure as suggested, but we determined that listing every single n would decrease readability (e.g., 20 numbers would need to be reported in Fig. 2a). We believe the 95% CIs shown in the figure, together with the sample sizes listed, are sufficiently informative of statistical resolution.

Line 90: high-coverage need to be specified

We have added the range of mean coverages across included cohorts to this line.

Line159-160: figure 2b shows prevalence not abundance, right?

Yes, that is correct. We have clarified the text to make this clearer:

“In blood samples from UKB, which recruited participants 40–70 years old, viral DNA prevalence of EBV and TTVs was greater in older individuals ($p = 6.7 \times 10^{-356}$ and 1.4×10^{-451} ; regression), as recently observed for TTVs¹⁴, whereas viral DNA prevalence was greater in younger individuals for HHV-7 ($p = 6.7 \times 10^{-200}$) and HHV-6B ($p = 1.7 \times 10^{-28}$), suggesting that viral load of HHV-7 and HHV-6B decline with age (Fig. 2b).”

Line 178: P value should be added in figure 2a and figure 2b for each time point.

We considered adding p-values for each time point, but this would amount to reporting 80 p-values in Fig. 2a and 48 p-values in Fig. 2b, which would not be readable. We believe the 95% CIs shown in the figure provide sufficient information about the statistical resolution of these results.